# Compact Lie Groups, Generalised Euler Angles, and Applications

Sergio Luigi Cacciatori [1,2,*,†] and Antonio Scotti [3,†]

1 Dipartimento di Scienza e Alta Tecnologia, Università degli Studi dell'Insubria, Via Valleggio 11, 22100 Como, Italy
2 INFN, Via Celoria 16, 20133 Milano, Italy
3 Dipartimento di Matematica, Università degli Studi di Milano, Via Saldini 50, 20133 Milano, Italy
* Correspondence: sergio.cacciatori@uninsubria.it
† These authors contributed equally to this work.

**Abstract:** This is mainly a review of an intense 15-year long collaboration between the authors on explicit realisations of compact Lie groups and their applications. Starting with an elementary example, we will illustrate the main idea at the foundation of the generalisation of the Euler parametrisation of $SU(2)$ to any compact Lie group. Based on this, we will provide a very detailed reconstruction of the possible Euler parametrisation associated with the so-called symmetric embedding. Then, we will recall how such constructions are related to the Dyson integrals, providing a geometrical interpretation of the latter, at least in certain cases. This includes a short review on the main properties of simple Lie groups, algebras, and their representations. Finally, we will conclude with some applications to nuclear physics and to measure theory in infinite dimensions and discuss some open questions.

**Keywords:** Lie groups; Euler angles; representations; Dyson integrals; symmetric spaces

## 1. Introduction

Everyone knows how the Euler parametrisation works for $SU(2)$ or $SO(3)$, but is there a natural generalisation to higher dimensional Lie groups? The answer is positive for every compact simple Lie group and indeed for any compact connected Lie group, see [1–10], and specific standard constructions for orthogonal and unitary groups are well-known, see for example [11–15]. Here, we want to illustrate the idea working with the compact simple Lie group $SU(3)$. The starting point is to look for a proper maximal Lie subgroup. Proper means that it is neither the unit element nor the whole group. Maximal means that it is not contained in a larger proper subgroup. For a generic given group $G$, there can exist several maximal proper subgroups. They have been completely classified by Dynkin, see [16]. However, we do not necessarily need to choose among all possible proper maximal subgroups, just among the symmetrically embedded proper maximal subgroups. This means what follows. Let $G$ be a connected compact Lie group of dimension $n$ and $K$ a proper maximal Lie subgroup of dimension $k < d$. Then, consider the respective Lie algebras. Therefore, we can choose a linear basis $\{t_1, \ldots, t_k\}$ spanning $\boldsymbol{k} = Lie(\boldsymbol{K})$ and extend it to a basis for the whole $Lie(\boldsymbol{G})$ by adding the generators $\{p_1, \ldots, p_{n-k}\}$. Let $\boldsymbol{p}$ be the linear subspace of $Lie(\boldsymbol{G})$ spanned by the generators $p_j$. Then, $Lie(\boldsymbol{G}) = \boldsymbol{k} \oplus \boldsymbol{p}$. We say that $K$ is symmetrically embedded in $G$ if

$$[\boldsymbol{k}, \boldsymbol{k}] \subseteq \boldsymbol{k}, \tag{1}$$

$$[\boldsymbol{k}, \boldsymbol{p}] \subseteq \boldsymbol{p}, \tag{2}$$

$$[\boldsymbol{p}, \boldsymbol{p}] \subseteq \boldsymbol{k}. \tag{3}$$

The first relation is obviously true for every subgroup. The second relation states that $\boldsymbol{p}$ is a space of representation for $\boldsymbol{k}$ (and then for $\boldsymbol{K}$). The third relation can be roughly

stated by saying that the elements of $\boldsymbol{p}$ are in some sense square roots of the elements of $\boldsymbol{k}$. However, its more important property is not that one. Indeed, first notice that $\boldsymbol{G}$, being compact, is a real group, so that $\boldsymbol{k}$ and $\boldsymbol{p}$ are real vector spaces. Now, the space $\tilde{\boldsymbol{p}} = i\boldsymbol{p}$, is again a real linear space, and the second relation above is invariant under the replacement $\boldsymbol{p} \mapsto \tilde{\boldsymbol{p}}$. The third relation ensures that the whole Lie algebra remains real despite the multiplication by an imaginary unit. This way, with the above substitution, we obtain a new real algebra and, after exponentiating, a new real group, which now will be a non-compact group. However, this is not yet what we want to do; we just needed the notion of symmetric embedding. This diminishes the possible subgroups to be considered but in general does not individuate a unique choice for the subgroup we are looking for.

Let us now work with the explicit example of $\boldsymbol{SU}(3)$. Its Lie algebra consists of the traceless $3 \times 3$ anti-hermitian matrices of which a canonical basis is given by the (anti-hermitian) Gell-Mann matrices:

$$
\lambda_1 = \begin{pmatrix} 0 & i & 0 \\ i & 0 & 0 \\ 0 & 0 & 0 \end{pmatrix}, \quad
\lambda_2 = \begin{pmatrix} 0 & 1 & 0 \\ -1 & 0 & 0 \\ 0 & 0 & 0 \end{pmatrix}, \quad
\lambda_3 = \begin{pmatrix} i & 0 & 0 \\ 0 & -i & 0 \\ 0 & 0 & 0 \end{pmatrix},
$$

$$
\lambda_4 = \begin{pmatrix} 0 & 0 & i \\ 0 & 0 & 0 \\ i & 0 & 0 \end{pmatrix}, \quad
\lambda_5 = \begin{pmatrix} 0 & 0 & 1 \\ 0 & 0 & 0 \\ -1 & 0 & 0 \end{pmatrix}, \quad
\lambda_6 = \begin{pmatrix} 0 & 0 & 0 \\ 0 & 0 & i \\ 0 & i & 0 \end{pmatrix}, \tag{4}
$$

$$
\lambda_7 = \begin{pmatrix} 0 & 0 & 0 \\ 0 & 0 & 1 \\ 0 & -1 & 0 \end{pmatrix}, \quad
\lambda_8 = \frac{1}{\sqrt{3}} \begin{pmatrix} i & 0 & 0 \\ 0 & i & 0 \\ 0 & 0 & -2i \end{pmatrix}.
$$

This is an algebra of rank 2, which substantially means that its maximal abelian subalgebras (with some regularity properties made precise in Section 3) have dimension 2. These are the Cartan subalgebras and are all equivalent. An example is given by the linear space generated by $\lambda_3$ and $\lambda_8$ that evidently commute, but we could also choose the one generated by $\lambda_1$ and $\lambda_8$ or by $\lambda_2$ and $\lambda_8$. Let us fix the first choice, which is the canonical one.

Now, this Lie algebra, $\mathfrak{su}(3)$, contains two different symmetrically embedded proper maximal Lie subalgebras, which, for the sake of simplicity for now, we will simply call maximal subalgebras. The first one is generated by the four matrices, $\lambda_1, \lambda_2, \lambda_3$, and $\lambda_8$, that identify a subalgebra of type $\mathfrak{u}(2) = Lie(\boldsymbol{U}(2))$ of which the first three matrices generate an $\mathfrak{su}(2)$ subalgebra. The second maximal subalgebra is generated by $\lambda_2, \lambda_5$, and $\lambda_7$, and it is of type $\mathfrak{so}(3)$. The reader can easily verify that both of these satisfy the conditions to be symmetrically embedded maximal subalgebras. Of course, one can find an infinity of other possible combinations of generators defining different maximal subalgebras, but he will find that if symmetrically embedded, they are each equivalent to one of the two prototypes above.

Notice that $\mathfrak{su}(2)$ and $\mathfrak{so}(3)$ are isomorphic algebras. Nevertheless, they are in two different representations since $\mathfrak{su}(2)$ is substantially generated by the (anti-hermitian version of the) Pauli matrices, whereas the second algebra is in the adjoint representation of $\mathfrak{su}(2)$. This means that in exponentiating $\mathfrak{su}(2)$ we obtain $\boldsymbol{SU}(2)$, whereas exponentiating $\mathfrak{so}(3)$, we obtain $\boldsymbol{SU}(2)/\mathbb{Z}_2 \simeq \boldsymbol{SO}(3)$ since the adjoint representation of $\boldsymbol{SU}(2)$ has kernel $\mathbb{Z}_2$. This is the reason why we have to keep them distinguished. Moreover, $\mathfrak{su}(2)$ is not maximal since it is properly contained in $\mathfrak{u}(2)$, whereas $\mathfrak{so}(3)$ is maximal.

We will call $\mathfrak{u}(2)$ the *largest maximal subalgebra*, whereas $\mathfrak{so}(3)$ is the *smallest maximal subalgebra* or also the *split subalgebra*. The same nomenclature extends to the corresponding groups. In general, there are several maximal subalgebras, among which there is always the smallest one (the split one) and the largest one. They are characterised by a rank, the rank associated with the split algebra being the same as the rank of the whole algebra (in the sense of Cartan). This rank is not the rank of the maximal subalgebra in the sense of Cartan, but the rank of the quotient space. In order to understand what it means, let us go back

to the general case of $K \subset G$. We know that there is an infinite freedom in choosing a Cartan subalgebra in $Lie(G) = \boldsymbol{k} \oplus \boldsymbol{p}$. However, for any fixed $K$, there always exists a choice (and then infinite equivalent choices) for the Cartan subalgebra $\boldsymbol{h}$ of $Lie(G)$ such that the dimension

$$r_{\boldsymbol{G}/\boldsymbol{K}} = \dim(\boldsymbol{h} \cap \boldsymbol{p}) \tag{5}$$

is maximal. Here, $r_{\boldsymbol{G}/\boldsymbol{K}} \leq r = \mathrm{rank}(\boldsymbol{G})$ is called the rank of the symmetric space. Recall that $\boldsymbol{G}$ is also a manifold and $K$ a submanifold of it. The fact that $K$ acts on $G$ by multiplication allows us to construct the quotient $S = \boldsymbol{G}/\boldsymbol{K}$, which also results to be a manifold of dimension $d = n - k$. The elements of $\boldsymbol{p}$ represent tangent vectors on $S$ on which the isotropy group $K$ (which fixes the points of $S$) acts via (2). More in general, $G$ translates the points of $S$ by acting on itself ( for example, by left multiplication), and we can identify the elements of $\boldsymbol{p}$ as left-invariant tangent vector fields generating such translations on $S$. Relation (3) then states that the Lie bracket among two such fields is an infinitesimal element of the isotropy group $K$, or, in more physical words, that two infinitesimal translations composed in different orders differ by an infinitesimal "rotation". For these reasons, one says that $S$ is a symmetric space, and this also justifies the name of symmetric embedding we used for the maximal subalgebras (or subgroups).

Let us go back again to our example; $\mathfrak{so}(3)$ is the split subalgebra of $\mathfrak{su}(3)$ since we see that $\lambda_3, \lambda_8$ selects a Cartan subalgebra all contained in $\boldsymbol{p}$, so

$$r_{\boldsymbol{SU}(3)/\boldsymbol{SO}(3)} = 2 = r. \tag{6}$$

Instead, for $\mathfrak{u}(2)$ it is not possible to find a Cartan subalgebra entirely outside it. It results that

$$r_{\boldsymbol{SU}(3)/\boldsymbol{U}(2)} = 1 < 2 = r. \tag{7}$$

For example, we can take

$$\lambda_4 \in \boldsymbol{p}, \quad \text{and} \quad \tilde{\lambda} = \frac{\sqrt{3}}{2}\lambda_3 - \frac{1}{2}\lambda_8 \in \boldsymbol{k}$$

as generators of a Cartan subalgebra with maximal $\dim(\boldsymbol{h} \cap \boldsymbol{p}) = 1$.

Now, we pass to the construction of the generalised Euler angles for $\boldsymbol{SU}(3)$. As we will now see, there is not a natural choice like for $\boldsymbol{SU}(2)$, but we can construct an Euler parametrisation for each maximal subgroup. However, the construction obtained starting from the split maximal subgroup is in a sense the most faithful one to the original Euler construction. We will call it the *split Euler parametrisation*.

## 1.1. Euler Parametrisation of $\boldsymbol{SU}(2)$

Recall that the Euler parametrisation of $\boldsymbol{SU}(2)$ can be written, for example, in the form

$$\boldsymbol{SU}(2)[\phi, \theta, \psi] = e^{\phi\sigma_3}e^{\theta\sigma_1}e^{\psi\sigma_3}, \tag{8}$$

where

$$\sigma_1 = \begin{pmatrix} 0 & i \\ i & 0 \end{pmatrix}, \qquad \sigma_2 = \begin{pmatrix} 0 & 1 \\ -1 & 0 \end{pmatrix}, \qquad \sigma_3 = \begin{pmatrix} i & 0 \\ 0 & -i \end{pmatrix}, \tag{9}$$

are the anti-hermitian versions of the Pauli matrices. We can think about

$$K[x] \equiv \boldsymbol{U}(1)[x] = e^{x\sigma_3} \tag{10}$$

as a parametrisation of the group $\boldsymbol{U}(1) \subset \boldsymbol{SU}(2)$. It is the unique maximal symmetric subgroup of $\boldsymbol{SU}(2)$. Of course, we could do the same with $\sigma_1$ (ore any given $\sigma \in \mathfrak{su}(2)$),

but we choose this for the way it appears in the Euler parametrisation and because only $\psi$ runs over a whole period, thus realising an entire $U(1)$. This way, we can identify $\sigma_1$ as the generator of a Cartan subalgebra all contained in $p$ according to the fact that we are in the split case. With $H[x] = e^{x\sigma_1}$, we can write

$$SU(2)[\phi, \theta, \psi] = \tilde{K}[\phi]H[\theta]K[\psi]. \tag{11}$$

Here, $\tilde{K}$ means that we are using the same parametrisation as for $K$, but the range of the parameter is reduced. This is due to the fact that $K \cap H = \mathbb{Z}_2$ is not trivial. Its generator is $K[\pi] = -I_2$. Thus, we have to quotient away from one of the two $K$ in order to avoid overcounting the points (since $K[\pi]$ can be freely moved from one to the other copy of $K$). Notice that $H$ is the centraliser of $H$ in $G$ in our example.

Finally, notice that the Cartan group $H = U(1)$ is an abelian torus of dimension $1 = r$. In order to guarantee to cover the whole group just once, the corresponding range of the coordinates, however, does not have to cover the whole torus, just a part called the fundamental region. First of all, we have to quotient it by $K \cap H = \mathbb{Z}_2$ since if $g \in K \cap H$, then we have $H[\theta]K[\psi] = H[\theta]g^{-1}gK[\psi] = H[\theta']K[\psi']$, and again, the parametrisation is redundant. We can solve it by reducing $H$ to $H/(K \cap H)$, so we work with a half torus. This, however, is not enough since there are elements in $K$ that are not in $H$ with respect to which $H$ is central; that is, $gHg^{-1} \in H$ for such a $g$. These elements form the so-called Weyl group $W(G)$, which coincides with the group of Weyl reflections acting on the roots. This means that we need to further reduce $H$ by the factor $W(G)$ in order to avoid redundancies. In our case of rank 1, there is only a non-trivial reflection generating $\mathbb{Z}_2$. Thus we finally obtain that for $SU(2)$ we have to take $\psi$ in the whole period, $\psi \in [0, 2\pi]$, $\phi$ in one-half of the period, $\phi[0, \pi]$, and $\theta$ in one-fourth of the period, $\theta \in [0, \pi/2]$.

One could repeat the reasoning by working with $SO(3)$ with the usual $3 \times 3$ matrices. The main difference is that now $K \cap H = I_3$ is trivial; therefore, we have only to quotient with respect to the Weyl group, which is the same since the algebra does not change, so the ranges are now $\psi \in [0, 2\pi)$, $\phi[0, 2\pi]$, and $\theta \in [0, \pi/2]$.

One may argue that such an interpretation of the Euler angles looks much too sophisticated and spoils the simplicity of the Euler parametrisation. This is of course true for $SU(2)$ and $SO(3)$, but it becomes the opposite for higher dimensional groups, exactly the general considerations that allow for obtaining a simple generalised Euler parametrisation of compact Lie groups. Let us see how it works for our prototype example.

### 1.2. Split Euler Parametrisation of $Su(3)$

We now repeat what we have learned for $SU(3)$ by choosing $K = SO(3)$ as the maximal subgroup. First, notice that

$$\dim G = 8 = 2 \cdot 3 + 2 = 2\dim K + r$$

so that a parametrisation of the form

$$SU(3)[\vec{x}, y_1, y_2, \vec{z}] = SO(3)[\vec{x}]H[y_1, y_2]SO(3)[\vec{z}],$$

mimicking the Euler parametrisation for $SO(3)$ is perfectly admissible. Here $\vec{w} = (w_1, w_2, w_3)$, and $SO(3)[\vec{w}]$ means any parametrisation of $SO(3)$, for example, the Euler one. We will see that the above dimensional relation is not accidental, but it holds for all groups $G$ if $K$ is the split maximal subgroup. It remains to individuate the discrete subgroups $\Gamma := H \cap SO(3)$ and the Weyl group. As we will see studying the general case, it happens that $\Gamma = \mathbb{Z}_2^r$ every time the maximal subgroup $K$ is simply connected. This is not the case for $SO(3) = SU(2)/\mathbb{Z}_2$, so that $\Gamma = \mathbb{Z}_2$ in this case. Its generator can be computed as follows.

Fix in $H$ (canonically isomorphic to its dual) the directions selected by the simple roots, see Section 1.4.1. They correspond to $y_2 = 0$ and to $y_2 = -\sqrt{3}y_1$, so select the elements

$$h_1[y] = e^{y\lambda_3}, \qquad h_2[y] = e^{y(-\frac{1}{2}\lambda_3 + \frac{\sqrt{3}}{2}\lambda_8)}.$$

They both have period $2\pi$. The generators of two possible $\mathbb{Z}_2$ are thus $\tau_1 = h_1[\pi]$ and $\tau_2 = h_2[\pi]$. However, $\tau_1$ is just the generator of the $\mathbb{Z}_2$ factor defining $SO(3) = SU(2)/\mathbb{Z}_2$, so we obtain that $\tau_2$ is the only generator of $\Gamma$. This implies that we have to reduce the range of the first $SO(3)$ factor, that is of the $\vec{x}$ parameters according to the action of $\Gamma$ on $SO(3)$. We are not interested in see exactly how it works since it is just a technical issue, useless for our purposes. The interested reader can easily check out the details by hand. On the torus $H$, we see that one has to reduce the ranges of $y_1$ and $y_2$ to half a period covering one-fourth of the torus.

We know that it is not enough; we have to reduce it further to a fundamental domain by dividing the quarter of torus into cells under the action of the Weyl group. One has that $W(SU(3)) = S_3$ is the permutation group of three elements, see [17]. It has cardinality six and then will subdivide the quarter of torus into six fundamental regions, all related by Weyl reflections. It is interesting to see how to individuate exactly such a fundamental region since here is exactly the point of connection with the Dyson integrals, the second topic of the present review. We postpone it to after a short discussion of the second parametrisation.

### 1.3. Non-Split Euler Parametrisation of $Su(3)$

The non-split case is a little bit more involved. There are many differences with respect to the previous case. First, we do not have the whole Cartan torus at our disposition since part of it is already contained in $K = \mathfrak{u}(2)$. So, we must work only with the one-dimensional sub-torus $U(1)[y] = e^{y\lambda_4}$ generated by $\lambda_4$, the part of the Cartan staying out of $K$. We see that now

$$\dim G = 8 < 2 \cdot 4 + 1 = 2\dim K + r_{G/K}.$$

Therefore, we cannot write the Euler parametrisation as before; otherwise we would obtain an enormous redundancy, too many parameters! This is due to the fact that the commutant of $U(1)[y]$ is quite large since it contains the whole $H$. The second factor of the two-dimensional torus $H$, which is $U(1)'[z] = e^{z\tilde{\lambda}}$, commutes with $U(1)[y]$ and can be moved from one $K$ factor to the other. This means that we can reduce the left $K$ factor to

$$\tilde{K} = K/U(1)'. \tag{12}$$

Thus, $\tilde{K}$ has dimension three and we have

$$\dim G = 8 = 3 + 4 + 1 = \dim(K/U(1)') + \dim K + r_{G/K}. \tag{13}$$

The dimension fits correctly now, and we just have to take discrete subgroups into account . The first one comes from the fact that $U(1)[y] \cap K = U(1)[y] \cap U(1)'[z] = U(1)[\pi]$, $2\pi$ being the period of $U(1)[y]$. This was already taken into account in the above quotient, but as in the previous section, this means that we have to quotient $U(1)[y]$ by the $\mathbb{Z}_2$ generated by $U(1)[\pi]$, thus reducing the period down to a semi-period. Finally, there is again the action of the Weyl group on the half torus. The torus is one-dimensional, and the Weyl group is the same as for $SU(2)$ so that the range of $y$ is further reduced to one-fourth of the period.

We do not want to enter further into details, but two comments are in order. First, in $\tilde{K} = U(2)/U(1)'$, the $U(1)'$ is not the same generated by $\lambda_8$ in $U(2)$. Thus, $\tilde{K}$ is not $SU(2)$ and not even a group since $U(1)'$ is not normal in $U(2)$. The second point is that in more general examples, the commutant of the central torus for non-split forms is larger than $H$ and is in general a non-abelian group.

### 1.4. Euler Parametrisation and Dyson Integrals

Let us now go back to the split parametrisation. We want to see how one can select the fundamental region inside the torus. In order to do that, it is convenient to further improve our knowledge of the details; we must analyse the structure in terms of the roots of the algebra and other interesting ingredients.

#### 1.4.1. The Roots Structure

We know that $\mathfrak{su}(3)$ has eight roots, two of which vanish. The non-vanishing roots can be easily computed, noting that if we define the six matrices

$$\mu_1^\pm = \frac{1}{2}(\lambda_1 \pm i\lambda_2), \quad \mu_2^\pm = \frac{1}{2}(\lambda_4 \pm i\lambda_5), \quad \mu_3^\pm = \frac{1}{2}(\lambda_6 \pm i\lambda_7), \tag{14}$$

we have

$$[\lambda_3, \mu_1^\pm] = \pm 2i\mu_1^\pm, \quad [\lambda_3, \mu_2^\pm] = \pm i\mu_2^\pm, \quad [\lambda_3, \mu_3^\pm] = \mp i\mu_3^\pm, \tag{15}$$

$$[\lambda_8, \mu_1^\pm] = 0, \quad [\lambda_8, \mu_2^\pm] = \pm i\sqrt{3}\mu_2^\pm, \quad [\lambda_8, \mu_3^\pm] = \pm i\sqrt{3}\mu_3^\pm. \tag{16}$$

The fact that all values are imaginary is a consequence of the fact that we are working with the compact form. In the dual basis $\lambda_3^*, \lambda_8^*$ defined by $\lambda_i^*(\lambda_j) = i\delta_{ij}$ for $i, j \in \{3, 8\}$, the roots have components

$$\pm\alpha_1 \equiv \pm(2, 0), \quad \pm\alpha_2 \equiv \pm(1, \sqrt{3}), \quad \pm\alpha_3 = \pm(-1, \sqrt{3}), \tag{17}$$

and belong to the vertices of a regular hexagon, see Figure 1. For example, $\alpha_1$ and $\alpha_3$ are a possible choice of simple roots. All other roots are linear combinations of these, with non-negative or non-positive integer coefficients. In particular, $\alpha_2$ is the longest root.[1]

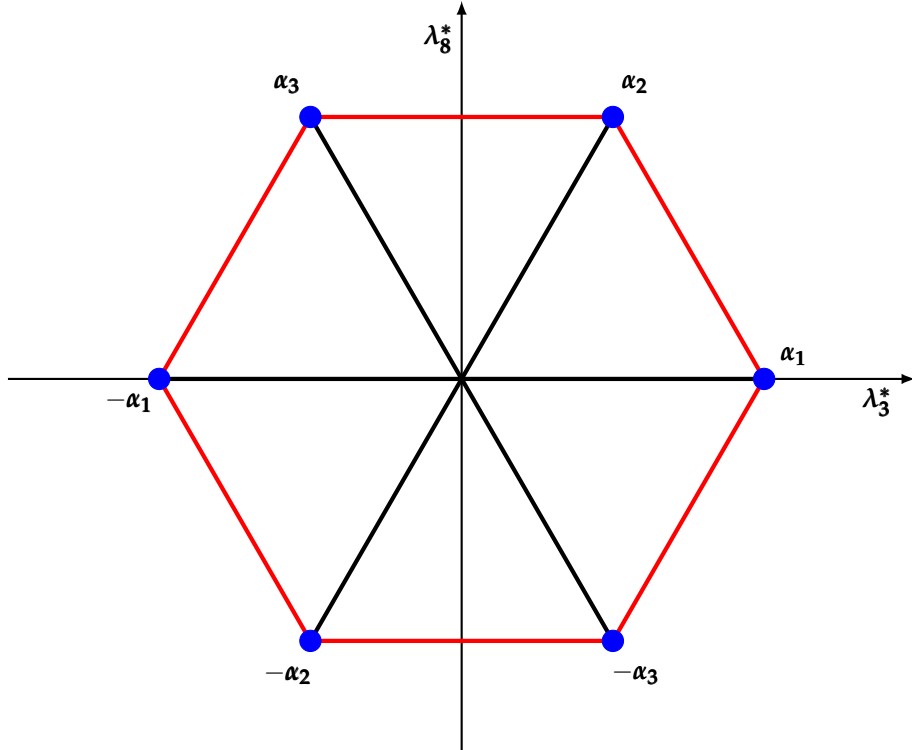

**Figure 1.** The non-vanishing roots of $\mathfrak{su}(3)$.

For the application we have in mind, it is useful to rewrite

$$\lambda_1 = \mu_1^+ + \mu_1^-, \quad \lambda_2 = i(\mu_1^- - \mu_1^+), \quad \lambda_4 = \mu_2^+ + \mu_2^-, \tag{18}$$

$$\lambda_5 = \mathrm{i}\,(\mu_2^- - \mu_2^+), \quad \lambda_6 = \mu_3^+ + \mu_3^-, \quad \lambda_7 = \mathrm{i}\,(\mu_3^- - \mu_3^+). \tag{19}$$

We will return to this later.

1.4.2. Invariant Measure

The next instrument we need is the invariant measure over $\boldsymbol{SU}(3)$. On the Lie algebra, it is defined as an invariant bilinear product by

$$\eta(a,b) = -\frac{1}{2}\mathrm{Tr}(ab), \quad a,b \in \mathfrak{su}(3). \tag{20}$$

It is a positive definite, and with respect to it, the Gell-Mann basis is orthonormal. By "invariant", we mean that it is invariant under the adjoint action of the group. Let $\boldsymbol{h} \equiv h[x_1 \ldots, x_8]$ be any parametrisation of $\boldsymbol{SU}(3)$. Then, $J = \boldsymbol{h}^{-1}d\boldsymbol{h}$ defines an $\mathfrak{su}(3)$-valued left-invariant one form. We can write it in the form

$$J = J^a_{\ b}(\underline{x})\lambda_a dx^b. \tag{21}$$

Inserting it in the invariant quadratic form $\eta$, we obtain an invariant metric $ds^2 = \eta(J,J)$ over $\boldsymbol{SU}(3)$, which is

$$ds^2 = J^a_{\ b}(\underline{x})J^c_{\ d}(\underline{x})\delta_{ac}dx^b \otimes dx^d. \tag{22}$$

This is like saying that the forms

$$J^a = J^a_{\ b}(\underline{x})dx^b \tag{23}$$

define a moving frame for the invariant metric; therefore, an invariant measure is simply given by

$$d\mu_{\boldsymbol{SU}(3)} = |\det(J^a_{\ b})| \prod_{i=1}^{8} dx^i. \tag{24}$$

Now, let us employ our split Euler parametrisation of $\boldsymbol{SU}(3)$ in the form

$$\boldsymbol{SU}(3)[\vec{x}, y_1, y_2, \vec{z}] = \tilde{\boldsymbol{K}}[\vec{x}]\boldsymbol{H}[y_1, y_2]\boldsymbol{K}[\vec{z}], \tag{25}$$

where $\boldsymbol{K}$ is the Euler parametrisation of $\boldsymbol{SO}(3)$. For the moment, we will not care about the action of the discrete subgroups. Compactly, for the invariant 1-form, we can write

$$J = \boldsymbol{K}^{-1}d\boldsymbol{K} + \boldsymbol{K}^{-1}(\boldsymbol{H}^{-1}d\boldsymbol{H} + \boldsymbol{H}^{-1}\tilde{\boldsymbol{K}}^{-1}d\tilde{\boldsymbol{K}}\boldsymbol{H})\boldsymbol{K}. \tag{26}$$

Let us define the two $3 \times 3$ matrices $M$ and $N$ as

$$M_{\alpha\beta}[\vec{z}] = \eta\left(\boldsymbol{K}^{-1}\frac{\partial \boldsymbol{K}}{\partial z^\beta}, \lambda_\alpha\right), \qquad \alpha = 2, 5, 7, \quad \beta = 1, 2, 3,$$

and

$$N_{\alpha\beta}[\vec{x}, y_1, y_2] = \eta\left(\boldsymbol{H}^{-1}\tilde{\boldsymbol{K}}^{-1}\frac{\partial \tilde{\boldsymbol{K}}}{\partial x^\beta}\boldsymbol{H}, \lambda_\alpha\right), \qquad \alpha = 1, 4, 6, \quad \beta = 1, 2, 3.$$

It is a simple exercise to prove that

$$\det(J^a_{\ b}) = \det M \det N. \tag{27}$$

Noticing that $|\det M|d^3z$ is the invariant measure of $\boldsymbol{SO}(3)$, we obtain

$$d\mu_{\boldsymbol{SU}(3)} = d\mu_{\boldsymbol{SO}(3)}[\vec{z}] \,|\det N[\vec{x}, y_1, y_2]|dy_1dy_2d^3x. \tag{28}$$

Now, $J_{\boldsymbol{K}}[\vec{x}] = \tilde{\boldsymbol{K}}^{-1}d\tilde{\boldsymbol{K}}$ is the invariant 1-form of $\boldsymbol{SO}(3)$ so that

$$\tilde{\boldsymbol{K}}^{-1}\frac{\partial\tilde{\boldsymbol{K}}}{\partial x^\beta} = \sum_{\alpha=2,5,7}(J_{\boldsymbol{K}})^\alpha{}_\beta\lambda_\alpha. \tag{29}$$

Therefore,

$$N_{\beta\tilde{\beta}} = \sum_{\alpha=2,5,7}(J_{\boldsymbol{K}})^\alpha{}_\beta\eta(\boldsymbol{H}^{-1}\lambda_\alpha\boldsymbol{H}, \lambda_{\tilde{\beta}}), \quad \tilde{\beta} = 1, 4, 6, \tag{30}$$

and

$$\det N_{\beta\tilde{\beta}} = \det[(J_{\boldsymbol{K}})^\alpha{}_\beta]\det\tilde{N}, \tag{31}$$

where

$$\tilde{N}_{\beta\tilde{\beta}} = \eta(\boldsymbol{H}^{-1}\lambda_\beta\boldsymbol{H}, \lambda_{\tilde{\beta}}), \quad \beta = 2, 5, 7, \ \tilde{\beta} = 1, 4, 6. \tag{32}$$

Thus, we see that the measure os $\boldsymbol{SU}(3)$ takes the form

$$d\mu_{\boldsymbol{SU}(3)} = d\mu_{\boldsymbol{SO}(3)}[\vec{z}] \, d\mu_{\boldsymbol{SO}(3)}[\vec{x}] \,|\det\tilde{N}[y_1, y_2]|dy_1dy_2, \tag{33}$$

and we are left with the calculation of $\det\tilde{N}$. This is the moment to use what we learned about the roots structure and, in particular, expressions (18) and (19). Using the fact that $\mu_\alpha^\pm$ are eigenmatrices for the adjoint action of $H$, setting $H = e^h$, we can write

$$e^{-h}\lambda_2e^h = \mathrm{i}\,(e^{\alpha_1(h)}\mu_1^- - e^{-\alpha_1(h)}\mu_1^+) = \cosh\alpha_1(h)\lambda_2 + \mathrm{i}\sinh\alpha_1(h)\lambda_1.$$

In the same way, we can compute $e^{-h}\lambda_\beta e^h$ for $\beta = 5, 7$ and obtain

$$\tilde{N} = i\begin{pmatrix} \sinh\alpha_1(h) & 0 & 0 \\ 0 & \sinh\alpha_2(h) & 0 \\ 0 & 0 & \sinh\alpha_3(h) \end{pmatrix}. \tag{34}$$

Finally, taking into account that $\alpha_j(h)$ are purely imaginary, we obtain

$$|\det\tilde{N}| = \prod_{j=1}^3\sin|\alpha_j(h)|. \tag{35}$$

If we introduce the "real" roots $\beta_j = -\mathrm{i}\alpha_j$, the invariant measure is thus

$$d\mu_{\boldsymbol{SU}(3)} = d\mu_{\boldsymbol{SO}(3)}[\vec{z}] \, d\mu_{\boldsymbol{SO}(3)}[\vec{x}] \prod_{j=1}^3\sin\beta_j(h)dy_1dy_2. \tag{36}$$

Essentially, this gives us all information on the fundamental region of the torus variables; it is defined by the inequalities

$$0 \le \beta_i(y_1\lambda_3 + y_2\lambda_8) = y_1\beta_i(\lambda_3) + y_2\beta_i(\lambda_8) \le \pi, \quad i = 1, 2, 3. \tag{37}$$

Before passing to the last step, some comments are in order.

First, $\beta_1$ and $\beta_3$ are simple roots, and $\beta_2 = \beta_1 + \beta_3$ is the longest root. Now, the simple roots are linearly independent, and this implies that the linear map

$$\vec{s} : \mathbb{R}^2 \longrightarrow \mathbb{R}^2, \quad s_i(y_1, y_2) := \beta_{2i-1}(h[y_1, y_2]), \quad i = 1, 2 \tag{38}$$

is an invertible transformation of coordinates. Thus, we can use $s_1$ and $s_2$ as coordinates. Notice that

$$dy_1 dy_2 = \frac{1}{\sqrt{3}} ds_1 ds_2. \tag{39}$$

In the $s_i$ coordinates, the range of the parameter is quite simple. They represent the simple coroot s directions and are constrained in a square with side length $\pi$. The third condition cuts the square along a diagonal. This is the tiling associated with the fundamental region. The choice of coordinates thus provides a universal characterisation of the fundamental region. They can be used to determine the correct range in any other coordinates system.

It is interesting to describe the fundamental region in terms of the original coordinates. Using the explicit values of the simple roots, the equations relative to the simple directions are

$$0 \leq 2y_1 \leq \pi, \qquad 0 \leq -y_1 + \sqrt{3}\, y_2 \leq \pi. \tag{40}$$

This is the parallelogram in Figure 2.

The third condition selects only one-half of the parallelogram, the coloured one in the figure. Notice that this is a small part of the range necessary to cover the whole torus. The period of $y_1$ is $2\pi$, and the period of $y_2$ is $2\pi\sqrt{3}/2$ (since we have to quotient $U(1)$ by $\mathbb{Z}_2$), so the area covering the torus is $2\sqrt{3}\pi^2$. The area of the fundamental region is $\pi^2/4\sqrt{3}$. It is 24 time smaller! A factor 2 arises from $\Gamma = H \cap SO(3) = \mathbb{Z}_2$. The remaining factor $12 = 2 \cdot 3!$ is a consequence of the fact that the Weyl group $W \equiv W(SU(3)) \equiv S_3$ has six elements, whereas the last factor 2 arises from the combination of $\Gamma$ and $W$, which do not commute.

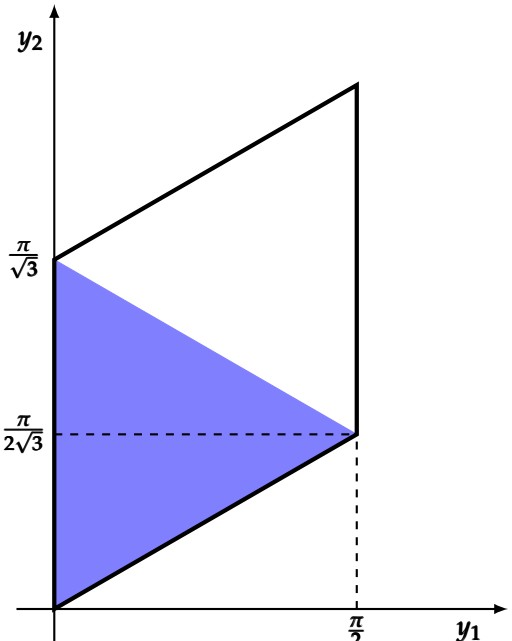

**Figure 2.** The parallelogram is the range determined by the simple coroots direction only. The coloured half is the fundamental region.

### 1.4.3. The Dyson Integral

We now know the invariant measure of $SU(3)$ and the range of parameters, so we can compute the volume of $SU(3)$. The measure (36) factorises in the product of the measures of two copies of $SO(3)$, one of which has to be quotiented with the action of $\Gamma$. If with $\Delta$ we mean the fundamental region, then we can write using the coordinates $s_i$:

$$Vol(SU(3)) = \frac{1}{2} Vol(SO(3))^2 \frac{1}{\sqrt{3}} \int_\Delta \sin s_1 \sin s_2 \sin(s_1 + s_2) ds_1 ds_2.$$

Apart from a normalisation factor that is not relevant to specify here—we will give a precise definition in the next section — the last integral is a *generalised Dyson integral*, which we call $I_D$. In this particular case, it is a quite simple integral and can be computed directly. In general, as we will see, such a kind of integral is not easy to compute. Nevertheless, from the above relation we can write

$$I_D = 2\sqrt{3} \frac{Vol(SU(3))}{Vol(SO(3))^2}. \tag{41}$$

Therefore, we can compute $I_D$ without integration, if we know the volume of the groups in some other way. Luckily, this can be computed quite easily; let us see how.

The matrices of $SO(3)$ are the orthogonal $3 \times 3$ matrices having determinant 1. This means that the three vectors forming such a matrix also form an oriented orthonormal system in $\mathbb{R}^3$. We can construct all of them as follows: fix the first row $e_1$. The second one, $e_2$, must then be chosen in the plane orthogonal to $e_1$, therefore along a circle. The angular measure of possibilities is $2\pi$. For each one of them, the third row is completely fixed by orthonormality and orientation (determinant 1). It remains to vary $e_1$ in all possible ways, which means along a sphere in $\mathbb{R}^3$, with angular measure $4\pi$. We thus deduce that the total angular measure of $SO(3)$ is $8\pi^2$. However, the invariant measure can also contain radial rescaling. To determine them, notice that the operations rotating the vectors we have now considered can be interpreted as compositions of the actions of the elements $h_i[x] = e^{x\lambda_i}$, $i = 2, 5, 7$. Since $\eta(\lambda_i, \lambda_j) = \delta_{ij}$ and $h_a[x]$ have period $2\pi$ for $a = 2, 5, 7$, $\eta$ associates length $2\pi$ to the orbits and therefore, radius 1. Therefore

$$Vol(SO(3)) = 8\pi^2. \tag{42}$$

Similarly, the matrices of $SU(3)$ are unitary with determinant 1. Their rows are an oriented orthonormal basis in $\mathbb{C}^3$. Fixing the first row $e_1$, we can choose the second one in the $\mathbb{C}^2$ subspace orthogonal to $e_1$ in any point of the sphere of modulus 1, which is an $S^3$ having volume $2\pi^2$. After that, the third row is completely fixed. It remains to fix $e_1$ on a sphere of radius 1 in $\mathbb{C}^3$, which is an $S^5$ with angular volume[2] $\pi^3$, so that the angular volume of $SU(3)$ is $2\pi^5$. As above, we can understand the radii rescaling by looking at the orbits generated by the orthonormal basis. These are $2\pi$ for all but one matrix since $e^{x\lambda_8}$ has period $2\sqrt{3}\pi$, and we have to quotient it by $\mathbb{Z}_2$. It corresponds to a radius $\sqrt{3}/2$ in place of 1. Thus, considering a radial rescaling only along such direction, we obtain

$$Vol(SU(3)) = \sqrt{3}\pi^5. \tag{43}$$

Formulas of this kind have been found for every simple Lie group by Macdonald [18]. Applying the Macdonald's formulas just obtained to the above expression for $I_D$, we obtain

$$I_D = \frac{3\pi}{8}. \tag{44}$$

We invite the reader to verify that it is indeed the right result by directly computing the double integral.

Our aim is to show that such a kind of connection between simple Lie groups and generalised Dyson integrals exists for every compact simple Lie group and for every kind of

Euler parametrisation starting from symmetrically embedded maximal proper subgroups. The groups indeed provide a subclass of the largest family of Dyson integrals as generalised by Macdonald (and proved by Opdam), defined in the next section. These are the ones constructed starting not from general lattices, but from the reduced lattices associated with symmetric spaces. It could be interesting to see if such geometrical analysis can be extended to cover the whole family.

Our general constructions are presented in Sections 5–7, while Sections 2–4 contain a detailed presentation of the well-known background material necessary to understand our constructions.

## 2. Macdonald's Conjecture

In this section, we provide a general description of the Dyson integrals and the Macdonald's conjecture. Our exposition is taken from un unpublished version of [1] that can be found on arXiv at [arXiv:1207.1262 [math.GR]].

We summarize the basic steps at the origin of Macdonald's conjecture following the clear and punctual paper of P. J. Forrester and S. O. Warnaar [19], to which we refer for a more extensive introduction. The story of Macdonald's conjecture begins in the 1940s in the paper of Atle Selberg "Über einen Satz von A. Gelfond" [20] where the author considered what is now known as the Selberg integral:

$$S_n(\alpha, \beta, \gamma) := \int_0^1 \cdots \int_0^1 \prod_{i=1}^n t_i^{\alpha-1}(1-t_i)^{\beta-1} \prod_{1 \le i < j \le n} |t_i - t_j|^{2\gamma} dt_1 \cdots dt_n$$

$$= \prod_{j=0}^{n-1} \frac{\Gamma(\alpha + j\gamma)\Gamma(\beta + j\gamma)\Gamma(1 + (j+1)\gamma)}{\Gamma(\alpha + \beta + (n+j-1)\gamma)\Gamma(1+\gamma)}. \tag{45}$$

This integral is valid for complex $\alpha$, $\beta$, and $\gamma$ such that:

$$\Re(\alpha) > 0, \qquad \Re(\beta) > 0, \qquad \Re(\gamma) > -\min\left\{\frac{1}{n}, \frac{\Re(\alpha)}{n-1}, \frac{\Re(\beta)}{n-1}\right\}, \tag{46}$$

corresponding to the domain of convergence of the integral. To limit the length of the paper, Selberg did not present the proof of his claim there, but he included it three years later, in 1944, in the work "Bemerkninger om et multiplet integral" [21]. Notice that the Euler beta integral is itself a Selberg integral with $n = 1$.

For over thirty years, the Selberg integral was essentially unnoticed. The exception was a study by S. Karlin and L. S. Shapley in 1953 [22], where they considered the special case $\alpha = 1$, $\beta = 1$, and $\gamma = 2$ in relation to the volume of a certain momentum space. However, in the 1960s there were good reasons to make use of (45). F. J. Dyson wrote a series of papers in the context of the statistical theory of energy levels of complex systems [23–27]. A part of this series was written jointly with M. L. Mehta and published in 1963. Here, random Hermitian matrices were used to model highly excited states of complex nuclei. They considered systems with different symmetries described by matrices with real complex or real quaternion elements. The ensembles of random matrices are called Gaussian orthogonal (GOE), unitary (GUE), and symplectic ensembles (GSE). The joint probability density function for the three ensembles can be computed explicitly as:

$$\mathcal{P}(t_1, \ldots, t_n) = \frac{1}{(2\pi)^{n/2} F_n(\beta/2)} \prod_{i=1}^n e^{-t_i^2/2} \prod_{1 \le i < j \le n} |t_i - t_j|^\beta, \tag{47}$$

where $\beta = 1, 2, 4$ for the GOE, GUE, and GSE, respectively, and $F_n$ is the normalization

$$F_n(\gamma) := \frac{1}{(2\pi)^{n/2}} \int_{-\infty}^\infty \cdots \int_{-\infty}^\infty \prod_{i=1}^n e^{-t_i^2/2} \prod_{1 \le i < j \le n} |t_i - t_j|^{2\gamma} dt_1 \cdots dt_n, \tag{48}$$

referred to as Mehta's integral, see [28]. Here, $t_j$ varies in the space of admissible eigenvalues for the given class of matrices. In [27], Mehta and Dyson evaluated $F_n(\beta/2)$ for each of the three special values of $\beta$. Combining this with the evaluations for $n = 2$ and $n = 3$ for general $\beta$, led them to conjecture that

$$F_n(\gamma) = \prod_{j=1}^{n} \frac{\Gamma(1 + j\gamma)}{\Gamma(1 + \gamma)}. \tag{49}$$

The conjecture (49) can be proved evaluating Mehta's integral using the Selberg integral; however, in 1963 the Selberg's result was still unknown. The proof was finally given in the late 1970s by Enrico Bombieri.

The considerations on the symmetries of the complex systems that led to considering the three ensembles of Hermitian matrices can also be applied to unitary matrices [23]. Making this choice of matrices, one obtains what are referred to as circular orthogonal ensemble (COE), circular unitary ensemble (CUE), and circular symplectic ensemble (CSE). Their joint eigenvalues probability density function is given explicitly by:

$$\mathcal{P}(\theta_1, \ldots, \theta_n) = \frac{1}{(2\pi)^n C_n(\beta/2)} \prod_{1 \leq i < j \leq n} |e^{i\theta_i} - e^{i\theta_j}|^\beta, \tag{50}$$

where $C_n$ is the normalization

$$C_n(\gamma) := \frac{1}{(2\pi)^n} \int_{-\pi}^{\pi} \cdots \int_{-\pi}^{\pi} \prod_{1 \leq i < j \leq n} |e^{i\theta_i} - e^{i\theta_j}|^{2\gamma} \, d\theta_1 \cdots d\theta_n, \tag{51}$$

and $\beta = 1, 2, 4$ for the COE, CUE, and CSE, respectively. This has a quite simple physical interpretation. Following [23], let us consider $n$ charged particles moving freely on a unit circle in a bidimensional world. With this, we mean that not only the motion of particles is bound on a two-dimensional plane, but also the electrostatic field generated by the charges. Therefore, if $e^{i\theta_j}$ is the position of the $j$-th particle on the circle, assuming $e$ to be the charge of all particles, the total potential energy of the system is

$$\phi(\theta_1, \ldots, \theta_n) = -e \sum_{1 \leq i < j \leq n} \ln \left| e^{i\theta_i} - e^{i\theta_j} \right|. \tag{52}$$

Considering it as a statistical ensemble at temperature $T$ and factorising the kinematical part, which is the usual one for a classical free gas, we obtain that their joint position probability density function is given by

$$\mathcal{P}(\theta_1, \ldots, \theta_n) = \frac{1}{(2\pi)^n C_n(e^2\beta/2)} e^{-\beta e \phi(\theta_1, \ldots, \theta_n)}, \tag{53}$$

where, as usual, $\beta = 1/\kappa T$, $\kappa$ being the Boltzmann constant. After making $\phi$ explicit, we obtain

$$\mathcal{P}(\theta_1, \ldots, \theta_n) = \frac{1}{(2\pi)^n C_n(e^2\beta/2)} \prod_{1 \leq i < j \leq n} |e^{i\theta_i} - e^{i\theta_j}|^{e^2\beta}, \tag{54}$$

whereas $C_n(\gamma)$ is the same as in (51).

As for (48), the random matrix calculations give (51) in terms of gamma functions for the three special values of $\beta$. The case $n = 2$ for general $\beta$ can be related to the Euler beta integral, and the case $n = 3$ gives a sum which is a special instance of an identity of Dixon for a well-poised $_3F_2$ series (cf. [19,29] for details). Using these results, Dyson made in [23] the conjecture that:

$$C_n(\gamma) = \frac{\Gamma(1 + n\gamma)}{\Gamma(1 + \gamma)^n}. \tag{55}$$

Moreover, Dyson observed that when $\gamma$ is a non-negative integer, say $k$, (51) can be rewritten as the constant term (CT) in a Laurent expansion. This allows (55) to be rewritten as

$$\mathrm{CT} \prod_{1 \le i < j \le n} \left(1 - \frac{x_i}{x_j}\right)^k \left(1 - \frac{x_j}{x_i}\right)^k = \frac{(kn)!}{(k!)^n}. \tag{56}$$

This "constant term identity" and the conjecture (55) were soon proved by J. Gunson and K. Wilson [30] and later by I.J. Good [31]. R. Askey [32] observed that the Selberg integral can be used to prove Dyson's conjecture (55) directly.

The Macdonald's conjecture [33] may be considered as a generalisation of the Dyson's conjecture (56). Let $R$ be a reduced root system, $e^\alpha$ denote the formal exponential corresponding to $\alpha \in R$, and $k$ a non-negative integer. Then, Macdonald conjectured (cf. [33], Conjecture 2.1) that the constant term in the polynomial

$$\prod_{\alpha \in R} (1 - e^\alpha)^k \tag{57}$$

should be equal to $\prod_{i=1}^{l} \binom{kd_i}{k}$, where the $d_i$ are the degrees of the fundamental invariants of the Weyl group of $R$ and $l$ the rank of $R$. Macdonald wrote this relation in an equivalent form which will turn out to be useful later. Let $G$ be a compact connected Lie group, $T$ a maximal torus of $G$, such that $R$ is the root system of $(G, T)$ and define:

$$\Delta(t) = \prod_{\alpha \in R^+} (e^{\alpha/2}(t) - e^{-\alpha/2}(t)), \tag{58}$$

where $t \in T$, the exponentials are regarded as characters of $T$, and $R^+$ is a choice of positive roots. Then, $|\Delta(t)|^2 = \prod_{\alpha \in R}(1 - e^\alpha(t))$ is a positive real-valued continuous function on $T$. This function enters in Weyl's integration formula

$$\int_G f(x)dx = \frac{1}{|W|} \int_T |\Delta(t)|^2 f(t)dt \tag{59}$$

for any continuous class function $f$ on $G$. In (59), $dx$ and $dt$ are the normalised Haar measure on $G$ and $T$, respectively, $(\int_G dx = \int_T dt = 1)$. Thus, the conjecture can be rewritten as (cf. [33] Conjecture 2.1'):

$$\int_T |\Delta(t)|^{2k} dt = \prod_{i=1}^{l} \binom{kd_i}{k}. \tag{60}$$

The equivalence of the two formulations follows from the fact that the integration over $T$ kills all but the trivial character or in other words selects the constant term in $|\Delta(t)|^{2k} = \prod_{\alpha \in R}(1 - e^\alpha(t))^k$. An observation that further generalises the conjecture is that (60) makes sense if the integer $k$ is replaced by a complex number, $s$, with positive real part, $\Re(s) > 0$. In this case, the right-hand side is replaced by

$$\prod_{i=1}^{l} \frac{\Gamma(sd_i + 1)}{\Gamma(s+1)\Gamma(sd_i - s + 1)}. \tag{61}$$

In the same paper, Macdonald generalised the conjecture further (cf. [33], Conjecture 2.3). For this, let $R$ be a root system, now not necessarily reduced, and for each $\alpha \in R$ let $k_\alpha$ be a non-negative integer such that $k_\alpha = k_\beta$ if $|\alpha| = |\beta|$. Then, the constant term in the Laurent polynomial

$$\prod_{\alpha \in R} (1 - e^\alpha)^{k_\alpha} \tag{62}$$

should be equal to the product

$$\prod_{\alpha \in R} \frac{\left(\left|\langle \rho_k, \check{\alpha}\rangle + k_\alpha + \frac{1}{2}k_{\alpha/2}\right|\right)!}{\left(\left|\langle \rho_k, \check{\alpha}\rangle + \frac{1}{2}k_{\alpha/2}\right|\right)!}, \tag{63}$$

where $\rho_k = \frac{1}{2}\sum_{\alpha \in R^+} k_\alpha \alpha$, $\check{\alpha} = \frac{2\alpha}{|\alpha|^2}$ is the coroot corresponding to $\alpha$, $k_{\alpha/2} = 0$ if $\frac{1}{2}\alpha \notin R$, and $\langle, \rangle$ is the usual scalar product induced by the Killing form. When the $k_\alpha$ are all equal, this reduces to the previous conjecture.

The Macdonald conjecture was finally proved in a slightly more general form by Opdam [34], considering $k_\alpha$ a complex valued Weyl invariant function with a positive real part. This is the content of Theorem 4.1 of [34]:

**Theorem 1** (Macdonald–Opdam). *Let R be a possibly non-reduced root system, and let $k \in \mathbb{K}$ such that[3] $\Re(k_\alpha) \geq 0$, $\forall \alpha \in R$. Then*

$$\int_T \sigma(k, t)dt = \prod_{\alpha \in R^+} \frac{\Gamma(\langle \rho(k), \check{\alpha}\rangle + k_\alpha + \frac{1}{2}k_{\alpha/2} + 1)\Gamma(\langle \rho(k), \check{\alpha}\rangle - k_\alpha - \frac{1}{2}k_{\alpha/2} + 1)}{\Gamma(\langle \rho(k), \check{\alpha}\rangle + \frac{1}{2}k_{\alpha/2} + 1)\Gamma(\langle \rho(k), \check{\alpha}\rangle - \frac{1}{2}k_{\alpha/2} + 1)}, \tag{64}$$

*where $\sigma(k, t) = \prod_{\alpha \in R^+} |t^{\frac{\alpha}{2}} - t^{-\frac{\alpha}{2}}|^{2k_\alpha}$, and T is the compact part in the "polar decomposition" of the maximal torus.*

The general proof of this theorem can be found in [34]. In the present article, we will provide a proof of this theorem only for a subclass of such integrals admitting a geometrical interpretation.

## 3. Compact Connected Lie Groups

In this section, we will review some facts about finite dimensional compact Lie groups, which will be useful for concrete applications of the notions developed in the following sections. The interested reader should consult [35,36].

### 3.1. Lie Groups and Lie Algebras
### 3.1.1. Lie Groups

For us[4], a Lie group $G$ is a finite dimensional smooth manifold endowed with the structure of group compatible with the structure of manifold. This means that:

1.  There is an associative product

$$\circ : G \times G \to G, \quad (g_1, g_2) \rightsquigarrow \circ(g_1, g_2) \equiv g_1 g_2,$$

which is a smooth map between smooth manifolds;

2.  There is a privileged point $e \in G$ such that $eg = ge = g$, $\forall g \in G$, called the unit element;

3.  There is an inverse map

$$\nu : G \to G, \quad g \rightsquigarrow \nu(g),$$

satisfying $\nu(g)g = g\nu(g) = e$, $\forall g \in G$.

As usual, we will denote $\nu(g) = g^{-1}$ and call $e$ the unit of the group. From the definitions, it follows that $e$ is unique as well as $\nu$. Moreover, the implicit function theorem implies that $\nu$ is smooth. In the infinite dimensional case, this must be assumed as a further assumption since the implicit function theorem is no more valid in general.

The product provides a free transitive left action of the group on itself called the left translation

$$L : G \times G \to G, \quad (g_1, g_2) \rightsquigarrow L_{g_1}g_2 := g_1 g_2,$$

seen as a left action of $g_1$ on $g_2$. Notice that the same map defines a right action of $g_2$ on $g_1$ called right translation $R$, so that $L_{g_1}g_2 = R_{g_2}g_1$. "Transitive" means that for any pair of elements $g_1, g_2 \in \mathbf{G}$, there exists an element $g \in \mathbf{G}$ such that $g_2 = L_g g_1$, and "free" means that such $g$ is unique. Indeed, $g = g_2 g_1^{-1}$.

3.1.2. Lie Algebras

Looking at $\mathbf{G}$ as a manifold, it is natural to consider vector fields on it, which are the sections of the tangent bundle over $\mathbf{G}$. The set of all vector fields is denoted by $\mathcal{X}(\mathbf{G}) := \Gamma(\mathbf{G}, T\mathbf{G})$. This is an infinite dimensional algebra endowed with a skew symmetric product defined by the Lie brackets

$$[\,,\,] : \mathcal{X}(\mathbf{G}) \times \mathcal{X}(\mathbf{G}) \to \mathcal{X}(\mathbf{G}), \quad (X, Y) \rightsquigarrow \mathcal{L}_Y X,$$

where $\mathcal{L}_Y X$ is the Lie derivative of $X$ along the direction of $Y$; $[\,,\,]$ is antisymmetric, $[X, Y] = -[Y, X]$ and satisfies the Jacobi identity

$$[X, [Y, Z]] + [Z, [X, Y]] + [Y, [Z, X]] = 0, \ \forall \, X, Y, Z \in \mathcal{X}(\mathbf{G}).$$

Any algebra with such a product is called a Lie algebra. However, we want to restrict our attention to the subset of those vector fields that are compatible with the left translation. Since for any $g \in \mathbf{G}$, $L_g : \mathbf{G} \to \mathbf{G}$ is a diffeomorphism, the pushforward $(L_g)_*$ is well defined on $\mathcal{X}(\mathbf{G})$. We say that $X \in \mathcal{X}(\mathbf{G})$ is left-invariant if $(L_g)_* X = X$ for any $g \in \mathbf{G}$. Fixing a point $h$, we thus must have

$$X(h) = ((L_g)_* X)(h) = (T_{g^{-1}h} L_g) X(g^{-1}h),$$

$T_p L_g$ being the tangent map (or differential) of $L_g$ in $p$. In particular, choosing $g = h$ we obtain

$$X(h) = (T_e L_g) X(e),$$

so that a left-invariant field is completely determined by its value in $e$. Thus, the set of left-invariant fields, call it $\mathcal{X}^L(\mathbf{G})$, is a finite dimensional vector space linearly isomorphic to $T_e \mathbf{G}$. Moreover, one has that the Lie bracket among left-invariant vector fields is again left-invariant so that $\mathcal{X}^L(\mathbf{G}), [\,,\,]$ is a finite dimensional Lie algebra. It is called the Lie algebra of $\mathbf{G}$; it is usually identified with $T_e \mathbf{G}$, and it is called $\mathfrak{g}$.

3.1.3. The Exponential Map

From a Lie group $\mathbf{G}$, we can move to the corresponding Lie algebra $\mathfrak{g}$. We can also do the opposite by means of the exponential map

$$\mathrm{Exp}_{\mathbf{G}} : \mathfrak{g} \to \mathbf{G}, \quad X \rightsquigarrow \gamma_X(1),$$

where $\gamma_X$ is the unique solution of the Cauchy problem

$$\dot{\gamma}(t) = X(\gamma(t)), \quad \gamma(0) = e.$$

**Theorem 2.** *Let $\mathbf{G}$ be a Lie group and $\mathfrak{g}$ be the corresponding Lie algebra. Let $X, Y, Z \in \mathfrak{g}$, and $s, t \in \mathbb{R}$. Then, the exponential map $\mathrm{Exp}_{\mathbf{G}}$ satisfies*

1.  $\mathrm{Exp}_{\mathbf{G}}(0) = e$;
2.  $\mathrm{Exp}_{\mathbf{G}}(tX) = \gamma_X(t)$;
3.  $\mathrm{Exp}_{\mathbf{G}}((t + s)X) = \mathrm{Exp}_{\mathbf{G}}(tX)\mathrm{Exp}_{\mathbf{G}}(sX)$;
4.  $\mathrm{Exp}_{\mathbf{G}}(-X) = \mathrm{Exp}_{\mathbf{G}}(X)^{-1}$;
5.  $\mathrm{Exp}_{\mathbf{G}}(X + Y) = \mathrm{Exp}_{\mathbf{G}}(X)\mathrm{Exp}_{\mathbf{G}}(Y)$ *if* $[X, Y] = 0$;
6.  $\mathrm{Exp}_{\mathbf{G}}$ *defines a local diffeomorphism between an open neighbourhood of $0$ in $\mathfrak{g}$ and an open neighbourhood of $e$ in $\mathbf{G}$;*

7. $T_e\mathrm{Exp}_{\boldsymbol{G}} : T_e\boldsymbol{G} \to \mathfrak{g}$ *is an isomorphism of vector spaces;*
8. $\mathrm{ev}_e \circ T_e\mathrm{Exp}_{\boldsymbol{G}} : T_e\boldsymbol{G} \to T_e\boldsymbol{G}$ *is the identity map over* $T_e\boldsymbol{G}$.

In the last point, $\mathrm{ev}_e : \mathfrak{g} \to T_e\boldsymbol{G}$ is the map evaluating a left-invariant vector field in $e$. The exponential map is not surjective in general, but it is so for connected compact groups. Obviously, it depends on $\boldsymbol{G}$ in the sense that we have to know $\boldsymbol{G}$ in order to compute $\mathrm{Exp}_{\boldsymbol{G}}$. Nevertheless, we will see that it behaves well with regard to representations, so that in a sense it can be used to "realise" the group starting from the algebra.

*3.2. Semisimple Lie Groups and Algebras*

A special class of Lie algebras is given by semi-simple algebras.

**Definition 1.** *A semi-simple Lie algebra* $\Lambda$ *is a Lie algebra which has a dimension at least two and that does not contain proper abelian ideals.*

Recall that an ideal $I$ of $\Lambda$ is a vector subspace of $\Lambda$ such that $[a, j] \in I$ for all $a \in \Lambda$ and $j \in I$. "Proper" means that $0 \subsetneqq I \subsetneqq \Lambda$, and "abelian" means that $[i, j] = 0$ for any $i, j \in I$.

**Definition 2.** *A simple Lie algebra* $\Lambda$ *is a Lie algebra which has a dimension at least two and that does not contain proper ideals.*

A simple algebra is also semi-simple, whereas any semi-simple algebra is the direct sum of simple algebras in a unique way up to reordering of the simple factors. Similar concepts can be defined for Lie groups:

**Definition 3.** *A semi-simple Lie group* $\boldsymbol{G}$ *is a non-abelian Lie group which does not contain proper abelian normal subgroups.*

Recall that a normal subgroup $\boldsymbol{H}$ of a Lie group $\boldsymbol{G}$ is a Lie subgroup of $\boldsymbol{G}$ such that $ghg^{-1} \in I$ for all $g \in \boldsymbol{G}$ and $h \in \boldsymbol{H}$. Proper means that $e \subsetneqq \boldsymbol{H} \subsetneqq \boldsymbol{G}$, and abelian means that $[h, k] \equiv hkh^{-1}k^{-1} = e$ for any $h, k \in \boldsymbol{H}$.

**Definition 4.** *A simple Lie group* $\boldsymbol{G}$ *is a non-abelian Lie Group which does not contain proper normal subgroups.*

A simple group is also semi-simple, whereas any semi-simple group is the direct product of simple groups in a unique way up to reordering of the simple factors and the quotient of a finite normal subgroup.

Notice that a group $\boldsymbol{G}$ is (semi-)simple if and only if the corresponding Lie algebra $\mathfrak{g}$ is. It is also important to mention that by Lie subgroup of $\boldsymbol{G}$, we mean a subgroup that is a Lie group and is also a submanifold of $\boldsymbol{G}$.

There is a simple way to characterise semi-simple groups, as we will see in Section 3.7.

*3.3. Abelian Compact Lie Groups*

It can be shown that $n$-dimensional abelian connected compact Lie groups must be of the form $\boldsymbol{G} = \mathbb{R}^n / L$, where $L$ is an $n$-dimensional lattice that is an abelian additive subgroup of $\mathbb{R}^n$ isomorphic to $\mathbb{Z}^n$ and discrete. So, it is topologically equivalent (and diffeomorphic) to an $n$-dimensional torus

$$\boldsymbol{G} \simeq T^n \simeq (S^1)^{\times n}.$$

*3.4. All Compact Lie Groups*

We are now ready to consider an arbitrary compact Lie group.

**Theorem 3.** *Let $G$ be a connected compact Lie group. Then,*

$$G \simeq G_0 \times T^s / \Delta,$$

*with $G_0$ a semi-simple subgroup, $T^s$ an abelian torus, and $\Delta$ a finite subgroup.*

**Proof.** (See [1,36]). Let us consider the corresponding derived group $G'$ of $G$, also called the commutator group, which is the group generated by all the commutators in $G$ that are the elements of the form $[g,h] = ghg^{-1}h^{-1}$ (not to be confused with the Lie brackets). It is easy to check that, by construction, $G'$ is a semi-simple group and that it is a normal Lie sub-group of $G$. Let us put $G_0 := G'$. Next, consider the center $Z$ of $G$, that is the subgroup of those elements commuting with the whole $G$. Put $T^s := Z^0$, the connected component of $Z$ containing $e$. Then, consider the multiplication map

$$m : G' \times Z^0 \to G, \quad (g,z) \rightsquigarrow gz.$$

We claim that it is surjective. Indeed, first notice that since $Z^0$ is central, we have that

$$\begin{aligned} m((g,z)(g',z')) = m((gg',zz')) &= (gg')(zz') \\ &= (gz)(g'z') = m((g,z))m((g',z')), \end{aligned}$$

so $m$ is an homomorphism. Moreover, by construction, $G/G'$ is abelian (it is called the abelianisation of $G$), and, $G'$ being semi-simple, passing to the corresponding Lie algebras we see that necessarily, with obvious notation,

$$\mathfrak{g}' \oplus \mathfrak{z}^0 = \mathfrak{g},$$

which implies that the differential of $m$ is surjective. This implies that the image of $m$ is open. On the other hand, $G'$ and $Z^0$ are compact, and $m$ is continuous so that the image of $m$ is compact and then closed. However, $G$ is connected so that $m$ is surjective.
Now, the kernel of $m$ is defined by the set of elements $(g,z) \in G' \times Z^0$ such that $gz = e$, that are the elements of the form $(g, g^{-1})$. This means that $g \in G' \cap Z^0$ and so the kernel of $m$ is identified by the embedding of $\Delta := G' \cap Z^0$ in $G' \times Z^0$ via the map $g \mapsto (g, g^{-1})$. This way, we have an exact sequence

$$e \to G' \cap Z^0 \to G' \times Z^0 \to G \to e$$

so that

$$G \simeq G' \times Z^0 / \Delta.$$

On the other hand, $G' \cap Z^0$ is the center of $G'$. As we will see, the center of a semi-simple Lie group is always a finite group, and we have

$$G \simeq G_0 \times T^s / \Delta,$$

as claimed.  □

*3.5. Cohomology of Compact Lie Groups*

Any compact Lie group $\mathbb{G}$ contains a maximal torus $\mathbb{T}^r$ which corresponds to a maximal abelian subgroup. It is unique up to isomorphisms. Its dimension $r$ is called the rank of the group. Since $G$ is a manifold, one can define the exterior bundle $\Lambda^* G$, whose sections are the differential forms. In particular, $\Lambda^* G = \mathbb{R} \oplus \sum_{k=1}^n \Lambda^k G$, where $n$ is the dimension of $G$ and $\Lambda^k G$ is the $k$-th external power of the cotangent bundle $T^* G$. If $d$ is the external

derivative and $d^k$ is its restriction to $\Lambda^k \boldsymbol{G}$, so that $d^0$ is the usual differential on smooth functions, then, to $\Lambda^* \boldsymbol{G}$ we have associated an elliptic complex

$$0 \hookrightarrow C^\infty(\boldsymbol{G}) \xrightarrow{d^0} T^* \boldsymbol{G} \xrightarrow{d^1} \Lambda^2 \boldsymbol{G} \xrightarrow{d^2} \cdots \xrightarrow{d^{k-1}} \Lambda^k \boldsymbol{G} \xrightarrow{d^k} \cdots \xrightarrow{d^{n-1}} \Lambda^n \boldsymbol{G} \xrightarrow{d^n} 0,$$

since $\operatorname{Im} d^{k-1} \subseteq \operatorname{Ker} d^k$. As usual, $\operatorname{Ker} d$ are the closed forms, $\operatorname{Im} d$ are the exact forms. The ring

$$H(\boldsymbol{G}, \mathbb{R}) := \frac{\operatorname{Ker} d}{\operatorname{Im} d} = \oplus_{k=0}^n H^k(\boldsymbol{G}, \mathbb{R}),$$

is the cohomology ring, and

$$H^k(\boldsymbol{G}, \mathbb{R}) = \frac{\operatorname{Ker} d^k}{\operatorname{Im} d^{k-1}}$$

is the cohomology (additive) group of order $k$. The cohomology of a compact connected Lie group is characterised by the following theorem due to Hopf (see [36] for a modern proof):

**Theorem 4.** *The cohomology of a connected compact Lie group $\boldsymbol{G}$ of rank $r$, over a field of characteristic $0$, is the same as the cohomology of a product of $r$ odd dimensional spheres.*

Indeed, we can say a little bit more; we know that a connected compact Lie group has the form

$$\boldsymbol{G} = \boldsymbol{G}_1 \times \cdots \times \boldsymbol{G}_p \times \mathbb{T}^q / \Delta,$$

where $\boldsymbol{G}_i$, $1 \leq i \leq p$, are simple, $\mathbb{T}^q$ is a torus, and $\Delta$ a finite group. The cohomology group is insensitive to the action of $\Delta$. By means of the Künneth formula, the cohomology of a product of spaces can be recovered by that of the factors via cup products so that we can understand the whole cohomology from the one of the torus and the simple factors. $\boldsymbol{G}_i$ has rank $r_i$, such that $r = q + \sum_{i=1}^p r_i$. The torus contributes to the cohomology, as usual, as the product of $q$ one-dimensional spheres. Therefore, the rest of the cohomology is determined by that of $\boldsymbol{G}_i$. By the Hopf theorem, we know that $H(\boldsymbol{G}_i, \mathbb{R})$ is that of a product of $r_i$ odd dimensional spheres $S^{2d_j^{(i)}-1}$, $j = 1, \ldots, r_i$. The numbers $d_j^{(i)}$ are completely classified for simple Lie groups and are called the fundamental invariant degrees. See below for their complete list and clarification of the term.

*3.6. Fundamental Group of Compact Lie Groups*

Let us briefly discuss the fundamental group of a compact connected Lie group $\boldsymbol{G}$. We know that

$$\boldsymbol{G} = \boldsymbol{G}_1 \times \cdots \times \boldsymbol{G}_p \times \boldsymbol{T}^s / \Delta.$$

where the $\boldsymbol{G}_i$'s are simple, and $\Delta = (\boldsymbol{G}_1 \times \cdots \times \boldsymbol{G}_p) \cap \boldsymbol{T}^s$ is the intersection between the torus and the simple components. In particular, $\Delta$ is in the center of the product of the simple components and is thus a finite group. Let us rewrite $\boldsymbol{G} = \boldsymbol{H}/\Delta$. Then, obviously,

$$\pi_1(\boldsymbol{H}) = \pi_1(\boldsymbol{G}_1) \times \cdots \times \pi_1(\boldsymbol{G}_p) \times \mathbb{Z}^s.$$

Now, since $\Delta$ acts freely on $\boldsymbol{H}$, we have that the map $\phi : \boldsymbol{H} \to \boldsymbol{G}$ is a covering map so that $\pi_1(\boldsymbol{G})/\pi_1(\boldsymbol{H}) = \Delta$. Therefore

$$\pi_1(\boldsymbol{G}) \simeq \pi_1(\boldsymbol{G}_1) \times \cdots \times \pi_1(\boldsymbol{G}_p) \times \mathbb{Z}^s \times \Delta.$$

*3.7. Representations*

Since we are working with finite dimensional Lie groups, we will speak about finite dimensional representations. Roughly speaking, a representation is a way of interpreting a group as a group of transformations. As such, it does not only depend from the group but also from the choice of the object to be transformed. A priori, there is no any reason to fix a particular class of objects to be transformed. However, the simplest objects to be considered are vector spaces. On a fixed vector space $V$, the group is expected to act as a subgroup of the general linear group $GL(V)$ (that is the set $Aut(V)$ of automorphisms of $V$, seen as a group with the composition law). In this case, we will speak about a linear representation. If the vector space is real or complex and endowed with a scalar product, or a symplectic structure, we can think to embed the group in the automorphisms of the given structure so that we will deal with orthogonal, unitary, or symplectic linear representations, and so on. If in place of vector spaces, we work with projective spaces, the representation is not linear but projective, etc.

We will limit ourselves to the simplest case of linear representations.

**Definition 5.** *A finite dimensional linear representation of a group $G$ is a group homomorphism*

$$R : G \to GL(V),$$

*where $V$ is a finite dimensional vector space called the support of the representation. In particular, $gh \rightsquigarrow R(g)R(h)$, and $R(e) = I_V$, the identity map $I_V(v) = v$ for any $v \in V$.*

Since Lie groups are strictly related to Lie algebras, it is interesting talso consider linear representations of Lie algebras. To this end, let us first note that the set $End(V)$ of all endomorphisms of $V$ becomes a Lie algebra if endowed with the Lie product $[ \, , \, ]_0$ given by the commutator: $[A, B]_0 := AB - BA$. This algebra is called $L(V)$, the linear algebra.

**Definition 6.** *A finite dimensional representation of a Lie algebra $\Lambda$ is a homomorphism of algebras*

$$\rho : \Lambda \to L(V),$$

*where $V$ is a finite dimensional vector space, said the support of the representation. In particular, $\rho : [a, b] \rightsquigarrow [\rho(a), \rho(b)]_0$ for any $a, b \in \Lambda$.*

Given the relation between a Lie group and the corresponding Lie algebra, it is natural to wonder if there is some relation between the respective representations. The answer is given by the following well-known propositions:

**Proposition 1.** *Let $V$ be a finite dimensional vector space. Then, $GL(V)$ is a Lie group of dimension $(\dim V)^2$, and the corresponding Lie algebra is $L(V)$.*

**Proposition 2.** *Let $(R, V)$ be a linear representation of a Lie group $G$:*

$$R : G \to GL(V).$$

*Then,*

$$T_e R : T_e G \simeq \mathfrak{g} \to T_{I_V} GL(V) \simeq L(V)$$

*is a linear representation of $\mathfrak{g}$. We will call it $\rho_R$.*

**Proposition 3.** *Let $R, V$ be a linear representation of the Lie group $G$ and $\rho_R$ the corresponding representation of the associated Lie algebra. Then, the diagram*

$$G \xrightarrow{\ R\ } GL(V)$$

$$\mathrm{Exp}_G \uparrow \qquad\qquad \uparrow \mathrm{Exp}_{\mathrm{GL}(V)}$$

$$\mathfrak{g} \xrightarrow{\ \rho_R\ } L(V)$$

*commutes.*

These propositions relate representations of groups to representations of algebras. We are interested in particular linear representations of groups. We want representations such that the image of $G$ in $Aut(V)$ is a faithful realisation of $G$, and, in some sense, we would like for it to be the smallest realisation.

**Definition 7.** *The representation $R, V$ of a group $G$ is said faithful if $R$ is injective.*

This means essentially that we can identify $G$ with its image in $GL(V)$. For the second notion we need, we recall that $V' \subset V$ is said to be invariant under $G$ if $R(g)v \in V'$ for any $g \in G, v \in V'$.

**Definition 8.** *The representation $R, V$ of a group $G$ is said irreducible if $V$ does not contain proper invariant subspaces.*

If $V$ contains a proper irreducible subspace $V'$, then $R|_{V'}, V$ defines a new representation that is "smaller" than $R, V$. In this sense, an irreducible representation is the smallest (non-trivial) one.

3.7.1. The Adjoint Representations

Given a group, its linear representations are not natural since in order to define them one has to introduce an extra structure, the one of linear space, which is not at all present in the definition of the group. For an algebra, particular representations could arise from the naturally underlying linear structure in the algebra itself. For a Lie group $G$, these things go together since it has associated a natural vector space, $T_e G$, isomorphic to $\mathfrak{g}$. In order to construct this natural representation, we start with the conjugation map associated with a given point $g \in G$:

$$c_g : G \to G, \ g \rightsquigarrow ghg^{-1}.$$

This map is a homomorphism. To it, we can associate the map

$$Ad_g := T_e c_g : T_e G \to T_e G,$$

called the Adjoint map associated with $g$. It is clear that $Ad_g$ is an automorphism of $\mathfrak{g}$, such that $Ad_e = I_{\mathfrak{g}}$ and $Ad_{g_1 g_2} = Ad_{g_1} \circ Ad_{g_2}$. Thus, the map

$$Ad : G \to GL(\mathfrak{g}), \ g \rightsquigarrow Ad_g$$

defines a representation of $G$ with support $\mathfrak{g}$, called the Adjoint representation. As above, it also defines a representation of $\mathfrak{g}$ over itself, called the adjoint representation

$$ad : \mathfrak{g} \to L(\mathfrak{g}), \ a \rightsquigarrow ad_a,$$

where

$$ad_a : \mathfrak{g} \to \mathfrak{g}, \ b \rightsquigarrow [a, b].$$

### 3.7.2. Simple Algebras and the Cartan Criterion

The fact that the adjoint representation is natural does not guarantee that it is faithful or irreducible, so, despite its naturalness, it may not be useful to characterise the group (or the algebra). The kernel of *ad* is

$$\text{Ker } ad = \{a \in \mathfrak{g} : ad_a(b) = 0 \; \forall b \in \mathfrak{g}\},$$

so that it is an abelian ideal. A sufficient (but not necessary) condition for it to be the set $\{0\}$ is that $\mathfrak{g}$ is semi-simple. In this case, the adjoint representation is thus faithful. This makes semi-simple algebras interesting since they can be recovered by looking at their natural representation. Suppose it is reducible. Then, we immediately obtain that the proper invariant subspace $\mathfrak{g}'$ is in fact a proper ideal so that $\mathfrak{g}$ is not simple but only semi-simple. This is what makes simple algebras so special; they can be reconstructed from their own natural representation in an irreducible way. Things are a little bit more subtle for the groups, as we will se below for the case of compact forms.

A first byproduct of the adjoin representation is the following characterisation of semi-simple Lie algebras (and groups): on the Lie algebra one can define a symmetric bilinear form

$$K : \mathfrak{g} \times \mathfrak{g} \to \mathbb{K}, \quad (a,b) \rightsquigarrow \text{Tr}(ad_a \circ ad_b),$$

where $\mathbb{K} = \mathbb{R}$ or $\mathbb{C}$ if the algebra is real or complex, respectively. The form $K$ is called the Killing form of $\mathfrak{g}$.

**Theorem 5.** *A Lie algebra $\mathfrak{g}$ is semi-simple if and only if its Killing form $K$ is non-degenerate.*

Proving that the non-degeneracy is a sufficient condition is a quite simple exercise. The converse requires the introduction of more sophisticated structures, which are not relevant for our purposes.

### 3.8. Roots and Classifications

In this section, we will concentrate on simple Lie algebras, briefly recalling the ingredients leading to their classifications. We are interested in real groups and algebras, but the simplest starting point is with complex algebras.

### 3.8.1. Classification of Complex Simple Lie Algebras

A simple Lie group $G$ of rank $r$ contains a maximal torus $\mathbb{T}^r$, whose Lie algebra is a maximal abelian subalgebra of $\mathfrak{g}$, usually called the Cartan subalgebra and denoted with $H$. The elements of $H$ have vanishing Lie product, which means that the corresponding operators, defined by the adjoint representation, commute. Moreover, as linear operators over $\mathfrak{g}$, they are diagonalisable. Since they commute, they are all simultaneously diagonalisable, thus having common eigenvectors. This means that a basis of vectors $g_i \in \mathfrak{g}_i$ exists such that

$$ad_h g_i = \lambda_i(h) g_i$$

for any $h \in H$. We have explicitly shown the dependence of the eigenvalue from $h$. Since $ad_h g_i = [h, g_i]$, $\lambda_j$ depends linearly on $H$ so that any $j$, $\lambda_j$ defines a linear functional over $H$ that is an element of the dual space $H^*$. These linear eigenfunctionals are called the roots of the algebra. Notice that these are not necessarily all distinct, and the corresponding eigenspaces can have dimensions larger than one. For example, the elements of $H$ are obviously eigenvectors corresponding to the zero eigenvalue, and one then understands that the Cartan subalgebra is indeed the eigenspace corresponding to the vanishing root. The vanishing root is thus degenerate with a degeneration index $r$. However, by using that

the Killing form, it must be non-degenerate, and the algebra has a finite dimension. One can show quite easily that ([35])

- For any non-vanishing root $\lambda \in H^*$, also $k\lambda$ is a root if and only if $k = 0, \pm 1$;
- The number of non-vanishing roots is at least $2r$;
- If $\lambda$ is a non-vanishing root, then its corresponding eigenspace has dimension one;
- If $\alpha$ and $\beta$ are two non-vanishing roots and $a$, $b$ are in the corresponding eigenspaces, then either $[a, b] = 0$ and $\alpha + \beta$ is not a root or $\alpha + \beta$ is a root and $[a, b] \neq 0$ belongs to the corresponding eigenspace;
- The eigenspaces $\mathfrak{g}_\alpha$ and $\mathfrak{g}_\beta$ of the roots $\alpha$ and $\beta$ are mutually orthogonal with regard to the Killing form unless $\alpha + \beta = 0$.

In particular, it follows that the Killing form restricted to the Cartan subalgebra is non-degenerate, and therefore, it defines a natural isomorphism between $H$ and its dual $H^*$. Using this and the (semi) simplicity of the algebra, one can further show that the set of roots generates $H^*$ over $\mathbb{C}$ and that, in particular, they are in rational dependence in the sense that after fixing any given choice of a basis of roots in $H^*$, all the remaining roots are linear combinations of the elements of the basis with coefficients in $\mathbb{Q}$. In particular, this means that the set of roots spans an $r$-dimensional real subspace of $H^*$, which is called the real form $H^*_{\mathbb{R}}$. Let $K_H$ be the restriction to $H$ of the Killing form. Let

$$\phi : H \to H^*, \quad \lambda \rightsquigarrow K_H(\lambda. \cdot)$$

be the corresponding isomorphism. Then, we can induce a bilinear form over $H^*$

$$( \mid ) : H^* \times H^* \to \mathbb{C}, \quad (\alpha, \beta) \rightsquigarrow K_H(\phi^{-1}(\alpha), \phi^{-1}(\beta)),$$

which, by construction, is symmetric and non-degenerate. However, one can say more; when restricted to the real form, $( \mid )$ is also positive definite and thus defines a Euclidean structure over $H^*_{\mathbb{R}}$. From now on, we will always refer to this restriction.

**Theorem 6** (Cartan). *Let $\alpha$ and $\beta$ be two non-vanishing roots of a semi-simple Lie algebra. Then,*

$$p_{\alpha\beta} = 2\frac{(\alpha|\beta)}{(\alpha|\alpha)} \in \mathbb{Z},$$

*and*

$$But w_\alpha(\beta) = \beta - 2\frac{(\alpha|\beta)}{(\alpha|\alpha)}\alpha$$

*is also a root.*

Obviously, the same holds true interchanging $\alpha$ and $\beta$. The linear map

$$w_\alpha : H^*_{\mathbb{R}} \to H^*_{\mathbb{R}}, \quad \beta \rightsquigarrow \beta - 2\frac{(\alpha|\beta)}{(\alpha|\alpha)}\alpha$$

is called the Weyl reflection. Geometrically, it is a reflection through the hyperplane orthogonal to $\alpha$. The set of $w_\alpha$, when $\alpha$ varies among all roots, by composition generates a discrete subgroup of the isometry group of $H^*_{\mathbb{R}}$, called the Weyl group of $\mathfrak{g}$.

Cartan's theorem is the key for the classification of all (complex) simple Lie algebras. Indeed, it is possible to prove the existence of a (non-unique) basis of roots for $H^*$, called a simple root system, such that any other root is a combination of the basis with only non-negative or non-positive integer coefficients. The elements of such a basis are called simple roots and allow for separating the roots in positive and negative roots in an obvious

way. From this, it follows that the scalar product on two simple roots must be non-positive. Note that

$$p_{\alpha\beta}p_{\beta\alpha} = 4\cos^2\theta_{\alpha\beta},$$

where $\theta_{\alpha\beta}$ is the angle between the roots $\alpha$ and $\beta$ defined by the Euclidean scalar product $(\ |\ )$. From Cartan's theorem, it then follows that the possible values of the scalar products among roots are such that $4\cos^2\theta_{\alpha\beta} = 0, 1, 2, 3, 4$. In the case it is zero, there are no relations among the lengths, and in the case it is four, then the two roots are collinear, a case already considered, thus not interesting. In the other cases, one also notices that

$$\frac{p_{\alpha\beta}}{p_{\beta\alpha}} = \frac{\|\beta\|^2}{\|\alpha\|^2}.$$

Therefore, the angles and rates of length among simple roots are constrained, and such constraints can be used to completely determine all possible root systems up to equivalences. When a root system is given, the action of the Weyl group generates all the remaining roots associated with a semi-simple algebra, and this allows us to reconstruct the whole algebra. This is the way one classifies simple Lie algebras. Indeed, $p_{\alpha\beta}$ associated with a given simple root system are the components of a matrix called the Cartan matrix. It has the following properties:

- The diagonal elements are al ltwo, whereas the non-diagonal elements are non-positive integers;
- $p_{\alpha\beta}$ is zero if and only if $p_{\beta\alpha}$ is zero;
- The root can be ordered so that the non-vanishing elements below the diagonal are $-1$.

Moreover, if the algebra is semi-simple, all roots relative to a simple block are orthogonal to the ones of any other block so that the corresponding Cartan matrix is a block diagonal. Thus, to classify simple algebras, one has to classify all the Cartan matrices that cannot be block-diagonalised via reordering of the roots. This can be performed using the method of Dynkin diagrams, where to each simple root one associates a dot, and two dots $\alpha$ and $\beta$ are connected by a number $n_{\alpha\beta} = p_{\alpha\beta}p_{\beta\alpha}$ of lines oriented from the longer to the shorter root. These diagrams are in biunivocal correspondence with the Cartan matrices, and in particular, a diagram corresponds to a simple algebra if and only if it is connected.

All possible Dynkin diagrams are depicted in Appendix A. They correspond to the following classical complex matrix algebras (with commutator as Lie bracket):

- $sl(n, \mathbb{C})$ of rank $n - 1$, $n \geq 2$ of traceless $n \times n$ complex matrices;
- $so(2n + 1)$ of rank $n$, $n \geq 2$ of $(2n + 1) \times (2n + 1)$ antisymmetric complex matrices;
- $sp(2n)$ of rank $n$, $n \geq 2$ of $(2n) \times (2n)$ symplectic complex matrices;
- $so(2n)$ of rank $n$, $n \geq 3$ of $(2n) \times (2n)$ antisymmetric complex matrices;

and the five exceptional complex algebras

- $\mathfrak{g}_2$, the linear algebra generated by the derivations acting on the octonionic algebra $\mathbb{O}$;
- $\mathfrak{f}_4$, the Lie algebra associated with the isometry group of the octonionic projective plane $\mathbb{O}\mathbb{P}^2$;
- $\mathfrak{e}_6$, the Lie algebra associated with the isometry group of the complex octonionic projective plane $(\mathbb{C} \otimes \mathbb{O})\mathbb{P}^2$;
- $\mathfrak{e}_7$, the Lie algebra associated with the isometry group of the quaternionic octonionic projective plane $(\mathbb{H} \otimes \mathbb{O})\mathbb{P}^2$;
- $\mathfrak{e}_8$, the Lie algebra associated with the isometry group of the bi-octonionic projective plane $(\mathbb{O} \otimes \mathbb{O})\mathbb{P}^2$.

We refer to [37] for recalling the definitions of octonions and octonionic planes.

3.8.2. Classification of Real Simple Lie Algebras

A little bit more involuted is the classification of real Lie algebras since for any complex Lie algebra there are more possible real forms, inequivalent real algebras whose tensorization by $\mathbb{C}$ gives the considered complex simple algebra. For example, $\boldsymbol{sl}(2,\mathbb{C})$ can be obtained by complexification of $\boldsymbol{sl}(2,\mathbb{R})$ or of $\boldsymbol{su}(2)$, the Lie algebra of anti-Hermitian matrices. We are not interested in report here how exactly all real forms associated with a complex form can be obtained since we are mainly interested in those real forms which exponentiate to a compact group, the so-called compact form. Nevertheless, since it will be of use in the next section, we illustrate here the main steps leading to the classification of real forms.

We will show the existence of two important real forms, the second one being the compact form. Let $\alpha_1, \ldots, \alpha_n$ be the positive roots of $\mathfrak{g}$, where $r + 2n$ is the dimension of the algebra. We know that they generate $H^*_{\mathbb{R}}$ on $\mathbb{R}$, so we can choose a real basis $h^*_i$, $i = 1, \ldots, r$ of $H^*_{\mathbb{R}}$ such that any root has a real coefficient with regard to it. It is easy to see that with regard to this choice, the Killing form is a positive definite over $H$. Indeed, given $h \in H$, we fix a basis $h_1, \ldots, h_r$ in $H$ and let $g_s$, $s = 1, \ldots, n$ the eigenvectors for the positive roots, and $g_{-s}$ the ones for the negative roots. With respect to this basis, we then find that

$$K(h,h) = \sum_s (\alpha_s(h)^2 + (-\alpha_s(h))^2) > 0. \tag{65}$$

In particular, we can always fix an orthonormal basis $h_i$ of $H$. Moreover, we can normalise $g_{-s}$ such that $K(g_s, g_{-s'}) = \delta_{ss'}$, which, together with $K(h_i, h_j) = \delta_{ij}$, are the only non-vanishing products. In particular, one can always choose $g_s$ (and $g_{-s'}$) so that $[g_s, g_t]$ is a real combination in the algebra. From this, it follows that the chosen generators define a real subalgebra of $\mathfrak{g}$. An interesting basis of this real subalgebra is given by

$$h_i, \ i = 1, \ldots, r, \qquad \lambda_s^{\pm} = \frac{g_s \pm g_{-s}}{\sqrt{2}}, \ s = 1, \ldots, n.$$

This basis is orthogonal, with

$$K(h_i, h_i) = K(\lambda_s^+, \lambda_s^+) = 1, \quad K(\lambda_s^-, \lambda_s^-) = -1.$$

This real Lie algebra is not compact ([36]).

**Proposition 4.** *Let $\boldsymbol{G}$ be a finite dimensional connected simple Lie group. Then, $\boldsymbol{G}$ is compact if and only if the Killing form on the corresponding Lie algebra is negative definite.*

We can obtain the compact form from the above real algebra by noticing that, if $\mathfrak{k}$ is the subalgebra generated by the $\{\lambda_s^-\}$ and $\mathfrak{p}$ the subspace generated by the remaining elements, then

$$[\mathfrak{k}, \mathfrak{k}] \subseteq \mathfrak{k},$$
$$[\mathfrak{k}, \mathfrak{p}] \subseteq \mathfrak{p},$$
$$[\mathfrak{p}, \mathfrak{p}] \subseteq \mathfrak{k}.$$

This can be rephrased by saying that $\mathfrak{k}$ is symmetrically embedded in $\mathfrak{g}$. It is a maximal compact subalgebra of $\mathfrak{g}$, and indeed, it can be shown that in the specific case of the real algebra we have just constructed, $\mathfrak{k}$ is the smallest possible maximal subalgebra in this sense. There exist smaller maximal subalgebras (not properly contained in proper subalgebras of $\mathfrak{g}$), but they are not symmetrically embedded. Using the property of maximal embedding, we see that the generators

$$\tilde{h}_i = i h_i, \ i = 1, \ldots, r, \qquad \tilde{\lambda}_s^+ = i \lambda_s^+, \qquad \tilde{\lambda}_s^- = \lambda_s^-, \ s = 1, \ldots, n, \tag{66}$$

also define a real Lie subalgebra of $\mathfrak{g}$, but now

$$K(\tilde{h}_i, \tilde{h}_i) = K(\tilde{\lambda}_s^+, \tilde{\lambda}_s^+) = K(\tilde{\lambda}_s^-, \tilde{\lambda}_s^-) = -1,$$

so that we obtained the compact real form.

This way, we have constructed two real forms: the compact form and a non-compact form with the smallest possible symmetrically embedded maximal compact subgroup. These are related by a simple trick. We considered the map $\sigma : \mathfrak{g} \to \mathfrak{g}$, such that $\mathfrak{k} \rightsquigarrow \mathfrak{k}$, $\mathfrak{p} \rightsquigarrow i\mathfrak{p}$, that can be easily extended to a linear map. To it, we can associate the linear map $\theta = \sigma^2$. It has the property that $\theta^2$ is the identity and $K(\theta(a), \theta(b)) = K(a, b)$. Moreover, $K_\theta(a, b) = K(a, \theta(b))$ is negative definite over $\mathfrak{k} \oplus \mathfrak{p}$, and $\mathfrak{k}$ and $\mathfrak{p}$ are eigenspaces for $\theta$ with eigenvalues $+1$ and $-1$, respectively. It turns out that any real form is associated with such a map: any linear involution $\theta : \mathfrak{g} \to \mathfrak{g}$, $\theta^2 = 1$, which is a $K$-isometry, is called a Cartan form. It determines a real form as a real space $\mathfrak{g}_\theta = \mathfrak{k} + \mathfrak{p}$ on which $K_\theta$ defined as above is negative definite, and the decomposition is $K$-orthogonal. It is immediate to verify that the positive eigenspace $\mathfrak{k}$ is a maximal compact subalgebra symmetrically embedded. The reader interested in the details should read for example the book [38].

### *3.9. Root Systems*

One interesting fact we have recalled in the previous sections is that simple algebras are associated with roots, which in general are generated by integer combinations of a fundamental system of simple roots. Therefore, the roots are in general a subset of the lattice linearly generated by the $\mathbb{Z}$-span of the simple roots system. A one-dimensional eigenspace is associated with each non-vanishing root. We will say that such roots have weight one. The vanishing root has weight equal to the linear dimension of the lattice. Recall that if $\alpha$ is a root, then $k\alpha$ is also a root if and only if $k = 0, \pm 1$. Combined with Cartan's theorem, this shows that the roots associated with any simple algebra define a crystallographic reduced root system. We will see in the next section that more general root systems, crystallographic but non-reduced, can be associated with the real forms (or, equivalently, to the associated symmetric spaces).

### *3.10. Compact Forms*

We have seen how complex finite dimensional simple Lie algebras are completely classified by the Dynkin diagrams. Any such diagram completely identifies a simple Lie algebra over $\mathbb{C}$. The same is not true for the real forms, but it becomes true if we restrict our attention to compact real forms. Nevertheless, the unicity is again lost if we pass to the group; to a given Dynkin diagram, there can correspond a number of connected simple compact Lie groups all having the same (simple compact real) Lie algebra. We will call these the compact forms of the group.

The compact forms can all be determined as follows.

**Proposition 5.** *Let* $\mathfrak{g}$ *be the simple compact form algebra associated with a given Dynkin diagram. Then, there exists a unique compact connected and <u>simply connected</u> Lie group* $\tilde{G}$ *whose associated Lie algebra is* $\mathfrak{g}$.

See [35]. Any other compact group $G$ associated with $\mathfrak{g}$ can be obtained from $\tilde{G}$. Recall that $\tilde{G}$ has finite center $Z$ (cf. [35], Proposition 23.11, p. 200). Let $\Gamma_k$, $k = 1, \ldots, m$ be all possible subgroups of $Z$, which of course are a finite number. In particular, we assume $\Gamma_1 = Id$, $\Gamma_m = Z$. Then,

$$G_k = \tilde{G}/\Gamma_k, \qquad k = 1, 2, \ldots, m, \tag{67}$$

are all possible compact forms associated with the given Dynkin diagram. Notice that

$$\phi_k : \tilde{G} \to G_k$$

is a covering, and since $\Gamma_k$ acts freely, we have $\pi_1(G_k) = \Gamma_k$.

### 3.11. Realizations

Each compact group admits at least a faithful linear representation (cf. [35], Theorem 4.2, p. 26). Let us call $(R_k, V_k)$ such a representation for $G_k$ (in particular $(Ad, \mathfrak{g})$ is faithful for $G_m$). It induces a faithful representation $(\rho_i, V_i)$ of $\mathfrak{g}$, so that the following diagram is commutative:

$$
\begin{array}{ccc}
G_k & \xrightarrow{\ R_k\ } & Aut(V_k) \\
\exp_{G_k}\Big\uparrow & & \Big\uparrow \text{Exp} \\
\mathfrak{g} & \xrightarrow{\ \rho_k\ } & End(V_k)
\end{array}
$$

$G_k$ being compact, it follows that $R_k$ is injective and continuous, and $Aut(V_k)$ is $T_2$. Hence, $G_k$ and $R_k(G_k)$ are homeomorphic ([39], Theorem 8.8) and have the same fundamental group[5]. Therefore, we can construct a realisation of the desired compact form just by exponentiating the matrices associated with the Lie algebra $\mathfrak{g}$ via the suitable representation $\rho_k$ induced by the faithful representation $R_k$ of $G_k$. We will call $\rho_k$ a $G_k$-*faithful representation*. Thus, we realise the desired compact $G$ form by working with the suitable $G$-faithful representation of the corresponding Lie algebra.

Now, let $H$ be a symmetrically embedded subgroup of $G$. If $(\rho, V)$ is a $G$-faithful representation of $\mathfrak{g}$, then $\rho$ will decompose into a direct sum of representations of the Lie algebra $\mathfrak{h}$ of $H$ among which at least one is surely $H$-faithful.

## 4. Compact Symmetric Spaces

In this section, we provide a short survey on compact symmetric spaces and generalised root systems. The reader that wants to enter the details is referred to [35,38,40].

### 4.1. Globally Symmetric Spaces

A symmetric space is an analytic Riemannian manifold, that is, a manifold endowed with an analytic structure and an analytic metric such that each of its points is an isolated fixed point for an involutive isometry. This is an isometry whose square is the identity. We will not give a detailed account of the consequences of this definition; we will just give a short intuitive description of compact symmetric manifolds at the level necessary for the applications in the next sections. Let $G$ be a finite dimensional compact Lie group and $K \subset G$ a proper Lie subgroup of $G$. With $\mathfrak{g}$ and $\mathfrak{k} \subset \mathfrak{g}$, we indicate the corresponding Lie algebras. Let $\mathfrak{p}$ be a linear complement of $\mathfrak{k}$ in $\mathfrak{g}$ (if $G$ is semi-simple, we can take the orthogonal complement with regard to the Killing form). We say that $K$ *is symmetrically embedded in $G$* if the corresponding Lie algebras satisfy

$$[\mathfrak{k}, \mathfrak{k}] \subseteq \mathfrak{k}, \tag{68}$$

$$[\mathfrak{k}, \mathfrak{p}] \subseteq \mathfrak{p}, \tag{69}$$

$$[\mathfrak{p}, \mathfrak{p}] \subseteq \mathfrak{k}. \tag{70}$$

Notice that $\mathfrak{p}$ is not a subalgebra unless it is abelian. In order to understand the meaning of these relations, let us pass to the groups. The point is that, generically, a globally symmetric manifold $\mathcal{M}$ has the form

$$\mathcal{M} = G/K, \tag{71}$$

where $K$ is symmetrically embedded in $G$. Of course

$$\dim \mathcal{M} = \dim G - \dim K = \dim \mathfrak{p}. \tag{72}$$

Roughly speaking, thinking of elements of a Lie algebra as left invariant vector fields, we can look at $\mathfrak{p}$ as defining at any point the tangent space to $\mathcal{M}$, obtained cutting out the

quotiented directions $\mathfrak{k}$ from $\mathfrak{g}$. However, it is more interesting to think about the fields as infinitesimal generators of the natural action of $G$ over $\mathcal{M} = G/K$. Indeed, obviously the action of $K$ over $\mathcal{M}$ leaves its points fixed and can then be identified with the isotropy group of $\mathcal{M}$. In contrast, $\mathfrak{p}$ represents the infinitesimal translations of $\mathcal{M}$ since they generate shifts from a point to a different one. In this view, (69) simply represents the action of the isotropy group on translations. Interestingly, (70) shows that translations usually do not commute since their commutator gives rise to a transformation under the isotropy group. This is to say that the manifold $M$ is curved. We can clarify further what we just said by considering the case $G$ is a simple group.

*4.2. A Little Bit of Differential Geometry*

The main characteristic of the case when $G$ is simple is that it is essentially endowed by a natural metric, the Killing metric $K$. First, recall that any Lie group $G$ is endowed with the Maurer–Cartan 1-form $j$, a $\mathfrak{g}$-valued left invariant 1-form, defined by

$$j(X) = X_e \tag{73}$$

for every left invariant vector field $X$ over $G$, $j$ does satisfy the Maurer–Cartan equation

$$dj + \frac{1}{2}[j, j] = 0, \tag{74}$$

where $[,]$ is the Lie product combined with the wedge product, as usual. To see this, it is sufficient to evaluate $dj$ on a pair of left invariant fields $X, Y$. Using the Cartan formula

$$dj(X, Y) = X(j(Y)) - Y(j(X)) - j([X, Y]) = -j([X, Y]), \tag{75}$$

where in the last step, we used that $j(X) = X_e$ and $j(Y) = Y_e$ are constants. Now,

$$j([X, Y]) = [X, Y]_e = [X_e, Y_e] = [j(X), j(Y)] = \frac{1}{2}[j, j](X, Y), \tag{76}$$

which proves the above equation.

For a compact simple Lie group, $j$ is related to the bi-invariant metric $ds^2$ over $G$ by

$$ds^2 = -\lambda^2 K(j \otimes j), \tag{77}$$

where $\lambda$ is a real normalisation constant. Since $G$ is compact, the minus sign ensures strict positivity. Set $n = \dim G$, $k = \dim K$ and choose an orthonormal (with regard to the metric $-\lambda^2 K$) basis $\tau_i$, $i = 1, \ldots, n$ of $\mathfrak{g} = Lie(G)$, assuming that the first $k$ elements are a basis for $\mathfrak{k} = Lie(K)$ so that the remaining ones generate $\mathfrak{p}$. We will use first Latin indices $a, b, c, d$ running from 1 to $k$ and Greek indices $\alpha, \beta, \gamma, \delta$ running from $k + 1$ to $n$.

As discussed in the next section, it can be shown that the generic element of the (compact) group can be written in the form

$$G = PK, \tag{78}$$

where $P = \exp(\mathfrak{p})$, $K = \exp(\mathfrak{k})$. Therefore, $P = G/K$ is a good representative for the quotient space. Let us consider the restriction of $j$ to $P$

$$j_P = j|_P. \tag{79}$$

Notice that $\mathfrak{g} = \mathfrak{p} \oplus \mathfrak{k}$ is an orthogonal direct sum with respect to the Killing metric. Accordingly, we can write

$$j_P = j_P^{\parallel} \oplus j_P^{\perp}, \tag{80}$$

where $j_P^\parallel$ takes value in $\mathfrak{p}$, while $j_P^\perp$ takes value in $\mathfrak{k}$. It follows that

$$d\sigma^2 = -\lambda^2 K(j_P^\parallel \otimes j_P^\parallel) \tag{81}$$

defines a positive definite metric over $\boldsymbol{P}$. Since $K$ is bi-invariant, the metric is well defined under the left action of $\boldsymbol{K}$. Using the Maurer–Cartan equation and (68), (69), and (70), we immediately obtain

$$dj_P^\parallel = -[j_P^\perp, j_P^\parallel]. \tag{82}$$

It is now convenient to introduce the structure constants of the algebra relative to a given basis $\tau_A$, $A = 1, \ldots, N$ by

$$[\tau_A, \tau_B] = c_{AB}{}^C \tau_C, \tag{83}$$

where the usual summation convention is adopted. Because (68)–(70), we have that the non-vanishing structure constants are

$$c_{ab}{}^c, \quad c_{\alpha\beta}{}^c, \quad c_{\alpha b}{}^\gamma = -c_{b\alpha}{}^\gamma. \tag{84}$$

Thus, Equation (82) can be rewritten as

$$dj_P^\alpha = -c_{b\beta}{}^\alpha j_P^b \wedge j_P^\beta. \tag{85}$$

Now, since the basis we have chosen is orthonormal with regard to the metric $-\lambda^2 K$, we have that

$$d\sigma^2 = j_P^\alpha \otimes j_P^\beta \delta_{\alpha\beta}, \tag{86}$$

so that $j_P^\alpha$ can be thought as a vielbein. Comparing with the structure equation

$$dj_P^\alpha = -\omega^\alpha{}_\beta \wedge j_P^\beta, \tag{87}$$

we obtain the following expression for the spin connection

$$\omega^\alpha{}_\beta = c_{a\alpha}{}^\beta j_P^a. \tag{88}$$

We can also compute the curvature tensor $\Omega^\alpha{}_\beta = d\omega^\alpha{}_\beta + \omega^\alpha{}_\gamma \wedge \omega^\gamma{}_\beta$,

$$\Omega^\alpha{}_\beta = \frac{1}{2} j_P^\gamma \wedge j_P^\delta c_{\gamma\delta}{}^a c_{\beta a}{}^\alpha, \tag{89}$$

which means

$$\Omega^\alpha{}_{\beta\gamma\delta} = c_{\gamma\delta}{}^a c_{\beta a}{}^\alpha. \tag{90}$$

Notice that

$$K(\tau_\alpha, \tau_\beta) = \mathrm{Tr}(ad_{\tau_\alpha} \circ ad_{\tau_\beta}) = c_{\alpha a}{}^\gamma c_{\beta\gamma}{}^a, \tag{91}$$

which means that our choice of $\lambda$ is such that

$$-\lambda^2 c_{\alpha a}{}^\gamma c_{\beta\gamma}{}^a = \delta_{\alpha\beta}, \tag{92}$$

which is always possible for a compact simple Lie group. It immediately follows that the Ricci tensor has components, with regard to the vielbein,

$$R_{\alpha\beta} = \Omega^\gamma_{\alpha\gamma\beta} = \frac{1}{\lambda^2}\delta_{\alpha\beta}, \tag{93}$$

so that

$$\boldsymbol{R} = R_{\alpha\beta}\, \boldsymbol{j}^\alpha_{\boldsymbol{P}} \otimes \boldsymbol{j}^\beta_{\boldsymbol{P}} = \frac{1}{\lambda^2}\, ds^2. \tag{94}$$

Thus, $M = \boldsymbol{G}/\boldsymbol{K}$ is an Einstein manifold with scalar curvature

$$R = \frac{n}{\lambda^2}. \tag{95}$$

*4.3. Real Forms, Subgroups, and Lattices.*

We know essentially all about the geometry of the quotient space once we have selected the subgroup $\boldsymbol{K}$ symmetrically embedded in $\boldsymbol{G}$. It remains to understand how many of such subgroups can be found in a given compact simple Lie group. This problem can be completely solved by passing, as usual, to the corresponding Lie algebras, as Shôhô Araki did in [40].

We will not report the details of the derivation, which can be found in [40], but just the result of the classification. Before doing so, we just provide a rough idea on what to expect, once again based on our simple example $\boldsymbol{SU}(3)$. We have seen that in that case there are two maximally symmetric embedded subgroups, one split and the other non-split . The split one is characterised by having a Cartan subalgebra of rank two all in the complement of the proper subalgebra. The non-split case instead has the property that the minimal possible dimension for the intersection between the Cartan subalgebra and the proper maximal subalgebra is one. The two Cartan subalgebras are generated by $\lambda_3$ and $\lambda_8$ in the first case and by $\lambda_4$ and $\tilde{\lambda}$ in the second case. Diagonalizing the adjoint action of the two systems, we obtain the same hexagonal pictures as usual and, therefore, the same two-dimensional lattices. Now, recall that we are interested in the quotient space. After quotienting, both $\lambda_3$ and $\lambda_8$ survive, and the corresponding lattice is the same as for the whole Lie algebra. In the non-split case, instead, only the $\lambda_4$ component survives, and the lattice is projected down to a one-dimensional lattice as shown in Figure 3.

The root system, consisting of the green dots in the figure and generating a one-dimensional lattice, is very different from the previous one. It contains double roots, and there are also roots of the form $k\alpha$ with $k \neq 0, \pm 1$; that is, there are also the cases $k = \pm 2$ and $k = \pm 1/2$. This is not a proper root system. This root system characterises the non-split quotient, while the split one is characterised by the root system of the starting algebra. The allowability of these two root systems is related to the complexification properties of the Lie algebra of $\boldsymbol{SU}(3)$. In both cases, it happens that the map $\iota : \mathfrak{p} \mapsto i\mathfrak{p}$, where $i$ is the imaginary unit, transforms the compact form into a non-compact real algebra (with the same complexification of the compact form). On the other hand, this property is strictly related to the symmetry of the embedding since $\iota(\mathfrak{g})$ is real if and only if (68)–(70) are all satisfied.

To classify all symmetric spaces is therefore equivalent to classify the maximal proper compact subgroups of the compact form, which is equivalent to classify all admissible $\iota$ mappings, which is equivalent to classify all possible real forms, which, finally, is equivalent to classify all admissible corresponding lattices. The reader interested in the details can see [35].

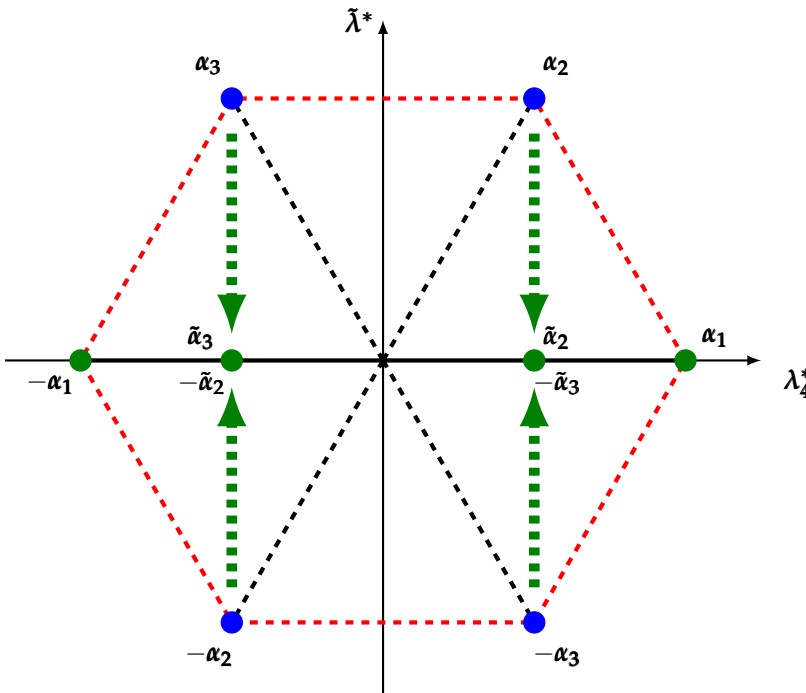

**Figure 3.** The non-vanishing roots of the non-split quotient.

We now briefly describe a few more instruments necessary to describe the final result in a compact form: the Satake diagrams. They are the analogue of Dynkin diagrams, but now for classifying the non-compact real forms (indeed, they work also on more general fields). First, notice that all groups we are working with are in fact algebraic groups over $\mathbb{R}$. The algebraic closure of $\mathbb{R}$ is $\mathbb{C}$, and the Galois group of the extension $\mathbb{C}/\mathbb{R}$ has just two elements (which are the identity and the conjugation, the only automorphisms of $\mathbb{C}$ that leave $\mathbb{R}$ invariant). Now, consider a non-compact real form of the group. One first looks for the maximal split torus (with respect to $\mathbb{C}$), which is a maximal torus with respect to the property to be isomorphic to $(\mathbb{C}^*)^\ell$ for some $\ell$, when viewed over $\bar{\mathbb{R}} = \mathbb{C}$; $\ell$ is the $\mathbb{R}$-rank of the torus. Let us call $T^\ell$ the split torus. It is just generated by the maximal component of the Cartan subalgebra in the orthogonal complement to the maximal compact subalgebra. Thus $\ell$; is the rank of the associated symmetric space. With $T^r$, we indicate the maximal torus of the complexification $G^\mathbb{C}$ of the group; $r$ is the rank of the group. Assuming $G^\mathbb{C}$ simple, after choosing a fundamental system of roots in $T^r$, we can draw the corresponding Dynkin diagram. From it, we can obtain the Satake diagram as follows. Some of the roots in the diagram vanish when computed in $T^\ell$. The corresponding dots in the diagram are filled in black. Next, the Galois group acts on the Dynkin diagram identifying some pairs of the white dots. The "surviving" white part of the diagram, with the suitable identifications drawn as arrows, determines the allowed root systems identifying the real form.

These root systems are not necessarily reduced. Recall that a root system $\mathcal{R}$ is a finite subset of a finite dimensional real vector space, with positive scalar product $(|)$, satisfying the following two condition. For any pairs of non-vanishing elements $\alpha, \beta \in \mathcal{R}$,

1. The numbers

$$p_{\alpha,\beta} = 2\frac{(\alpha|\beta)}{(\beta|\beta)} \tag{96}$$

are integers;

2. $\alpha - p_{\alpha,\beta}\beta \in \mathcal{R}$.

$\mathcal{R}$ is said to be "reduced" if for a given non-vanishing $\alpha \in \mathcal{R}$, the only non-vanishing elements in $\mathcal{R}$ of the form $k\alpha$ are for $k = \pm 1$. This is what happens for the root system of

a simple group. Most of the properties of such a root system follow from the above two properties but not the condition to be reduced. Indeed, from the first condition, we see that, if $\alpha \in \mathcal{R}$ is different from zero, then both $p_{k\alpha,\alpha} = 2k$ and $p_{\alpha,k\alpha} = 2/k$ must be integers. This is possible only for $k = \pm 1, k = \pm 2, k = \pm 1/2$. In particular, if $2\alpha$ is a root, then $\alpha/2$ is not, and vice versa. Therefore, if there are two parallel roots of different length, they are naturally called the "shorter" root and the "longer" root. When the only coefficients are $k = \pm 1$, the root is called shorter, and the longer root is said to have zero multiplicity.

We are now ready to draw all possible Satake diagrams and classify all real forms (for the details see [35,40]). For the Dynkin diagrams, we refer to Appendix A.

Non-Compact Real Forms

The compact forms of rank $r$ are the $\mathfrak{su}(r+1, \mathbb{R})$ special unitary algebras. There are four kind of Satake diagrams.

**AI.** The diagrams coincide with the Dynkin diagrams and correspond to the split real forms. The lattice is the same as the one of the algebra. There are only shorter roots with multiplicity one. They correspond to the algebras $\mathfrak{sl}(r+1, \mathbb{R})$, consisting of the $(r+1) \times (r+1)$ traceless matrices with real entries and have $r(r+1)$ compact directions corresponding to the subalgebra of $(r+1) \times (r+1)$ antisymmetric matrices. They generate the symmetrically embedded maximal compact subalgebra (MCS) $\mathfrak{so}(r+1)$ of $r+1$ dimensional rotations.

**AII.** These diagrams exist for rank $r = 2n - 1, n > 1$.

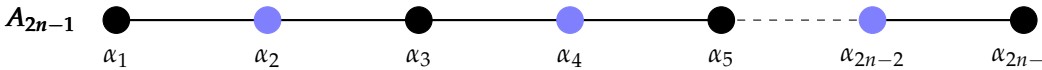

The corresponding lattice is of Dynkin type $A_{n-1}$ with only shorter roots having multiplicity four. The algebra is $\mathfrak{su}^*(2n)$. It is obtained from the compact form by selecting the maximal subalgebra $\mathfrak{usp}^*(2n)$ and applying the $\iota$ involution to its orthogonal complement. The MCS is $\mathfrak{usp}^*(2n)$.

**AIIIa.** These are defined for rank $r = n - 1$, $n = p + q$ with $1 < p < q$.

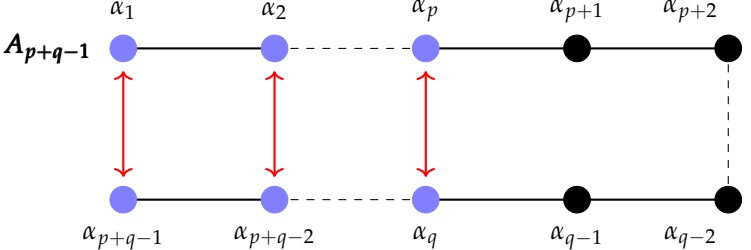

The corresponding lattice is of Dynkin type $B_p$. The long roots of $B_p$ have shorter multiplicity two and longer multiplicity zero. The short roots of $B_p$ have shorter multiplicity $2(q - p)$ and longer multiplicity 1. The algebra is $\mathfrak{su}(p, q)$, the Lie algebra of special complex transformations leaving invariant a hermitian scalar product with signature $(p, q)$. The MCS is $\mathfrak{s}(\mathfrak{u}(p) \oplus \mathfrak{u}(q))$. Obviously $\mathfrak{su}(q, p) \simeq \mathfrak{su}(p, q)$.

**AIIIb.** These are defined for rank $r = 2p - 1$, $p > 1$.

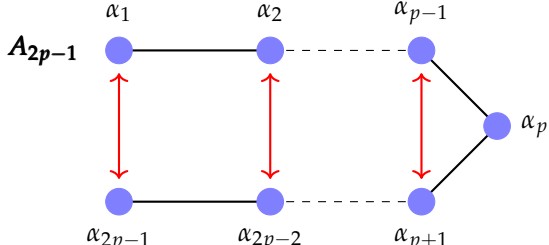

The corresponding lattice is of Dynkin type $C_p$. There are only shorter roots. The long roots of $C_p$ have multiplicity two and the short roots have multiplicity one. The algebra is $\mathfrak{su}(p,q)$, with MCS $\mathfrak{s}(\mathbf{u}(p) \oplus \mathbf{u}(p))$.

**AIV.** Exist for rank $n > 1$.

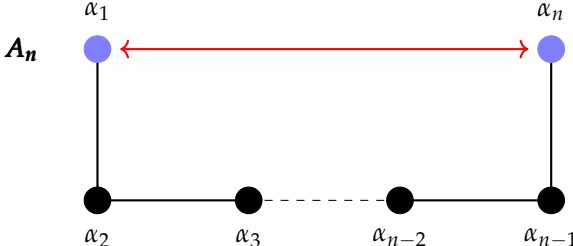

The corresponding lattice is of Dynkin type $A_1$. There is a shorter root of multiplicity $2n - 2$ and a longer root of multiplicity 1. The algebra is $\mathfrak{su}(1,n)$, the Lie algebra of complex Lorentz transformations. The MCS is $\mathfrak{s}(\mathbf{u}(1) \oplus \mathbf{u}(n))$. Obviously $\mathfrak{su}(n,1) \simeq \mathfrak{su}(1,n)$.

**BI.** Exist for rank $n > 1, 2 \le p \le n$.

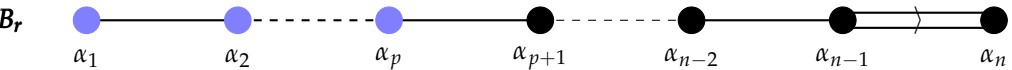

The corresponding lattice is of Dynkin type $B_p$. There are only shorter roots. The short roots of $B_p$ have multiplicity $2(n - p) + 1$, while the long ones have multiplicity 1. The algebra is $\mathfrak{so}(p,q)$, $p + q = n + 1$, the Lie algebra of linear special transformations leaving invariant a scalar product with signature $(p,q)$. The MCS is $\mathfrak{so}(p) \oplus \mathfrak{so}(q)$. Obviously, $\mathfrak{so}(q,p) \simeq \mathfrak{so}(p,q)$.

**BII.** Exist for rank $n > 1$.

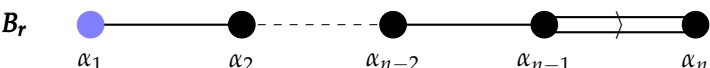

The corresponding lattice is of Dynkin type $A_1$, with only a shorter root of multiplicity $2n - 1$. The algebra is $\mathfrak{so}(1, 2n)$, the Lie algebra of the Lorentz group in odd spacetime dimensions larger than three. The MCS is $\mathfrak{so}(2n)$. Obviously $\mathfrak{so}(2n,1) \simeq \mathfrak{so}(1,2n)$.

**CI.** Exist for rank $n > 2$.

   The Satake diagram is the same as the Dynkin diagram for $C_n$, and the lattice is of Dynkin type $C_n$, with only shorter roots of multiplicity one. The algebra is $\mathfrak{sp}(2n)$, the symplectic algebra. The MCS is $\mathbf{u}(n)$.

**CIIa.** Exist for rank $n > 2, 0 < p < n/2$.

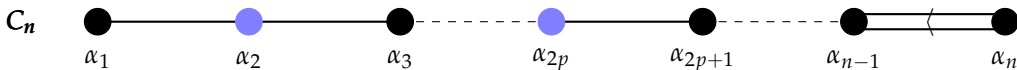

The corresponding lattice is of Dynkin type $B_p$. Its long roots are only shorter with multiplicity 4, while the short roots are both shorter with multiplicity $4n - 8p$ and longer with multiplicity 3. The algebra is $\mathfrak{usp}(2p, 2q)$, $p + q = n$, the Lie algebra of transformations preserving a symplectic form with signature $(p, q)$. The MCS is $\mathfrak{usp}(2p) \oplus \mathfrak{usp}(2q)$. Obviously, $\mathfrak{usp}(2q, 2p) \simeq \mathfrak{usp}(2p, 2q)$.

**CIIb.** Exist for rank $2n$, $n > 1$.

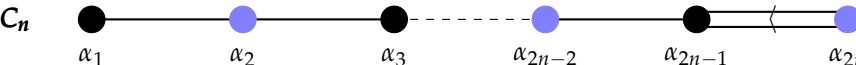

The corresponding lattice is of Dynkin type $C_n$. Its roots are only shorter, the short ones with multiplicity four, while the long ones with multiplicity three. The algebra is $\mathfrak{usp}(2n, 2n)$. The MCS is $\mathfrak{usp}(2n) \oplus \mathfrak{usp}(2n)$.

**DIa.** They exist for rank $n \geq 4$.
The Satake diagram is the same as the Dynkin diagram for $D_n$, and the lattice is of Dynkin type $D_n$, with only shorter roots of multiplicity one. The algebra is $\mathfrak{so}(n, n)$. The MCS is $\mathfrak{so}(n) \oplus \mathfrak{so}(n)$.

**DIb.** Exist for rank $n > 2$.

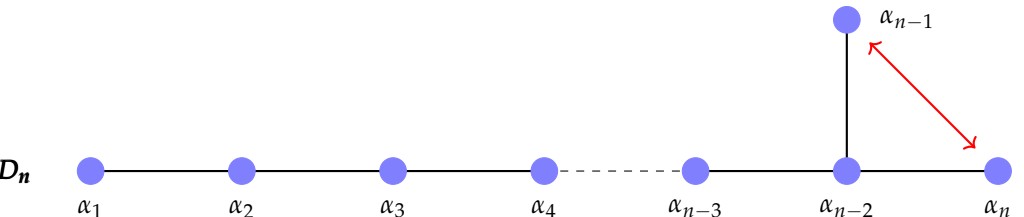

The corresponding lattice is of Dynkin type $B_{n-1}$. It has only shorter roots, the long ones with multiplicity one, while the short ones with multiplicity two. The algebra is $\mathfrak{so}(n - 1, n + 1)$, with MCS $\mathfrak{so}(n - 1) \oplus \mathfrak{so}(n + 1)$. Of course, $\mathfrak{so}(n + 1, n - 1) \simeq \mathfrak{so}(n - 1, n + 1)$.

**DIc.** Exist for rank $n > 2$, $1 < p < n - 1$.

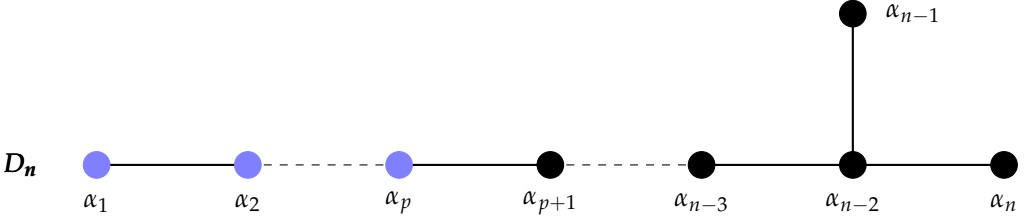

The lattice is of Dynkin type $B_p$ with only shorter roots, the long ones with multiplicity 1, while the short ones with multiplicity $2(n - p)$. The algebra is $\mathfrak{so}(p, q)$, $p + q = n$. The MCS is $\mathfrak{so}(p) \oplus \mathfrak{so}(q)$. Of course, $\mathfrak{so}(q, p) \simeq \mathfrak{so}(p, q)$.

**DII.** Exist for rank $n > 2$.

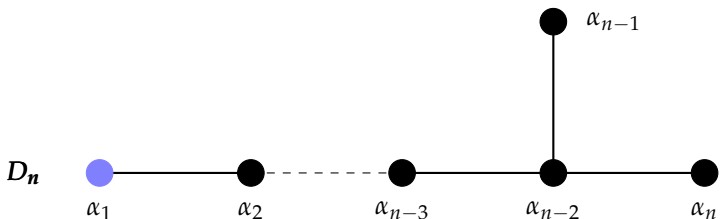

The corresponding lattice is of Dynkin type $A_1$. It has only a shorter root with multiplicity $2n - 2$. The algebra is $\mathfrak{so}(1, 2n - 1)$. The MCS is $\mathfrak{so}(2n - 1)$.

**DIIIa.** Exist for rank $2n + 1$, $n > 1$.

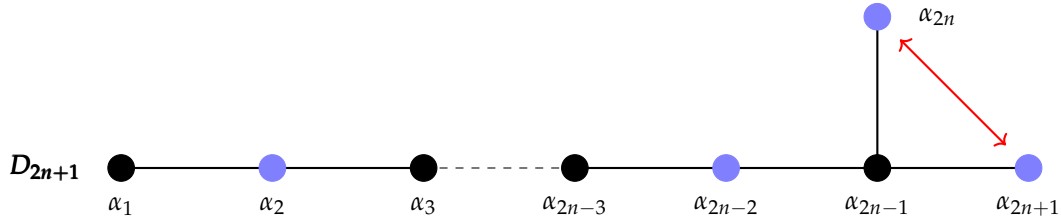

The associated lattice is of Dynkin type $B_n$. The long roots are only shorter with multiplicity four. The short roots are shorter, with multiplicity four and longer with multiplicity one. The algebra is $\mathfrak{so}^*(4n + 2)$ and is obtained from the compact form by selecting a maximal $\mathfrak{u}(2n + 1)$ and applying the map $\iota$ to its orthogonal complement. The MCS is $\mathfrak{u}(2n + 1)$.

**DIIIb.** Exist for rank $2n$, $n > 1$.

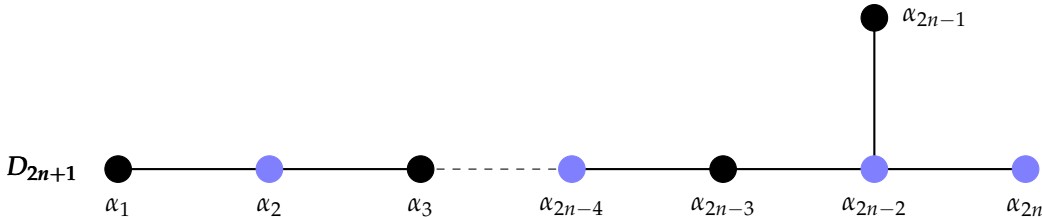

The corresponding lattice is of Dynkin type $C_n$. There are only shorter roots of which the short ones have multiplicity four, and the long ones have multiplicity one. The algebra is $\mathfrak{so}^*(4n)$ and is obtained from the compact form by selecting a maximal $\mathfrak{u}(2n)$ and applying the map $\iota$ to its orthogonal complement. The MCS is $\mathfrak{u}(2n)$.

**EI.** It is of type $E_6$.
The Satake diagram is the same as the Dynkin diagram for $E_6$, and the lattice is of Dynkin type $E_6$, with only shorter roots of multiplicity one. The algebra is $\mathfrak{e}_{6(6)}$. The MCS is $\mathfrak{usp}(8)$.

**EII.** It is of type $E_6$.

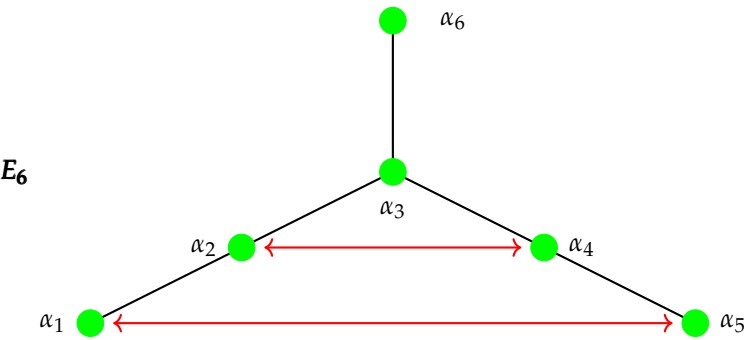

The corresponding lattice is of Dynkin type $F_4$. There are only shorter roots of which the short ones have multiplicitytwo, and the long ones have multiplicity one. The algebra is $\mathfrak{e}_{6(2)}$. The MCS is $\mathfrak{usp}(2) \oplus \mathfrak{su}(6)$.

**EIII.** It is of type $E_6$.

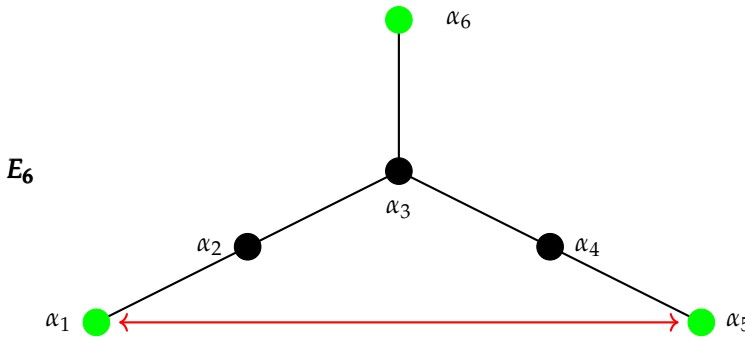

The corresponding lattice is of Dynkin type $B_2$. The long roots are only shorter with multiplicity six. The short roots have shorter components with multiplicity eight and longer components with multiplicity one. The algebra is $\mathfrak{e}_{6(-14)}$. The MCS is $\mathfrak{u}(1) \oplus \mathfrak{so}(10)$.

**EIV.** It is of type $E_6$.

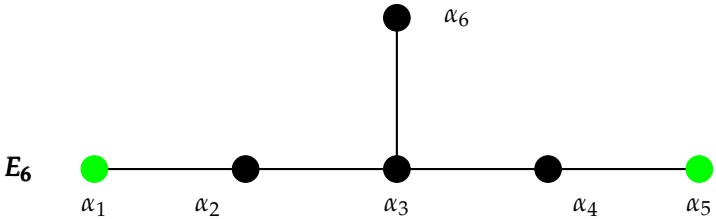

The corresponding lattice is of Dynkin type $A_2$, with only shorter roots of multiplicity eight. The algebra is $\mathfrak{e}_{6(-26)}$. The MCS is $\mathfrak{f}_4$.

**EV.** It is of type $E_7$.
The Satake diagram is the same as the Dynkin diagram for $E_7$, and the lattice is of Dynkin type $E_7$ with only shorter roots of multiplicity one. The algebra is $\mathfrak{e}_{7(7)}$. The MCS is $\mathfrak{su}(8)$.

**EVI.** It is of type $E_7$.

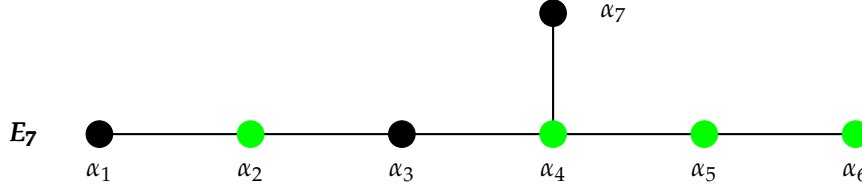

The corresponding lattice is of Dynkin type $F_4$, with only shorter roots, the long ones of multiplicity one and the short ones of multiplicity four. The algebra is $\mathfrak{e}_{7(-5)}$. The MCS is $\mathfrak{su}(2) \oplus \mathfrak{so}(12)$.

**EVII.** It is of type $E_7$.

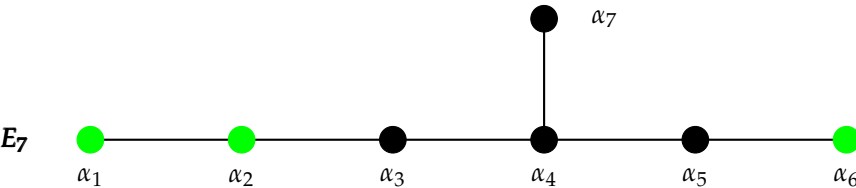

The corresponding lattice is of Dynkin type $C_3$, with only shorter roots, the long ones of multiplicity one and the short ones of multiplicity eight. The algebra is $\mathbf{e}_{7(-25)}$. The MCS is $\mathbf{u}(1) \oplus \mathbf{e}_6$.

**EVIII.** It is of type $E_8$.
The Satake diagram is the same as the Dynkin diagram for $E_8$, and the lattice is of Dynkin type $E_8$, with only shorter roots of multiplicity one. The algebra is $\mathbf{e}_{8(8)}$. The MCS is $\mathbf{so}(16)$.

**EIX.** It is of type $E_8$.

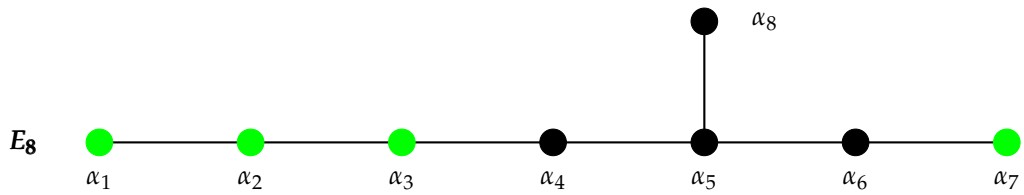

The corresponding lattice is of Dynkin type $F_4$, with only shorter roots, the long ones of multiplicity one and the short ones of multiplicity eight. The algebra is $\mathbf{e}_{8(-24)}$. The MCS is $\mathbf{su}(2) \oplus \mathbf{e}_7$.

**FI.** It is of type $F_4$.
The Satake diagram is the same as the Dynkin diagram for $F_4$, and the lattice is of Dynkin type $F_4$, with only shorter roots of multiplicity one. The algebra is $\mathbf{f}_{4(4)}$. The MCS is $\mathbf{usp}(6) \oplus \mathbf{usp}(2)$.

**FII.** It is of type $F_4$.

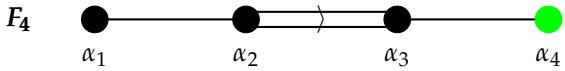

The corresponding lattice is of Dynkin type $A_1$, with only shorter roots of multiplicity eight. The algebra is $\mathbf{f}_{4(-20)}$. The MCS is $\mathbf{so}(9)$.

**G.** It is of type $G_2$.
The Satake diagram is the same as the Dynkin diagram for $G_2$, and the lattice is of Dynkin type $G_2$, with only shorter roots of multiplicity one. The algebra is $\mathbf{g}_{2(2)}$. The MCS is $\mathbf{so}(4)$.

All these real forms can be obtained by the corresponding compact form, selecting the right proper maximal subalgebra and multiplying the complementary generators by $i$.

## 5. Generalised Euler Angles

Here we define the generalised Euler parametrisation following [1].

### 5.1. The General Strategy

The Euler angles for compact simple Lie groups have been provided in [2–5,8–10,14,15]. We have already shown the basic idea in the introductory section; however, let us discuss the general strategy once again.

Let $G$ be a compact Lie group and $\mathbf{g} \simeq \mathrm{Lie}(G)$ some matrix realisation of the Lie algebra, supporting a faithful representation of $G \simeq \exp \mathbf{g}$. Let $H$ be a maximal symmetrically

embedded proper subgroup of $G$. We know that there is a non-compact real form $G'$ whose maximal compact subgroup is $H$ and that there is a one to one correspondence between maximal symmetrically embedded proper subgroups of $G$ and its real forms. In particular, when $G'$ is the split form, we will call $H$ *the maximal compact subgroup of $G$ or* MCS. The MCS has the property

$$\dim G = 2h + r, \tag{97}$$

where $h = \dim H$, and $r$ is the rank of $G$. In this case, the Euler parameterization of $G$ takes the form

$$\begin{align}
G[\vec{x}; \bar{y}; \vec{z}] &= H[\vec{x}] T^r[\bar{y}] H[\vec{z}], \tag{98}\\
T^r[\bar{y}] &= \exp(y_1 c_1 + \ldots + y_r c_r), \tag{99}\\
\vec{x} &= (x_1, \ldots, x_h),\\
\bar{y} &= (y_1, \ldots, y_r),\\
\vec{z} &= (z_1, \ldots, z_h),
\end{align}$$

where $H[\vec{x}]$ is any parametrization of $H$, and $c_1, \ldots, c_r$ is a basis for a Cartan subalgebra $\mathfrak{c}$ complementary to $\mathrm{Lie}(H)$ in $\mathfrak{g}$. The $c_i$ can always be chosen so that $y_i$ is periodic. However, if we allow the coordinates $\bar{y}$ to cover all periods and $\vec{x}, \vec{z}$ such to describe the whole $H$, then we obtain a redundant parametrization for two reasons.

First, the intersection $\Gamma = H \cap T^r$ is a non-trivial finite group. We will see that $\Gamma \simeq (\mathbb{Z}_2)^r$ if $G$ is simply connected, or to one of its proper subgroups otherwise. Therefore,

$$H[\vec{x}] T^r[\bar{y}] H[\vec{z}] = H[\vec{x}] \gamma^{-1} T^r[\bar{y}] \gamma H[\vec{z}],$$

for any $\gamma \in \Gamma$. We can write

$$H[\vec{x}] \gamma^{-1} = H[\tilde{\vec{x}}], \tag{100}$$

which shows that we must reduce the range of the coordinates $\vec{x}$ with regard to the action of $\Gamma$.

The second origin of redundancies is the fact that the Weyl group $W$ acts non-trivially on $t \in T^r$

$$t \mapsto w^{-1} t w \in T^r, \tag{101}$$

for any $w \in W$.[6] This redundancy can be resolved by also reducing the range of the coordinates $\bar{y}$. As we will see, this problem can be completely characterised in terms of the highest root of $G$. A suitable linear change of variables $s_i = \sum_{j=1}^r A_{ij} y^j$ will allow us to express the right range of coordinates as the set of inequalities

$$0 \le n_1 s_1 + \ldots + n_r s_r \le \pi, \qquad 0 \le s_i \le \pi, \quad i = 1, \ldots, r, \tag{102}$$

where $(n_1, \ldots, n_r)$ are the coefficients of the highest root $\tilde{\alpha}$ with regard to a basis of simple roots, $\tilde{\alpha} = n_1 \alpha_1 + \ldots + n_r \alpha_r$.

This construction is easily extended to non-MCS subgroups $H$ maximally symmetrically embedded in $G$.

Notice that we cannot limit ourselves to work with simple groups. Let $G_0$ be a compact connected semi-simple Lie group, $G_0 \simeq G_1 \times G_2 \times \cdots \times G_n$, with $G_i$, $i = 1, \ldots n$ simple Lie groups, uniquely determined up to permutations. Take $H \simeq H_1 \times H_2 \times \cdots \times H_n$ with $H_i$ a maximal Lie subgroup symmetrically embedded in $G_i$. $H$ is connected (see [38], Chapter VI, Theorem 1.1). In general, $H_i$ is not simple, nor semi-simple, but it has the form $H_i \simeq H_{0,i} \times T^{s_i} / \Delta_i$, where $H_{0,i}$ is semi-simple, $T^{s_i}$ is an abelian torus, and $\Delta_i$ a finite subgroup. Since our aim is to construct the Euler parameterisation of $G_0$ relative to the

subgroup $H$, and then apply the same procedure to $H$ inductively, we are forced to consider the more general case

$$G \simeq G_0 \times T^s / \Delta, \tag{103}$$

with $G_0$ as before, $T^s$ an abelian torus, and $\Delta$ a finite subgroup. We have already seen in Section 3 that this is the general form of any connected compact Lie group. Obviously, the parametrisation of $G$ from $G_0$ is quite elementary, so we will assume

$$G \equiv G_0. \tag{104}$$

Notice that $H$ is symmetrically embedded in $G$, but is not maximal, unless $G$ is simple. Of course, it is sufficient to understand the last case.

*5.2. MCS Euler Parametrisation*

As we just said, in order to realise the Euler parametrisation it is sufficient to know how to parametrise its semi-simple part and, indeed, its simple components. Therefore, we will assume, without loss of generality, that $G$ is a compact connected semi-simple Lie group. We want to construct a Euler parametrisation of $G$ relative to a specific choice of the corresponding subgroup $H$. In this section, we will assume that $H$ is MCS, meaning that $H_i$ is an MCS subgroup of $G_i$ for all $i$. Let $\mathfrak{g}$ and $\mathfrak{h}$ be the Lie algebras of $G$ and $H$, respectively. In the MCS case, we can fix a Cartan subalgebra $\mathfrak{c}$ of $\mathfrak{g}$ such that $\mathfrak{c} \cap \mathfrak{h} = 0$, so that the Euler parametrisation of $G$ with regard to $H$ takes the form

$$G = (H'/\Gamma)T^r H, \tag{105}$$

where $H'$ is a copy of $H$, and $\Gamma = H \cap e^{\mathfrak{c}}$ is a finite subgroup of the maximal torus that will be specified later.

The problem of parameterising $H$ and $H'$ is then the same as for $G$, with the caveat that they will not be semi-simple, and can be obtained inductively. In order to obtain a (almost everywhere) one to one parametrisation of $G$, beyond individuating $\Gamma$, we have to just,determine the right range for the angles of the toric part

$$T^r[\bar{y}] = \exp(y_1 c_1 + \ldots + y_r c_r). \tag{106}$$

We will show that the range for the $y$'s is independent from the starting $G$-faithful representation $R$ but depend only on the Adjoint representation. In other words, the details discriminating among the different compact forms of $G$ will depend only from the periodicities of the $S^1$ factors entering the paramtrisations and the action of the finite subgroups. Furthermore, since $G \simeq G_1 \times G_2 \times \cdots \times G_n$, we can parametrise each factor independently. Therefore, without loss of generality, we focus on the case when $G$ is simple. In particular, then, $H$ is maximal in $G$.

For the algebra, we can write

$$\mathfrak{g} = \mathfrak{h} \oplus \mathfrak{p} = \mathfrak{h} \oplus \mathfrak{c} \oplus \mathfrak{p}', \tag{107}$$

with $\dim \mathfrak{h} = \dim \mathfrak{p}' = h$ and $\dim \mathfrak{c} = r$. Here, the prime just indicates that $\mathfrak{p}'$ is relative to the copy $H'$ of $H$. Let $\mathfrak{g}_{\mathbb{C}}$ be the complexification of $\mathfrak{g}$. We can write $\mathfrak{g}_{\mathbb{C}} = \Lambda_- \oplus \mathfrak{c} \oplus \Lambda_+$, where $\Lambda_{\pm}$ is the direct sum of the root spaces $\Lambda_{\pm\alpha}$ such that $\alpha$ is a positive root. We can select two convenient bases for $\mathfrak{g}_{\mathbb{C}}$: The *Weyl basis*

$$\{c_i\}_{i=1}^r \cup \{\lambda_{\alpha_a} \cup \lambda_{-\alpha_a}\}_{a=1}^h, \tag{108}$$

where $\lambda_\alpha$ indicates the eigenvector corresponding to the root $\alpha$, and the $\alpha_a$ are the positive roots; the *compact basis*

$$\{c_i\}_{i=1}^r \cup \{t_a\}_{a=1}^h \cup \{p_b\}_{b=1}^h, \tag{109}$$

where $t_a$ and $p_b$ generate $\mathfrak{h}$ and $\mathfrak{p}'$, respectively, and are chosen so that $\mathrm{ad}_{t_a}$ and $\mathrm{ad}_{p_b}$ are diagonalisable, and the decomposition is Killing orthogonal. Of course, only the second one is a real basis for the compact algebra $\mathfrak{g}$. It satisfies the following relations:

$$\begin{aligned}
[t_a, t_b] &\in \mathfrak{h}, & [t_a, c_i] &\in \mathfrak{p}', & [t_a, p_b] &\in \mathfrak{p}', \\
[c_i, c_j] &= 0, & [c_i, p_b] &\in \mathfrak{h}, & [p_a, p_b] &\in \mathfrak{h}.
\end{aligned} \tag{110}$$

These follow directly from the symmetric properties of the embedding. In particular, ad-invariance of the Killing form $K$ implies

$$K([t_a, c_i], c_j) = K(t_a, [c_i, c_j]) = 0, \tag{111}$$

so that $[t_a, c_i] \in \mathfrak{p}'$.

There is a simple relation between the Weyl basis and the compact basis. Using the Weyl unitary trick, we obtain from $\mathfrak{g}$ a new real form with basis

$$\tilde{t}_a = t_a, \qquad \tilde{c}_j = ic_j, \qquad \tilde{p}_b = ip_b. \tag{112}$$

This is the split form $\mathfrak{g}_{(r)}$, with signature $r$. The operators $\mathrm{ad}_{\tilde{c}_j}$ are represented by symmetric matrices since ad-invariance and symmetry of the Killing form give

$$K([\tilde{c}_i, \tilde{p}_a], \tilde{t}_b) = -K(\tilde{p}_a, [\tilde{c}_i, \tilde{t}_b]), \tag{113}$$

and the form is negative definite over the $\tilde{t}_b$ and positive definite over the complementary space. Therefore, $\mathrm{ad}_{\tilde{c}_j}$ can be diagonalised, with real eigenvalues, by means of real combinations of the vectors $\tilde{t}_a$, $\tilde{p}_b$. It follows that an eigenvector corresponding to a non-zero root $\alpha$ must have the form $\lambda_\alpha = t_\alpha + ip_\alpha$, with $t_\alpha \in \mathfrak{h}$ and $p_\alpha \in \mathfrak{p}$, with both $t_\alpha$ and $p_\alpha$ non-zero. Indeed, $[c, \lambda_\alpha] = \alpha(c)\lambda_\alpha$ for all $c \in \mathfrak{c}$ implies

$$[c, t_\alpha] = i\alpha(c)p_\alpha, \quad [c, p_\alpha] = -i\alpha(c)t_\alpha, \tag{114}$$

so that $t_\alpha = 0$ or $p_\alpha = 0$ would imply $\alpha(c) = 0$ for any $c \in \mathfrak{c}$.

In conclusion, the relation between the Weyl basis and the compact basis is

$$\lambda_{\alpha_a} = t_a + ip_a, \qquad \lambda_{-\alpha_a} = t_a - ip_a, \qquad a = 1, \ldots, h. \tag{115}$$

The possibility to orthonormalise the compact basis follows from the fact that $\langle \lambda_\alpha, \lambda_\beta \rangle \neq 0$ if and only if $\alpha + \beta = 0$.

Now, let us go back to the Euler parametrisation. We can write

$$G[\vec{x}, \vec{y}, \vec{z}] = e^{\sum_{a=1}^h x_a t_a} e^{\sum_{i=1}^r y_i c_i} e^{\sum_{b=1}^h z_b t_b} \equiv (H'/\Gamma)T^r H. \tag{116}$$

Now, we can compute the invariant measure on $B = G/H$. Referring to the previous section, it is clear that it is simply given by

$$d\mu_B[\vec{x}; \vec{y}] = \sqrt{\det(d\sigma^2)} = |\det J_P^{\parallel}|. \tag{117}$$

A short calculation gives

$$d\mu_B[\vec{x}; \vec{y}] = |\det J(\vec{x}, \vec{y})| \prod_{\alpha=1}^h dx_\alpha \prod_{i=1}^r dy_i, \tag{118}$$

where $J$ is the $h \times h$ matrix with components

$$J^\alpha_\beta := K\left(\boldsymbol{T}^{r-1}\boldsymbol{H'}^{-1}\frac{\partial \boldsymbol{H'}}{\partial x_\alpha}\boldsymbol{T}^r, p_\beta\right). \tag{119}$$

Now, $\boldsymbol{H'}^{-1}d\boldsymbol{H'} =: J_{\boldsymbol{H}}$ is the left invariant one form for the $\boldsymbol{H}$ subgroup in the $\boldsymbol{H'}$ parametrisation,

$$J_{\boldsymbol{H}} = J_{\boldsymbol{H}}^\alpha t_\alpha = J_{\boldsymbol{H}}^\alpha{}_\beta t_\alpha dx^\beta \tag{120}$$

Thus,

$$d\mu_{\boldsymbol{B}}[\vec{x};\vec{y}] = d\mu_{\boldsymbol{H}}[\vec{x}]\det M \prod_{i=1}^r dy_i, \qquad M^a_b := \langle e^{-\boldsymbol{\mathfrak{c}}}t_a e^{\boldsymbol{\mathfrak{c}}}, p_b\rangle. \tag{121}$$

Now, $t_a = (\lambda_{\alpha_a} + \lambda_{-\alpha_a})/2$ so that

$$e^{-\boldsymbol{\mathfrak{c}}}t_a e^{\boldsymbol{\mathfrak{c}}} = \cosh(\alpha_a(\boldsymbol{\mathfrak{c}}))t_a + i\sinh(\alpha_a(\boldsymbol{\mathfrak{c}}))p_a. \tag{122}$$

Since the roots are real on $\tilde{c}_i$, if we define $\vec{\alpha}_a \equiv (\alpha_a^1, \ldots, \alpha_a^r)$ with $\alpha_a^i = \alpha_a(\tilde{c}_i)$, we obtain $\alpha_a(\boldsymbol{\mathfrak{c}}) = -i\sum_{i=1}^r \alpha_a^i y_i \equiv -i\vec{\alpha}_a \cdot \vec{y}$. Then,

$$\det M = \prod_{a=1}^h \sin(\vec{\alpha}_a \cdot \vec{y}). \tag{123}$$

Thus, the invariant measure takes the form

$$d\mu_{\boldsymbol{G}}[\vec{x};\vec{y};\vec{z}] = d\mu_{\boldsymbol{H}}[\vec{z}]d\mu_{\boldsymbol{H}}[\vec{x}]\prod_{a=1}^h \sin(\vec{\alpha}_a \cdot \vec{y})\prod_{i=1}^r dy_i. \tag{124}$$

The range of the $z$ coordinates is such to cover the subgroup $\boldsymbol{H}$, whereas the range $R_y$ for the $y$ coordinates is defined by the conditions $0 \le \vec{\alpha}_a \cdot \vec{y} \le \pi$, and the range for the $x$ coordinates is such to cover $\boldsymbol{H'}/\boldsymbol{\Gamma}$. In particular, as a consequence of Equation (122), the range for the $y_i$'s depends on the adjoint representation and not on the particular $\boldsymbol{G}$-faithful representation we are considering.

When $\boldsymbol{H}$ is simply connected, it is easy to see that $\boldsymbol{\Gamma} \simeq \mathbb{Z}_2^r$. Indeed, the elements of $\boldsymbol{\Gamma} = \boldsymbol{H} \cap e^{\boldsymbol{\mathfrak{c}}}$ are the elements of $e^{\boldsymbol{\mathfrak{c}}}$ whose square is the identity in $\boldsymbol{H}$ (see [38], section VII, Theorem 8.5). Since the basis $c_1, \ldots, c_r$ of $\boldsymbol{\mathfrak{c}}$ can be chosen so that $e^{tc_i}$ has period $T$, $\boldsymbol{\Gamma}$ is generated by $e^{\frac{T}{2}c_i}$ that proves our claim. In particular, $|\boldsymbol{\Gamma}| = 2^r$.

When $\boldsymbol{H}$ is not simply connected, this is not true in general and $\Gamma$ is isomorphic to a proper subgroup of $\mathbb{Z}_2^r$. Indeed, if $\Phi : \tilde{H} \longrightarrow H$ is the universal covering map, then $\Gamma \simeq \Phi(\mathbb{Z}_2^r)$.

The range of the coordinates $\vec{y}$ can be made more explicit as follows. Choose a fundamental system of simple roots $\alpha_{j_1}, \ldots, \alpha_{j_r}$. We define coordinates $\tilde{y}_i$, $i = 1, \ldots, r$ by

$$\tilde{y}_i := \vec{y} \cdot \vec{\alpha}_{j_i}. \tag{125}$$

The Jacobian of this transformation can be computed by noticing that

$$d\tilde{y}_1 \wedge \ldots \wedge d\tilde{y}_r = V_0 \prod_{i=1}^r \frac{\|\alpha_{j_i}\|^2}{2}dy_1 \wedge \ldots \wedge dy_r, \tag{126}$$

where $V_0$ is the volume of the parallelogram defined by the simple coroots $\check{\alpha}_{j_i}$. The third factor in (124) becomes

$$\prod_{a=1}^{h} \sin(\vec{\alpha}_a \cdot \vec{y}) \prod_{i=1}^{r} dy_i = \frac{2^r}{V_0 \prod_{i=1}^{r} \|\alpha_{a_i}\|^2} \prod_{j=1}^{h} \sin(\vec{n}_j \cdot \vec{y}) \prod_{i=1}^{r} d\tilde{y}_i, \tag{127}$$

where $\vec{n}_j$ are the integer coordinates of the positive roots with regard to the simple roots. For the simple roots, we obtain the relations $0 \leq \tilde{y}_i \leq \pi$, which fix the coordinates in a hypercube. The other conditions $0 \leq \vec{n}_a \cdot \vec{\tilde{y}} \leq \pi$ determine a tiling of the hypercube by the hyperplanes $\vec{n}_j \cdot \vec{\tilde{y}} = k\pi$, with $k$ integer. In particular, there is a unique highest root $\alpha_{\tilde{j}} = \sum_{s=1}^{r} \tilde{n}_s \alpha_{a_s}$, such that $\tilde{n}_s \geq n_j^s$ for all $j$ and $s$. This means that all the inequalities defining the tiling reduce just to the one defined by the highest root:

$$0 \leq \vec{\tilde{n}} \cdot \vec{\tilde{y}} \leq \pi \tag{128}$$

inside the hypercube. This is sufficient since all the pieces of the tiling are equivalent, being related by the Weyl transformations. So, Equation (128) defines the range $\tilde{R}_y$ of the parameters $\vec{\tilde{y}}$ and, therefore, the range $R_y$ for the parameters $\vec{y}$.

*5.3. The Non-Split Case*

Again, we can assume for $G$ to be simple. Since we do not require $H$ to be MCS, in this case we have

$$l := \text{Rank}(G/H) < \text{Rank}(G) = r. \tag{129}$$

This means that the largest possible intersection between a Cartan subalgebra $\mathfrak{c}$ of $\mathfrak{g}$ and the complement of $\mathfrak{h}$ in $\mathfrak{g}$ has dimension $l$. Choosing the Cartan subalgebra $\mathfrak{c}$ with this property, we have

$$\mathfrak{c} = \mathfrak{c}_h \oplus \mathfrak{c}_p, \qquad \mathfrak{c}_p := \mathfrak{c} \cap \mathfrak{p}, \qquad \dim \mathfrak{c}_p = l, \tag{130}$$
$$\mathfrak{c}_h := \mathfrak{c} \cap \mathfrak{h}, \qquad \dim \mathfrak{c}_h = r - l. \tag{131}$$

Let $\mathfrak{k}$ be the Lie algebra of the normaliser $K$ of $\mathfrak{c}_p$ in $H$ be, that is the largest Lie subalgebra of $\mathfrak{h}$ commuting with $\mathfrak{c}_p$, $\dim K = k$. Thus, we can write $\mathfrak{h} =: \mathfrak{k} \oplus \tilde{\mathfrak{h}}$ and $\mathfrak{p} =: \mathfrak{c}_p \oplus \tilde{\mathfrak{p}}$, in such the way that

$$\mathfrak{g} = (\mathfrak{k} \oplus \tilde{\mathfrak{h}}) \oplus (\mathfrak{c}_p \oplus \tilde{\mathfrak{p}}). \tag{132}$$

$\mathfrak{h}$ is symmetrically embedded, and $[\mathfrak{k}, \mathfrak{c}_p] = 0$ implies

$$[\tilde{\mathfrak{h}}, \mathfrak{c}_p] \subseteq \tilde{\mathfrak{p}}, \qquad [\tilde{\mathfrak{p}}, \mathfrak{c}_p] \subseteq \tilde{\mathfrak{h}}. \tag{133}$$

Now, fix a basis $\tau_1, \ldots, \tau_{r-l}$ for $\mathfrak{c}_h$, and $\sigma_1, \ldots, \sigma_l$ for $\mathfrak{c}_p$. We can represent the roots as the simultaneous eigenvalues of the operators

$$(\text{ad}_{\tau_1}, \ldots, \text{ad}_{\tau_{r-l}}; \text{ad}_{\sigma_1}, \ldots, \text{ad}_{\sigma_l}).$$

The eigenvectors of the roots $\alpha_{\mathfrak{h},a}$, $a = 1, \ldots, k - s$, of the subalgebra $\mathfrak{k}$ are in the complexification of $\mathfrak{k}$ and, in particular, in the kernel of $\text{ad}_{\sigma_i}$, $i = 1, \ldots, l$. This means that the last $l$ components are zero. These are all the non-vanishing roots having this property; all the other ones have necessarily at least some non-vanishing elements in the last $l$ components. The corresponding set of roots, say $\alpha_{\mathfrak{p},b}$, $b = 1, \ldots, 2q$, where $q$ is the number of positive roots, is not reduced. This means that each root $\alpha_{\mathfrak{p},b}$ has multiplicity $m_b$, and $\sum_{b=1}^{q} m_b = h - k$. In conclusion, we obtain a decomposition of the restricted root system, $R_{\mathfrak{p}} = R_{\mathfrak{p}}^+ \oplus R_{\mathfrak{p}}^-$, similar to the one of the group the main difference with regard to the case of a MCS subgroup is that now $R_{\mathfrak{p}}$ is not a reduced root lattice system, and

generically each root $\alpha$ is characterised by a multiplicity $m_\alpha \geq 1$. A complete classification is given in [40], see also [38].

Now, let us proceed as in the previous section. First, we fix an orthonormal basis of $\mathfrak{g}$

$$B = \{\tau_1, \ldots, \tau_{r-l}, g_1, \ldots, g_{k-r+l}\} \cup \{t_1, \ldots, t_{h-k}\} \cup \{h_1, \ldots, h_l\} \cup \{p_1, \ldots, p_{h-k}\},$$

where $\{\tau_1, \ldots, \tau_s, g_1, \ldots, g_{k-s}\}$ is an orthonormal basis for $\mathfrak{k}$, the $t_a$ generate $\tilde{\mathfrak{h}}$, and the $p_b$ generate $\tilde{\mathfrak{p}}$. The generalised Euler parameterisation for $G$ is

$$G[\vec{x}; \vec{y}; \vec{z}] = e^{\sum_{a=1}^{h-k} x^a t_a} e^{\sum_{i=1}^{l} y^i h_i} H[z_1, \ldots, z_h], \tag{134}$$

where $H$ can in turn be parametrised in the same way, but it is irrelevant here. The range of the $z$ coordinates must be chosen to cover exactly once the whole subgroup $H$. As before, we can compute the invariant measure that comes out to be

$$d\mu_G[\vec{x}; \vec{y}; \vec{z}] = d\mu_H[\vec{z}] \, d\mu_{H/K}[\vec{x}] \prod_{a=1}^{q} \sin^{m_a}(\vec{\alpha}_{\mathfrak{p},a} \cdot \vec{y}) \prod_{i=1}^{l} dy_i, \tag{135}$$

where $\vec{\alpha}_{\mathfrak{p},a} := (\alpha^1_{\mathfrak{p},a}, \ldots, \alpha^l_{\mathfrak{p},a})$, $a = 1, \ldots, q$ are the last $l$ components of the positive $\alpha_{\mathfrak{p}a}$, corresponding to the eigenvalues of the $\mathrm{ad}_{h_i}$ only. After choosing a basis of $l$ simple roots, $\vec{\alpha}_1, \ldots, \vec{\alpha}_l$ in $R^+_{\mathfrak{p}}$, it follows that the range for the coordinates $\vec{y}$ is given by

$$0 \leq \vec{\alpha}_i \cdot \vec{y} \leq \pi, \qquad 0 \leq \sum_{i=1}^{l} n_i \vec{\alpha}_i \cdot \vec{y} \leq \pi, \tag{136}$$

where $\sum_{i=1}^{l} n_i \vec{\alpha}_i$ is the longest reduced root.

## 6. Euler versus Dyson

Now we concentrate on the geometrisation of the Dyson integrals.

### 6.1. The Split Integrals

Inspired by (127), let us consider the integral

$$I = \int_{\tilde{R}_y} \prod_{j=1}^{h} \sin(\vec{n}_j \cdot \vec{y}) \prod_{i=1}^{r} d\tilde{y}_i. \tag{137}$$

Since all pieces of the tiling are equivalent, we can rewrite it in terms of an integral over the whole hypercube $C_\pi = \{\vec{y} | 0 < \tilde{y}_i < \pi, i = 1, \ldots, r\}$. Since

$$\int_{\tilde{R}_y} \prod_{i=1}^{r} d\tilde{y}_i = \frac{\pi^r}{r! \prod_{i=1}^{r} \tilde{n}_i}, \tag{138}$$

we see that the hypercube is divided in $r! \prod_{i=1}^{r} \tilde{n}_i$ parts, and we can write

$$I = \frac{1}{r! \prod_{i=1}^{r} \tilde{n}_i} \int_{C_\pi} \prod_{j=1}^{h} |\sin(\vec{n}_j \cdot \vec{y})| \prod_{i=1}^{r} d\tilde{y}_i. \tag{139}$$

If we introduce the change of variables $2\vec{y} = \vec{\gamma}$, we finally obtain

$$I = \frac{1}{2^{h-r} r! \prod_{i=1}^{r} \tilde{n}_i} \int_{C_{2\pi}} \prod_{\alpha \in R} (1 - e^{\vec{n}_\alpha \cdot \vec{\gamma}})^{\frac{1}{2}} \prod_{i=1}^{r} d\gamma_i. \tag{140}$$

The integral

$$
J_{\frac{1}{2}} := \frac{1}{(2\pi)^r} \int_{C_{2\pi}} \prod_{\alpha \in R} (1 - e^{\vec{n}_\alpha \cdot \vec{\gamma}})^{\frac{1}{2}} \prod_{i=1}^{r} d\gamma_i, \tag{141}
$$

is a particular case of the Macdonald integrals appearing in [33], Conjecture 1, which we report here for simplicity, for any root lattice $R$:

**Conjecture 1.** *For all $s \in \mathbb{C}$ with $\mathrm{Re}(s) > 0$,*

$$
J_s = \frac{1}{(2\pi)^r} \int_{C_{2\pi}} \prod_{\alpha \in R} (1 - e^{\vec{n}_\alpha \cdot \vec{\gamma}})^s \prod_{i=1}^{r} d\gamma_i, = \prod_{i=1}^{r} \frac{\Gamma(sd_i + 1)}{\Gamma(s+1)\Gamma(sd_i - s + 1)}. \tag{142}
$$

Actually, it is a particular case of Theorem 1.
From (140) we can write

$$
J_{\frac{1}{2}} = \frac{2^{h-r} r! \prod_{i=1}^{r} \tilde{n}_i}{(2\pi)^r} I. \tag{143}
$$

On the other hand, (124) implies

$$
\int_{R_y} \prod_{a=1}^{h} \sin(\vec{\alpha}_a \cdot \vec{y}) \prod_{i=1}^{r} dy_i = \frac{\mathrm{Vol}(\boldsymbol{G}) \, |\boldsymbol{\Gamma}|}{\mathrm{Vol}(\boldsymbol{H})^2}, \tag{144}
$$

where $|\boldsymbol{\Gamma}|$ is the cardinality of $\boldsymbol{\Gamma}$. Using (126), we see that the left-hand side of (144) is just

$$
\frac{2^r}{V_0 \prod_{i=1}^{r} \|\alpha_{a_i}\|^2} \, I. \tag{145}
$$

Combining all things, we finally obtain

$$
J_{\frac{1}{2}} = \frac{2^h V_0 r! \prod_{i=1}^{r} (n_i \|\alpha_{j_i}\|^2)}{\pi^r} \, \frac{\mathrm{Vol}(\boldsymbol{G})}{\mathrm{Vol}(\boldsymbol{H})^2} \, \frac{|\boldsymbol{\Gamma}|}{2^r}, \tag{146}
$$

where the volumes can be computed by means of Macdonald's formula [18]. This formula proves Conjecture 1 for $s = \frac{1}{2}$ and for all the reduced simple lattices. For these cases, we thus have a geometric interpretation of the Dyson integrals.

*6.2. The Non-Split Integrals*

We can obtain more general results by considering the non-split case of the Euler parametrisation. Define

$$
J^{\mathfrak{p}}_{\{k_\alpha\}} := \frac{1}{(2\pi)^r} \int_{C_{2\pi}} \prod_{\alpha \in R_{\mathfrak{p}}} (1 - e^{\vec{n}_\alpha \cdot \vec{\gamma}})^{k_\alpha} \prod_{i=1}^{r} d\gamma_i, \tag{147}
$$

with $k_\alpha$ as in Theorem 1. Its value is indeed provided by Theorem 1.

We can repeat exactly the same reasoning as above, but referring to Section 5.3 to obtain the formula

$$
J^{\mathfrak{p}}_{\{\frac{m_\alpha}{2}\}} = \frac{2^{h-k} |\vec{\alpha}_1 \wedge \ldots \wedge \vec{\alpha}_l| \, l! \, \prod_{i=1}^{l} n_i}{\pi^l} \, \frac{\mathrm{Vol}(\boldsymbol{G})\mathrm{Vol}(\boldsymbol{K})}{\mathrm{Vol}(\boldsymbol{H})^2}, \tag{148}
$$

where $m_\alpha$ are the multiplicities of the roots in the lattice correspondent to the associated symmetric space. Compared with Theorem 1, with the invariant functions $k_\alpha = m_\alpha/2$, this expression provides the right value for the generalised Dyson integrals $J^{\mathfrak{p}}_{\{\frac{m_\alpha}{2}\}}$. This proves

Macdonald's conjecture ([33], conjecture 2.3) for $k_\alpha = \frac{m_\alpha}{2}$ and for the lattices associated with all the irreducible symmetric spaces.

The ingredients to verify the formulas are given in [1], Table 1.

## 7. Open Questions and Further Applications

Here, we describe a non-symmetric embedding [5].

### 7.1. Non-Symmetric Embeddings

The results presented up to now are based on the choice of a proper maximally symmetrical embedded subgroup of a compact simple Lie group. However, given a connected compact Lie group $\boldsymbol{G}$, there are several maximal proper subgroups that are non symmetrically embedded in $\boldsymbol{G}$. We want now to show a very explicit example based on [5].

#### 7.1.1. The Group $\boldsymbol{G}_2$ and Its Lie Algebra

Consider the oriented Fano triangle (Figure 4).

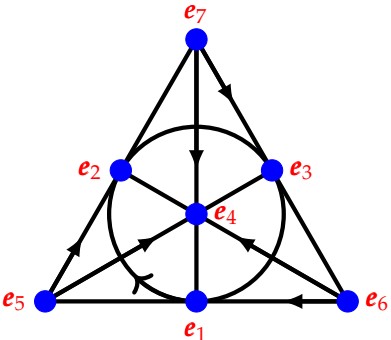

**Figure 4.** The Fano triangle.

It is composed of seven oriented lines: the three sides of the triangle, the three heights, and the circle. Each line must be thought of as if it were an oriented circle. The side lines are oriented so that the boundary of the triangle looks clockwise oriented. The heights are oriented from the vertex to the base. The circle is oriented clockwise. The triangle defines the octononic algebra. Octonions are the vectors of the eight-dimensional real vector space generated by the basis $\boldsymbol{e}_\mu$, $\mu = 0, \ldots, 7$. They form a non-associative algebra with real unit $\boldsymbol{e}_0 \equiv 1$ and seven imaginary units $\boldsymbol{e}_j$, $j = 1, \ldots, 7$ satisfying $\boldsymbol{e}_i \boldsymbol{e}_j + \boldsymbol{e}_j \boldsymbol{e}_i = -2 \boldsymbol{e}_0 \delta_{ij}$ and the products defined by the $\mathbb{R}$-bilinear extension of the multiplication rules among imaginary units, given by the Fano triangle as follows: if $i \neq j$, $\boldsymbol{e}_i \boldsymbol{e}_j = \pm \boldsymbol{e}_k$ where $\boldsymbol{e}_k$ is the third element on the line selected by the pair $\boldsymbol{e}_i, \boldsymbol{e}_j$. The sign is positive if the order $ijk$ is concordant with the orientation of the circle corresponding to the selected line and negative otherwise. For example, $\boldsymbol{e}_2 \boldsymbol{e}_7 = \boldsymbol{e}_5$ and $\boldsymbol{e}_2 \boldsymbol{e}_4 = -\boldsymbol{e}_6$. The space $\boldsymbol{O}$ of real octonions is endowed with an $\mathbb{R}$-linear conjugation $\boldsymbol{o} \mapsto \bar{\boldsymbol{o}}$ that changes sign to the imaginary components. It satisfies $\overline{\boldsymbol{o}\boldsymbol{o}'} = \bar{\boldsymbol{o}}'\bar{\boldsymbol{o}}$. This induces a Euclidean scalar product $(\,|\,) : \mathbb{O} \times \mathbb{O} \to \mathbb{R}$ defined by $(\boldsymbol{o}|\boldsymbol{o}') = \frac{1}{2}(\bar{\boldsymbol{o}}\boldsymbol{o}' + \bar{\boldsymbol{o}}'\boldsymbol{o})$. With respect to it, the basis of units is orthonormal.

**Definition 9.** *The group of automorphisms of $\mathbb{O}$ is called* $\boldsymbol{G}_2$.

We want to prove that $\boldsymbol{G}_2$ is a 14-dimensional manifold that, indeed, is an $\boldsymbol{SU}(3)$ fibration over $S^6$. To this aim, we first describe octonions as follows. Let $\vec{a}_R, \vec{a}_I \in \mathbb{R}^3$ and set $\vec{\boldsymbol{e}} \equiv (\boldsymbol{e}_1, \boldsymbol{e}_2, \boldsymbol{e}_3) \in \mathbb{O}^3$. For any $\vec{v} \in \mathbb{K}^3$, we will write $\vec{v} \cdot \vec{\boldsymbol{e}} := v_1 \boldsymbol{e}_1 + v_2 \boldsymbol{e}_2 + v_3 \boldsymbol{e}_3$, $\mathbb{K} = \mathbb{R}, \mathbb{O}$. Moreover, for $c \in \mathbb{K}$, we define $c\vec{v} = (cv_1, cv_2, cv_3)$ and $\vec{v}c = (v_1 c, v_2 c, v_3 c)$. Finally, we will identify $\boldsymbol{e}_0 \equiv 1$.

With this notation, any octonion can be written as

$$\boldsymbol{a} = \alpha_0 + \vec{\alpha} \cdot \vec{\boldsymbol{e}}, \tag{149}$$

where

$$\alpha_0 = a_0 + a_4 \boldsymbol{e}_4, \quad \vec{\alpha} = \vec{a}_R + \vec{a}_I \boldsymbol{e}_4, \quad a_0, a_4 \in \mathbb{R}, \vec{a}_R, \vec{a}_I \in \mathbb{R}^3. \tag{150}$$

We will consider the identification $\mathbb{C} \simeq \mathbb{R} \oplus \mathbb{R}\boldsymbol{e}_4$. We notice that with this identification $\mathbb{C}$-numbers do not commute with octonions. If we also define the $\mathbb{C}$-scalar and $\mathbb{C}$-vector products in $\mathbb{C}^3$ by

$$(\vec{\alpha}|\vec{\beta})_{\boldsymbol{C}} := \vec{a}_R \cdot \vec{b}_R + \vec{a}_I \cdot \vec{b}_I - \vec{a}_I \cdot \vec{b}_R \boldsymbol{e}_4 + \vec{a}_R \cdot \vec{b}_I \boldsymbol{e}_4, \tag{151}$$

$$(\vec{\alpha} \times \vec{\beta})_{\boldsymbol{C}} := \vec{a}_R \times \vec{b}_R - \vec{a}_I \times \vec{b}_I + \vec{a}_I \times \vec{b}_R \boldsymbol{e}_4 + \vec{a}_R \times \vec{b}_I \boldsymbol{e}_4, \tag{152}$$

where $\times$ is the usual vector product in $\mathbb{R}^3$, then we can write the product of two octonions as

$$\boldsymbol{ab} = \alpha_0 \beta_0 + \alpha_0 \vec{\beta} \cdot \boldsymbol{e} + \beta_0 \vec{\alpha} \cdot \boldsymbol{e} - (\vec{\alpha}|\vec{\beta})_{\mathbb{C}} + (\vec{\alpha} \times \vec{\beta})_{\boldsymbol{C}} \cdot \boldsymbol{e}. \tag{153}$$

Let $\Phi \in \boldsymbol{SU}(3)$ a special unitary transformation on $\mathbb{C}^3$. It extends to an action on octonions by

$$\Phi(\boldsymbol{a}) := \alpha_0 + \Phi^\dagger(\vec{\alpha}) \cdot \boldsymbol{e}, \tag{154}$$

where $\Phi^\dagger$ is the adjoint of $\Phi$. It follows from (153) that $\Phi(\boldsymbol{ab}) = \Phi(\boldsymbol{a})\Phi(\boldsymbol{b})$, it is sufficient to notice that, by definition of unitary map, $(\Phi(\vec{\alpha})|\Phi(\vec{\beta}))_{\mathbb{C}} = (\vec{\alpha}|\vec{\beta})_{\mathbb{C}}$, and since $\det \Phi = 1$, $(\Phi(\vec{\alpha}) \times \Phi(\vec{\beta}))_{\mathbb{C}} = \Phi((\vec{\alpha} \times \vec{\beta})_{\mathbb{C}})$. This means that $\Phi$ is an automorphism of $\mathbb{O}$ so that $\boldsymbol{SU}(3) \subset \boldsymbol{G}_2$. It is vice versa, if $\Phi$ is an automorphism that fixes $\boldsymbol{e}_4$. It follows from (153) that it must have the above form and, therefore, it is in $\boldsymbol{SU}(3)$.

Let us consider a generic imaginary unit, that is an element $\boldsymbol{\varepsilon} \in \mathbb{O}$ satisfying $\varepsilon^2 = -1$. A simple calculation shows that

$$\boldsymbol{\varepsilon} = \sum_{j=1}^{7} \varepsilon_j \boldsymbol{e}_j, \tag{155}$$

with

$$\underline{\varepsilon} \equiv (\varepsilon_1, \dots, \varepsilon_7) \in S^6 = \{ \underline{x} \in \mathbb{R}^7 | \sum_{j=1}^{7} x_j^2 = 1 \}. \tag{156}$$

Therefore, if $[\,,]$ is the commutator in $\mathbb{O}$, we have[7]

$$[\boldsymbol{e}_4 \boldsymbol{\varepsilon}, \boldsymbol{e}_4] = 2\boldsymbol{\varepsilon}, \tag{157}$$

which is equivalent to

$$Ad_{\exp(\frac{x}{2}\boldsymbol{e}_4\boldsymbol{\varepsilon})}(\boldsymbol{e}_4) = \cos x \, \boldsymbol{e}_4 + \sin x \, \boldsymbol{\varepsilon}. \tag{158}$$

Now, suppose $\Psi \in \boldsymbol{G}_2$. It follows that $\Psi(\boldsymbol{e}_4)^2 = \Psi(\boldsymbol{e}_4^2) = -1$, so that $\boldsymbol{\varepsilon} = \Psi(\boldsymbol{e}_4)$ is an imaginary unit. Therefore,

$$Ad_{\exp(-\frac{\pi}{4}\boldsymbol{e}_4\boldsymbol{\varepsilon})} \circ \Psi(\boldsymbol{e}_4) = \boldsymbol{e}_4 \tag{159}$$

so that $Ad_{\exp(-\frac{\pi}{4}\boldsymbol{e}_4\boldsymbol{\varepsilon})} \circ \Psi \in \boldsymbol{SU}(3)$. This implies that

$$\Psi = Ad_{\exp(\frac{\pi}{4}\boldsymbol{e}_4\boldsymbol{\varepsilon})} \circ \Phi \tag{160}$$

with $\Phi \in \boldsymbol{SU}(3)$. Thus, the elements of $\boldsymbol{G}_2$ are parametrised by $\boldsymbol{SU}(3)$ and by $\underline{\varepsilon} \in S^6$. Since $\dim(\boldsymbol{SU}(3)) = 8$, it follows that $\dim(\boldsymbol{G}_2) = 14$ and we have proved:

**Proposition 6.** $G_2$ *is an* $SU(3)$ *fibration over* $S^6$.

We are now ready to compute the Lie algebra. Its elements are derivations acting on the seven-dimensional space generated over $\mathbb{R}$ by the imaginary units. Indeed, if $\Psi_t$ is a one parameter family of automorphisms of $\mathbb{O}$ such that $\Psi_0 = id$ (the identity), then the corresponding Lie algebra element is

$$D = \frac{d}{dt}\bigg|_{t=0} \Psi_t, \tag{161}$$

and from

$$\Psi(\boldsymbol{o}_1\boldsymbol{o}_2) = \Psi(\boldsymbol{o}_1)\Psi(\boldsymbol{o}_2) \tag{162}$$

it follows

$$D(\boldsymbol{o}_1\boldsymbol{o}_2) = D(\boldsymbol{o}_1)\boldsymbol{o}_2 + \boldsymbol{o}_1 D(\boldsymbol{o}_2). \tag{163}$$

In particular, $D(1) = 0$ so we obtain a seven-dimensional representation.
A basis for the algebra is given by the eight generators of $Lie(SU(3))$, plus six non-trivial generators of the form $ad_{\boldsymbol{e}_4\boldsymbol{\varepsilon}}$, that we can fix choosing $\boldsymbol{\varepsilon} \equiv \boldsymbol{e}_j$, $j \neq 4$. For the Lie algebra of $SU(3)$, we choose the Gell-Mann matrices given in (4). Each of them acts on the elements $\vec{v} = \vec{x} + \boldsymbol{e}_4\vec{y} \in \mathbb{C}^3$ in the obvious way, with $i$ replaced by $\boldsymbol{e}_4$. In order to obtain the seven-dimensional action, we have to consider the embedding $\mathbb{C}^3 \hookrightarrow \mathbb{R}^7$ given by

$$\vec{x} + \boldsymbol{e}_4\vec{y} \longmapsto (x_1, x_2, x_3, 0, y_1, y_2, y_3). \tag{164}$$

This gives us the $7 \times 7$ matrices $\Lambda_1, \ldots, \Lambda_8$. The remaining are obtained by considering the adjoint actions $\sigma_i : \boldsymbol{e}_j \mapsto [\boldsymbol{e}_4\boldsymbol{e}_i, \boldsymbol{e}_j]$, $i \neq 4$, $j = 1, \ldots, 7$, and then normalising the matrices so that $\text{Tr}(\Lambda_a\Lambda_b) = -4\delta_{ab}$, $a, b = 1, \ldots, 14$. We suggest the reader compute the simple roots, starting from the matrices in Appendix B, for example, by choosing the Cartan subalgebra generated by the matrices $\Lambda_2$ and $\Lambda_{12}$ and verify that the corresponding Dynkin diagram is exactly the one of $G_2$ given in Appendix A.

Let $\mathfrak{g}$, $\mathfrak{k}$, and $\mathfrak{p}$, respectively, be the Lie algebra of $G_2$, of $SU(3)$ and its complement. Thus, $\mathfrak{k}$ is spanned by $\Phi_{\lambda_i} \equiv \Lambda_i$, $i = 1, \ldots, 8$, while $\mathfrak{p}$ is spanned by $\Psi_{\boldsymbol{e}_4\boldsymbol{e}_j} \equiv \Lambda_{f(j)}$, where $j = 1, 2, 3, 5, 6, 7$ and $f(j) = 9, 10, 11, 12, 13, 14$, respectively. We now show that $SU(3)$ is maximal but not symmetrically embedded. First, notice that obviously the algebra satisfies the relations

$$[\mathfrak{k}, \mathfrak{k}] \subseteq \mathfrak{k}, \tag{165}$$

$$[\mathfrak{k}, \mathfrak{p}] \subseteq \mathfrak{p}. \tag{166}$$

The first relation is obvious; the second one follows from the ad-invariance of the Killing form and the orthogonality of the decomposition. The second relations mean that $\mathfrak{p}$ is a representation space for $SU(3)$. As a representation of Lie, $(SU(3))$ we have

$$\mathfrak{p} = \mathbf{3} \oplus \bar{\mathbf{3}}. \tag{167}$$

To prove it, it is sufficient to compute the $6 \times 6$ matrices representing the adjoint action of $\Lambda_3$ and $\Lambda_8$ on $\mathfrak{p}$ and to diagonalise them simultaneously in order to compute the weights. We leave this as an exercise for the reader. It results that $\mathbf{3}$ is generated by $i\Lambda_9 + \Lambda_{14}$, $i\Lambda_{10} + \Lambda_{13}$, $i\Lambda_{11} + \Lambda_{12}$, and $\bar{\mathbf{3}}$ by the complex conjugates. Since these are the only two possible invariant subspaces and are not real, it follows that any proper real subalgebra of $\mathfrak{g}$ containing $\mathfrak{k}$ cannot exist. This shows maximality.

Now, we prove that

$$[\mathfrak{p}, \mathfrak{p}] \not\subseteq \mathfrak{k}. \tag{168}$$

To this end, notice that $\mathfrak{p}$ is generated by the maps $\sigma_i$ defined above, $i \neq 4$. If $[\sigma_1, \sigma_2] \in \mathfrak{k}$, then we would have $[\sigma_1, \sigma_2](\boldsymbol{e}_4) = 0$. However, since $\sigma_i(\boldsymbol{e}_4) = 2\boldsymbol{e}_i$, and $\boldsymbol{e}_4\boldsymbol{e}_1 = \boldsymbol{e}_7$, $\boldsymbol{4}_4\boldsymbol{e}_2 = \boldsymbol{e}_6$, we obtain

$$[\sigma_1, \sigma_2](\boldsymbol{e}_4) = 2(\sigma_1(\boldsymbol{e}_2) - \sigma_2(\boldsymbol{e}_1)) = 2([\boldsymbol{e}_7, \boldsymbol{e}_2] - [\boldsymbol{e}_6, \boldsymbol{e}_1]) = -2\boldsymbol{e}_5. \tag{169}$$

In conclusion, we have shown that $\boldsymbol{SU}(3)$ is maximal in $\boldsymbol{G}_2$ but not symmetrically embedded, and $\boldsymbol{G}_2/\boldsymbol{SU}(3) \simeq S^6$.

7.1.2. A Euler Parametrisation of $\boldsymbol{G}_2$

Now, we show that despite the non-symmetric immersion, we can construct the Euler angles with respect to $\boldsymbol{SU}(3)$. To this aim, we will follow [5]. We know that we can parametrise $\boldsymbol{SU}(3)$ with respect tu $\boldsymbol{U}(2)$ by writing it in the form

$$\boldsymbol{SU}(3)[\underline{x}; y; \vec{z}] = \boldsymbol{U}(2)[\underline{x}]e^{y\Lambda_4}\boldsymbol{SU}(2)[\vec{z}], \tag{170}$$

where $\underline{x} = (x_1, \ldots, x_4)$, $\vec{z} = (z_1, z_2, z_3)$, $\boldsymbol{SU}(2)$ is generated by $\Lambda_1, \Lambda_2, \Lambda_3$, and $\boldsymbol{U}(2) = \boldsymbol{SU}(2) \times \boldsymbol{U}(1)/\mathbb{Z}_2$, with $\boldsymbol{U}(1)$ generated by $\Lambda_8$. The strategy is to find an element of $\mathfrak{p}$ that commutes with the $\boldsymbol{SU}(2)$ and whose commutators with the remaining matrices of $\boldsymbol{SU}(3)$ generate a basis for the whole $\mathfrak{p}$. Looking at Appendix B, we see immediately that $\Lambda_{12}$ commutes with $\Lambda_1, \Lambda_2$, and $\Lambda_3$ and, indeed, it does the job.
Therefore, our ansatz is that $\boldsymbol{G}_2$ can be parametrised as

$$\boldsymbol{G}_2 = \boldsymbol{U}(2)[\underline{x}]e^{y\Lambda_4}e^{w\Lambda_{12}}\boldsymbol{SU}(3)[\tilde{x}; \tilde{y}; \vec{\tilde{z}}]. \tag{171}$$

In order to prove it, the last step consists of proving that with a suitable choice of the parameters $\underline{x}$, $y$, and $w$, the quotient $\boldsymbol{G}_2/\boldsymbol{SU}(3)$, parametrised by $\boldsymbol{U}(2)[\underline{x}]e^{y\Lambda_4}e^{w\Lambda_{12}}$, cover the sphere $S^6$ exactly once.
Let

$$\Sigma = \boldsymbol{U}(2)[\underline{x}]e^{y\Lambda_4}e^{w\Lambda_{12}}. \tag{172}$$

The metric on the quotient $\boldsymbol{G}_2/\boldsymbol{SU}(3)$ can be computed from $J_\sigma := \Sigma^{-1}d\Sigma$, projecting out the components tangent to $\boldsymbol{SU}(3)$. This gives the "reduced current" $J_\Sigma^{\parallel}$, from which we obtain the metric on the quotient

$$d\sigma^2 = -\frac{1}{4}\mathrm{Tr}J_\Sigma^{\parallel} \otimes J_\Sigma^{\parallel}. \tag{173}$$

The factor $-\frac{1}{4}$ is just to compensate the normalisation of the matrices $\Lambda_a$. The remaining details are just a direct and quite tedious calculation. As there is nothing to learn, we will not report the details of the calculations here, which can be found in [5].

7.2. *Open Questions*

The above considerations suggest some interesting questions.
First, we have seen that it is possible to construct generalised Euler angles even starting from a non-symmetric embedding. It seems that the condition $[\mathfrak{k}, \mathfrak{p}] \subseteq \mathfrak{p}$ is sufficient. However, it must be noted that such a rule is quite general, also for non-maximal subgroups. So, we wonder if maximality plays a role. It is important to note that maximal proper subgroups, symmetrically embedded or not, can be computed following the strategies developed in [16]. There, for example, all $\boldsymbol{SU}(2)$ subgroups are classified, and, in particular, every simple compact group contains a maximal $\boldsymbol{SU}(2)$ subgroup. It is not easy to

imagine how a generalised Euler parametrisation with respect to $SU(2)$ could work for simple groups of high dimension. Nevertheless, it would be interesting to investigate the possibility to define generalised Euler parametrisations of a given group $G$ with regard to all possible maximal proper subgroups, including the smallest ones.

Related to the previous question, there is the second one: does the Euler parametrisation with respect to a non-symmetrically embedded maximal subgroup originate a more general kind of Dyson integrals? We have seen that the symmetric Euler parametrisation is related to the Dyson integrals associated with the lattices corresponding to the compact symmetric spaces. For the non-symmetrical case, we do not have naturally associated lattices, an, therefore, we have not any hint to understand how to associate Dyson integrals. The condition of symmetry of the embedding seems to be crucial for better understanding the geometry underlying the invariant measures. It would be interesting to deepen our understanding about the geometry of all maximal embeddings.

Of course, it has not been possible to investigate such questions with the simplest example of $SU(3)$ since its only proper maximal subgroups are $SU(2)$ and $SO(3)$, which are both symmetrically embedded. It necessarily requires work with examples of higher dimensions, such as $G_2$, $SU(4)$, and $SO(5)$. Working out these examples very explicitly could suggest general rules sufficient to understand the general case.

*7.3. Applications*

There are several obvious applications of generalised Euler parametrisation of groups; the original motivation of the general program was for numerical applications in Lattice Gauge Theories and other numerical simulations in non-perturbative constructions. However, here we want to briefly illustrate a couple of applications that are less standard and in a sense unexpected a priori.

The Problem of Measure Concentration

The phenomenon of concentration of measure is well explained by Levy in [41], where it is shown that if we consider a family of spheres in $\mathbb{R}^{n+1}$ of fixed radius, parametrised by the dimension $n$ and endowed with the Lebesgue measure, then, with the increasing of the dimensions, the measure concentrates along the equator. This can be understood intuitively as follows. In $\mathbb{R}^k$, consider a filled ball of radius $R$. Its Lebesgue volume is $V = KR^k$ where $K$ is a constant, depending on $k$, but irrelevant here. If we now take a crust of thickness $dR$ on the surface of the ball, its volume will be approximatively $dV = KkR^{k-1} dr$. Therefore,

$$\frac{dV}{V} \approx k\frac{dR}{R}. \tag{174}$$

If we keep the radius $R$ fixed and the thickness $dR$, independently of how small $dR$ is, for $k$ large enough, the quotient on the right-hand side stops to be infinitesimal, and the approximation is no longer valid. In other words, it will appear that the whole volume concentrates on the crust. If we imagine to be in the center of the ball, it will appear to us that all volume is uniformly smeared in the farthest region around us. After a short meditation, this is not so much surprising; the measure is uniform and distributed in a larger number of angular directions as the dimensions increase. Now, if in place of being in $\mathbb{R}^k$ we are in $S^k$, in a sense, the farthest region from us is the equator with regard to which we are in a pole, in the sense that the volume of the boundary of a ball essentially depends on the minimum between the radius of the ball and the radius of its complement.

This phenomenon of concentration of the measure is important because it is also related to the existence of fixed points under the action of infinite dimensional groups on a manifold (or a set), see [42]. Indeed, already in the above example we can understand the measure on the spheres as induced by the action of the rotation groups $SO(k+1)$ and their invariant measure.

These facts suggest that indeed one can try to understand infinite dimensional properties of measure and geometry by studying them at arbitrary finite dimension $n$, for example,

for classical series of Lie groups and their quotients, see [43]. In this regard, the generalised Euler parametrisation allows us to construct both the invariant metric and the corresponding invariant measure for the Lie group explicitly, but also for the quotient spaces thereof, in terms of simple trigonometric functions. The easiest constructions one can do are the symmetric spaces associated with the classical Lie series. These have been studied in [43], showing that one can extend the elementary direct calculations of Levy in [41] to all sequences of symmetric spaces. For example, if one considers $SU(n+1)/U(n) = \mathbb{CP}^n$, what happens is that the invariant[8] measure on $\mathbb{CP}^n$ concentrates on the hyperplane at infinity, the "analogous" of the equator for the spheres, but now of real co-dimension two. It would be interesting to extend these results to non-symmetric quotients.

*7.4. Applications to Nuclear Physics.*

In principle, nuclear physics is expected to be deducible as a low energy limit of quantum chromodynamics (QCD). However, at the nuclear energies QCD is very far from the perturbative regime, and actually there are not deductions of nuclear phenomena directly from first principles but just by means of effective models (with the exclusion of few effects). A good effective model, initially for describing mesons, was introduced in the 1950s by Skyrme [44–47]. Surprisingly, this model effectively described not only mesons but also baryons, despite not containing fermions a priori.

In $(3+1)$ dimensions, the action of the Skyrme model with group $G$ is

$$S[U] = \int d^4x \sqrt{-g} \left[ \frac{K}{4} \text{Tr} \left( L_\mu L^\mu + \frac{\lambda}{8} R_{\mu\nu} R^{\mu\nu} \right) \right], \tag{175}$$

$$L_\mu = U^{-1} \partial_\mu U, \qquad R_{\mu\nu} = [L_\mu, L_\nu], \qquad U(x) \in G,$$

where $K$ and $\lambda$ are positive coupling constants, and $g$ is the determinant of the spacetime metric. The equations of motion are obtained by variation of the Skyrme field, which is a map

$$U : \mathbb{R}^{1,3} \longrightarrow G.$$

We assume that $G$ is a compact semi-simple Lie group. After choosing a basis $\{T_i\}$ for the Lie algebra $\mathfrak{g} = Lie(G)$, we can write

$$L_\mu = \sum_{i=1}^{dim(G)} L_\mu^i T_i.$$

Despite the Lagrangian and the equations of motion appearing to be quite simple, they resisted analytic approaches and numerical studies to obtain information for several years. However, for several reasons it is of a certain relevance to also determine some explicit analytical properties of the solutions. Only recently, the first achievements in this direction have been completed for $G = SU(2)$, allowing for determining several interesting properties of the matter described by Skyrmions with finite energy density in a finite volume and with non-spherical symmetries, see e.g., [48–54]. Notice that the difficulty was not to find explicit solutions, but to find explicit solutions having a non-trivial topological charge

$$B = \frac{1}{24\pi^2} \int_{\mathcal{V}} \text{Tr}(\mathcal{L} \wedge \mathcal{L} \wedge \mathcal{L}), \tag{176}$$

where $\mathcal{V}$ is the spatial region occupied by the Skyrme field at any fixed time $t$, $\mathcal{L} = U^{-1} dU$, and Tr is the trace over the matrix indices. This quantity is always an integer and represents the baryon number associated with the solution. Therefore, non-zero $B$ corresponds to solutions containing $B$ baryons. A deeper analysis showed that the key strategy for allowing the determination of explicit solutions with baryons was a suitable parametrisation

of $U$, together with guessing a good ansatz for the space–time dependence. After understanding this, it was quite natural to employ generalised Euler angles to try finding new solutions with the Skyrme field taking value in an arbitrary compact simple Lie group. This program has been successfully applied recently, see [6,7,55]. Interestingly, what resulted is that assuming the generalised Euler parametrisation with a simple linear dependence of the angles leads to solutions that suitably describe lasagna phases of nuclear matter. Assuming instead the exponential parametrisation, and keeping for the exponent a linear dependence on the space–time coordinates, gives solutions that describe spaghetti states. The knowledge of the explicit analytic expression of the solution allowed us to compute different physical properties of them, including the calculation of the shear modulus for the lasagna states and a phase transition between spaghetti and lasagna states. Much more interesting analysis using this strategy are matters of current research.

**Author Contributions:** Both the authors equally contributed to all aspects of the work. Only in the writing and reading of the very final version the second author has been unable to participate due to force majeure. All authors have read and agreed to the published version of the manuscript.

**Funding:** This research received no external funding.

**Institutional Review Board Statement:** Not applicable.

**Informed Consent Statement:** Not applicable.

**Data Availability Statement:** Not applicable.

**Acknowledgments:** The authors would like to thank F. Dalla Piazza and B. L. Cerchiai for several collaborations on Lie groups constructions, as well as F. Bernardoni, G. Ortenzi and A. Dalla Vedova. S.L.C. also thanks F. Canfora for introducing him to the Skyrme equation and showing several important direct applications of the generalise Euler constructions to concrete physical questions, and P. Ursino for showing the relevance of our construction in understanding also problem in infinite dimensional Lie groups theory. Finally, S.L.C. thanks the second author A.S. for the long collaboration, the friendship and for having taught him the meaning of doing science.

**Conflicts of Interest:** The authors declare no conflict of interest.

## Appendix A. Simple Lie Algebras

Complex simple Lie algebras are classified by Dynkin diagrams. There are four classical series

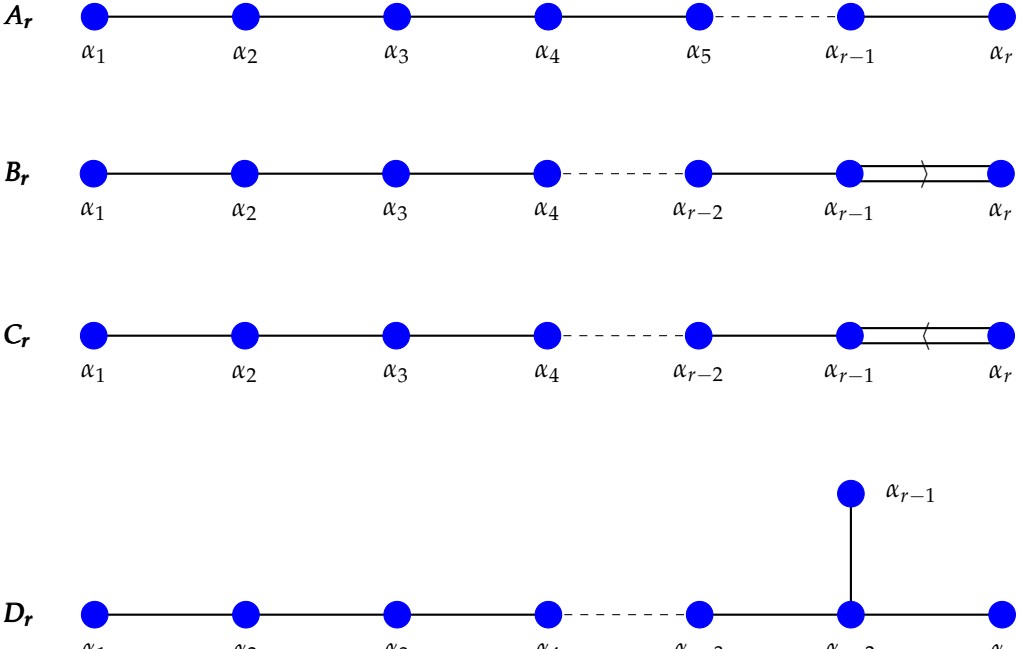

and five exceptional cases

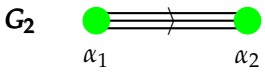

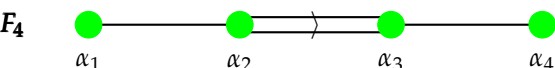

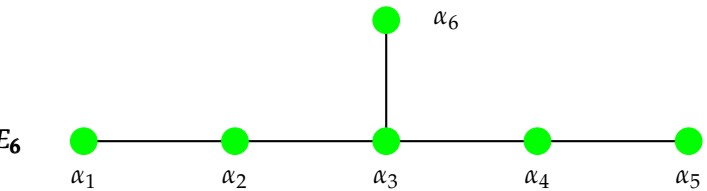

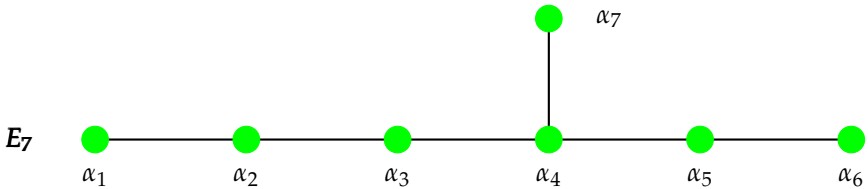

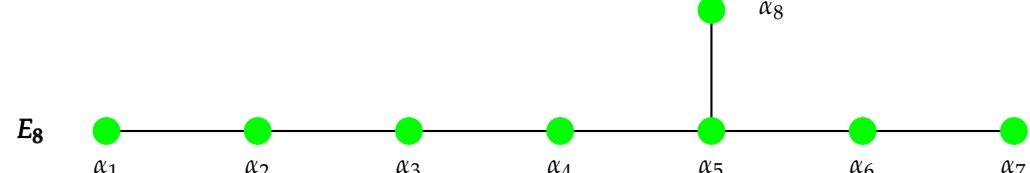

## Appendix B. Matrices of Lie($G_2$)

The matrices of Lie($g_2$) as computed in Section 7.1.1:

$$\Lambda_1 = \begin{pmatrix} 0 & 0 & 0 & 0 & 0 & -1 & 0 \\ 0 & 0 & 0 & 0 & 0 & 0 & -1 \\ 0 & 0 & 0 & 0 & 0 & 0 & 0 \\ 0 & 0 & 0 & 0 & 0 & 0 & 0 \\ 0 & 0 & 0 & 0 & 0 & 0 & 0 \\ 1 & 0 & 0 & 0 & 0 & 0 & 0 \\ 0 & 1 & 0 & 0 & 0 & 0 & 0 \end{pmatrix}, \quad \Lambda_2 = \begin{pmatrix} 0 & 1 & 0 & 0 & 0 & 0 & 0 \\ -1 & 0 & 0 & 0 & 0 & 0 & 0 \\ 0 & 0 & 0 & 0 & 0 & 0 & 0 \\ 0 & 0 & 0 & 0 & 0 & 0 & 0 \\ 0 & 0 & 0 & 0 & 0 & 0 & 0 \\ 0 & 0 & 0 & 0 & 0 & 0 & -1 \\ 0 & 0 & 0 & 0 & 0 & 1 & 0 \end{pmatrix},$$

$$\Lambda_3 = \begin{pmatrix} 0 & 0 & 0 & 0 & 0 & 0 & -1 \\ 0 & 0 & 0 & 0 & 0 & 1 & 0 \\ 0 & 0 & 0 & 0 & 0 & 0 & 0 \\ 0 & 0 & 0 & 0 & 0 & 0 & 0 \\ 0 & 0 & 0 & 0 & 0 & 0 & 0 \\ 0 & -1 & 0 & 0 & 0 & 0 & 0 \\ 1 & 0 & 0 & 0 & 0 & 0 & 0 \end{pmatrix}, \quad \Lambda_4 = \begin{pmatrix} 0 & 0 & 0 & 0 & -1 & 0 & 0 \\ 0 & 0 & 0 & 0 & 0 & 0 & 0 \\ 0 & 0 & 0 & 0 & 0 & 0 & -1 \\ 0 & 0 & 0 & 0 & 0 & 0 & 0 \\ 1 & 0 & 0 & 0 & 0 & 0 & 0 \\ 0 & 0 & 0 & 0 & 0 & 0 & 0 \\ 0 & 0 & 1 & 0 & 0 & 0 & 0 \end{pmatrix},$$

$$\Lambda_5 = \begin{pmatrix} 0 & 0 & 1 & 0 & 0 & 0 & 0 \\ 0 & 0 & 0 & 0 & 0 & 0 & 0 \\ -1 & 0 & 0 & 0 & 0 & 0 & 0 \\ 0 & 0 & 0 & 0 & 0 & 0 & 0 \\ 0 & 0 & 0 & 0 & 0 & 0 & -1 \\ 0 & 0 & 0 & 0 & 0 & 0 & 0 \\ 0 & 0 & 0 & 0 & 1 & 0 & 0 \end{pmatrix}, \quad \Lambda_6 = \begin{pmatrix} 0 & 0 & 0 & 0 & 0 & 0 & 0 \\ 0 & 0 & 0 & 0 & -1 & 0 & 0 \\ 0 & 0 & 0 & 0 & 0 & -1 & 0 \\ 0 & 0 & 0 & 0 & 0 & 0 & 0 \\ 0 & 0 & 1 & 0 & 0 & 0 & 0 \\ 0 & 0 & 0 & 1 & 0 & 0 & 0 \\ 0 & 0 & 0 & 0 & 0 & 0 & 0 \end{pmatrix},$$

$$\Lambda_7 = \begin{pmatrix} 0 & 0 & 0 & 0 & 0 & 0 & 0 \\ 0 & 0 & 1 & 0 & 0 & 0 & 0 \\ 0 & -1 & 0 & 0 & 0 & 0 & 0 \\ 0 & 0 & 0 & 0 & 0 & 0 & 0 \\ 0 & 0 & 0 & 0 & 0 & -1 & 0 \\ 0 & 0 & 0 & 0 & 1 & 0 & 0 \\ 0 & 0 & 0 & 0 & 0 & 0 & 0 \end{pmatrix}, \qquad \Lambda_8 = \frac{1}{\sqrt{3}} \begin{pmatrix} 0 & 0 & 0 & 0 & 0 & 0 & -1 \\ 0 & 0 & 0 & 0 & 0 & -1 & 0 \\ 0 & 0 & 0 & 0 & 2 & 0 & 0 \\ 0 & 0 & 0 & 0 & 0 & 0 & 0 \\ 0 & 0 & -2 & 0 & 0 & 0 & 0 \\ 0 & 1 & 0 & 0 & 0 & 0 & 0 \\ 1 & 0 & 0 & 0 & 0 & 0 & 0 \end{pmatrix},$$

$$\Lambda_9 = \sqrt{\tfrac{2}{3}} \begin{pmatrix} 0 & 0 & 0 & 1 & 0 & 0 & 0 \\ 0 & 0 & 0 & 0 & 1 & 0 & 0 \\ 0 & 0 & 0 & 0 & 0 & -1 & 0 \\ -1 & 0 & 0 & 0 & 0 & 0 & 0 \\ 0 & -1 & 0 & 0 & 0 & 0 & 0 \\ 0 & 0 & 1 & 0 & 0 & 0 & 0 \\ 0 & 0 & 0 & 0 & 0 & 0 & 0 \end{pmatrix}, \qquad \Lambda_{10} = \sqrt{\tfrac{2}{3}} \begin{pmatrix} 0 & 0 & 0 & 0 & -1 & 0 & 0 \\ 0 & 0 & 0 & 1 & 0 & 0 & 0 \\ 0 & 0 & 0 & 0 & 0 & 0 & 1 \\ 0 & -1 & 0 & 0 & 0 & 0 & 0 \\ 1 & 0 & 0 & 0 & 0 & 0 & 0 \\ 0 & 0 & 0 & 0 & 0 & 0 & 0 \\ 0 & 0 & -1 & 0 & 0 & 0 & 0 \end{pmatrix},$$

$$\Lambda_{11} = \sqrt{\tfrac{2}{3}} \begin{pmatrix} 0 & 0 & 0 & 0 & 0 & 1 & 0 \\ 0 & 0 & 0 & 0 & 0 & 0 & -1 \\ 0 & 0 & 0 & 1 & 0 & 0 & 0 \\ 0 & 0 & -1 & 0 & 0 & 0 & 0 \\ 0 & 0 & 0 & 0 & 0 & 0 & 0 \\ -1 & 0 & 0 & 0 & 0 & 0 & 0 \\ 0 & 1 & 0 & 0 & 0 & 0 & 0 \end{pmatrix}, \qquad \Lambda_{12} = \sqrt{\tfrac{2}{3}} \begin{pmatrix} 0 & 1 & 0 & 0 & 0 & 0 & 0 \\ -1 & 0 & 0 & 0 & 0 & 0 & 0 \\ 0 & 0 & 0 & 0 & 0 & 0 & 0 \\ 0 & 0 & 0 & 0 & -1 & 0 & 0 \\ 0 & 0 & 0 & 1 & 0 & 0 & 0 \\ 0 & 0 & 0 & 0 & 0 & 0 & 1 \\ 0 & 0 & 0 & 0 & 0 & -1 & 0 \end{pmatrix},$$

$$\Lambda_{13} = \sqrt{\tfrac{2}{3}} \begin{pmatrix} 0 & 0 & -1 & 0 & 0 & 0 & 0 \\ 0 & 0 & 0 & 0 & 0 & 0 & 0 \\ 1 & 0 & 0 & 0 & 0 & 0 & 0 \\ 0 & 0 & 0 & 0 & 0 & -1 & 0 \\ 0 & 0 & 0 & 0 & 0 & 0 & -1 \\ 0 & 0 & 0 & 1 & 0 & 0 & 0 \\ 0 & 0 & 0 & 0 & 1 & 0 & 0 \end{pmatrix}, \qquad \Lambda_{14} = \sqrt{\tfrac{2}{3}} \begin{pmatrix} 0 & 0 & 0 & 0 & 0 & 0 & 0 \\ 0 & 0 & 1 & 0 & 0 & 0 & 0 \\ 0 & -1 & 0 & 0 & 0 & 0 & 0 \\ 0 & 0 & 0 & 0 & 0 & 0 & -1 \\ 0 & 0 & 0 & 0 & 0 & 1 & 0 \\ 0 & 0 & 0 & 0 & -1 & 0 & 0 \\ 0 & 0 & 0 & 1 & 0 & 0 & 0 \end{pmatrix}.$$

## Notes

1.   That is, it is the sum of simple roots with largest possible non-negative coefficients.

2.   recall that the surface of a sphere $S^{2d-1}$ of radius 1 is $2\frac{\pi^d}{(d-1)!}$

3.   The vector space $\mathbb{K} \simeq \mathbb{C}^n$ is the space of all complex valued Weyl invariant functions on $R$, $m$ equals the numbers of conjugacy classes of roots in $R$ and elements of $\mathbb{K}$ are called multiplicity functions on $R$. The notation $k_\alpha$ denotes the evaluation of $k \in \mathbb{K}$ on $\alpha \in R$.

4.   in general no requirements are done on the dimensions.

5.   We are grateful to S. Pigola for explaining us these points.

6.   Indeed, on $T^r$ there is the adjoint action of the normalizer $N$: $T^r \to (T^r)^N \subseteq T^r$. Moreover, $N/T^r = W$ is the Weyl group. Since the invariant measure restricted to the torus is just $d\mu_{T^r} = \prod_{i=1}^r ds_i$, we see that the action of the Weyl group sends $T^r$ isometrically onto itself. Thus, the cube is divided in equivalent sectors by the Weyl group action. The maximal number of such sectors is thus $|W|$, the cardinality of the Weyl group. More precisely, the adjoint action $\sigma : T^r \mapsto (T^r)^W$ is a surjective homomorphism over $T^r$, with a non-trivial kernel given by $\mathrm{Ker}\,\sigma \simeq \Lambda_W / \Lambda_R$, the quotient between the weight lattice $\Lambda_W$ with regard to the root lattice $\Lambda_R$. This lattice is isomorphic to the center $Z$ of the (covering) group. Then, we find that the number of cells in the cube is $\nu = \frac{|W|}{|\Lambda_W/\Lambda_R|} = \frac{|W|}{|Z|}$. We will see another way to compute the number of cells.

7.   Notice that we cannot use associativity in general, but the reader can check that $a(bc)$ satisfies associativity if two among $a, b, c$ are equal. For example if associativity would be true we would have $[e_4 e_1, e_2] = (e_4 e_1)e_2 - e_2(e_4 e_1) = (e_4 e_1)e_2 - (e_2 e_4)e_1 = (e_4 e_1)e_2 + (e_4 e_2)e_1 = e_4(e_1 e_2 + e_2 e_1) = 0$, which is wrong since $e_4 e_1 = e_7$ and $[e_7, e_2] = -2e_5$.

8.   with respect to the action of $SU(n+1)$.

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
