# Peer review of "Compact Lie Groups, Generalised Euler Angles, and Applications"

_universe, doi:10.3390/universe8100492_

Round 1
Reviewer 1 Report
The article deals with the problem on how to define generalized Euler angles in compact Lie groups.
The presentation is very extensive with many details explained. In general, it is a solid work. However, most of the subject seems quite familiar to me: For example, the definition of Euler angles in SO-and U-groups was dealt with in the 1970's. Just consult the following references:
1) R.L Anderson and K.B. Wolf, JMP 11 (No. 14 (1970), 3176
2) K.B. Wolf, JMP 12 (No. 2) (1971), 197
3) K.B. Wolf, JMP 13 (No. 10) (1972), 1634
Also the theorems on page 19 and their applications are well known. Just consult the book of R. Gilmore, "Lie groups, Lie algebras and some of their applications", J&Wiley 1974. The rules of Theorem 2 are known as the Baker-Campbell-Hausdorff formulas and found numerous applications in physics. The application to nuclear physics is not new to me, even, the use of the techniques cited in the above references have found a standard place in nuclear physics. I.e., I do not see in hat the results present are of new use. The authors have to explain this part and convince me.
I am not sure (as always) if I do not miss something. Therefore, I ask the authors to convince me what are the new elements of the manuscript, which are really new and in what sense these elements are of advantage compared to the ones already known and in use.
I also recommend that the authors read again the manuscript carefully and eliminate grammatical errors and misprints. For example (and there are many more), "generalised" is correctly written as "generalized". On page 3 "but she will" -> "but he will"; page 4 "onece" -> "once, etc. May be a spell check could help. But also often a sentence is wrongly structured and difficult to understand. I recommend that a different person revises this manuscript.
To resume: I Can only recommend this article for publication if the authors respond to my concerns convincingly
Reviewer 2 Report
Report on the article "Compact Lie groups, generalized Euler, and applications" by S.L. Cacciatori and A. Scotti.
The authors review a large variety of interesting topics that originate from compact Lie groups. These include symmetric embeddings, maximal subalgebras, parametrizations by Euler angles for SU(2) and SU(3) in split and non-split from, the Dyson and Selberg integrals, and the cohomology of compact group. They also review the classification of complex semi-simple Lie algebra by root systems and Dynkin diagrams, in each case their possible real forms represented by Sakate diagrams, and the classification of globally symmetric spaces arising as coset spaces M=G/K. An interesting aspect is the geometric interpretation of the Dyson integral in section 6. In the same way section 7 gives an explicit and transparent construction of the exceptional group G_2 as the automorphism group of the octonions O, together with an Euler angle parametrization of G_2. The 14 basis elements \Lambda_1,...,14 of the exceptional Lie algebra G_2 are given in Appendix B in the form of antisymmetric 7x7 matrices, which is material that can be hardly found in the literature. At that point the authors suggest to readers to compute the 12 roots and two simple roots. But then a hint for choosing the 2-dimensional Cartan subalgebra would be helpful.
One page 3, third paragraph, the statements "Moreover, su(2) is not maximal" and "We will call su(2) the largest maximal subalgebra" seem contradict each other. Presumably, su(2) should just be replaced by u(2).
Round 2
Reviewer 1 Report
After having read the comments of the authors and the changes
made in the new version of the manuscript, I recommend this
manuscript for publication. The authors made a quite convincing
case of what is new in the mansucript.